# Comparison of different droplet measurement techniques in the Braunschweig Icing Wind Tunnel

Inken Knop[1], Stephan Bansmer[1], Valerian Hahn[2,3], Christiane Voigt[2,3]

[1]Institute of Fluid Mechanics, Technische Universität Braunschweig, 38108 Braunschweig, Germany

[2]Deutsches Zentrum für Luft- und Raumfahrt (DLR), Institute of Atmospheric Physics, 82234 Wessling, Germany

[3]Institute of Atmospheric Physics, University Mainz, 55881 Mainz, Germany

*Correspondence to*: Inken Knop (i.knop@tu-braunschweig.de)

**Abstract.** The generation, transport and characterisation of supercooled droplets in multiphase wind tunnel-test facilities is of great importance for conducting icing experiments and to better understand cloud microphysical processes such as coalescence, ice nucleation, accretion and riming. To this end, a spray system has been developed, tested and calibrated in the Braunschweig Icing Wind Tunnel. Liquid droplets in the size range of 1 to 150 µm produced by pneumatic atomizers were accelerated to velocities between 10 and 40 m s$^{-1}$ and supercooled to temperatures between 0 and -20 °C. Thereby, liquid water contents between 0.07 and 2.5 g m$^{-3}$ were obtained in the test section. The wind tunnel conditions were stable and reproducible within 3% standard variation for median volumetric diameter (MVD) and 7% standard deviation for liquid water content (LWC). Different instruments were integrated in the icing wind tunnel measuring the particle size distribution (PSD), MVD and LWC. Phase Doppler Interferometry (PDI), laser spectroscopy with a Fast Cloud Droplet Probe (FCDP) and shadowgraphy were systematically compared for present wind tunnel conditions. MVDs measured with the three instruments agreed within 15% in the range between 8 µm and 35 µm, and showed high coefficients of determination (R²) of 0.985 for FCDP and 0.799 for shadowgraphy with respect to PDI data. Between 35 and 56 µm MVD, the shadowgraphy data exhibit a low bias with respect to PDI. The instruments' trends and biases for selected droplet conditions are discussed. LWCs determined from mass flow calculations in the range 0.07 – 1.5 g m$^{-3}$ are compared to measurements of the bulk phase rotating cylinder technique (RCT) and the above mentioned single particle instruments. For RCT, agreement to the mass flow calculations of approximately 20% in LWC was achieved. For PDI 84% of measurement points with LWC < 0.5 g m$^{-3}$ agree to mass flow calculations within a range of ± 0.1 g m$^{-3}$. Using the different techniques, a comprehensive wind tunnel calibration for supercooled droplets was achieved, which is a prerequisite to provide well characterized liquid cloud conditions for icing tests for aerospace, wind turbines and power networks.

## 1 Introduction

Supercooled water droplets cause icing of aircraft (Poots et al., 2000), helicopters (Kreeger et al., 2015), wind turbines (Battisti, 2015), and power networks (Farzaneh, 2008). As numerical icing codes are now widely used in the design and certification stages, the need for reliable experimental validation increases. The precise detection of the microphysical particle properties

and the liquid water content (LWC) of droplet distributions produced by spray systems in wind tunnel test facilities thereby is of great importance to improve ice accretion models. Besides icing research, other technical applications of spray systems, such as fuel sprays (Bossard and Peck, 1996), agricultural sprays (Tuck et al., 1997), or spray painting (Snyder et al., 1989), are of interest for related industry and research.

Various measurement techniques that differ in terms of the underlying physical principles and the probe design are currently used to characterize droplet clouds. One way to classify these is the differentiation between integrating systems investigating liquid clouds as entities and single-particle instruments (Brenguier et al., 1998). Another possible criterion distinguishes between intrusive and non-intrusive systems (Tropea, 2011). Three types of measurement techniques allow to measure the total mass of an ensemble of liquid droplets: systems that calculate the LWC on the basis of single droplet size measurements
(e.g., Fast Cloud Droplet Probe (FCDP), Phase Doppler Interferometry (PDI)), hot-wire methods (e.g., King LWC probe, Nevzorov probe), and ice accretion methods (e.g., rotating cylinder technique (RCT), icing blade) (Ide, 1999). A comprehensive overview of available techniques for cloud measurements is given by Baumgardner et al. (2017) including results from previous methodological papers (Tropea, 2011; Fansler and Parrish, 2015; Linne, 2013).

There are numerous icing wind tunnels worldwide that were used for intercomparison of droplet measurement techniques in
the past, including the NASA Glenn Research Center Icing Tunnel (Ide and Oldenburg, 2001) and the Altitude Icing Wind Tunnel of the National Research Council of Canada (Strapp and Schemenauer, 1982). Here we show results from droplet measurements performed in the Braunschweig Icing Wind Tunnel (BIWT) (Bansmer et al., 2018) initially designed to provide large supercooled droplets (median volumetric diameter (MVD) ≈ 80µm) and ice particles for icing experiments in mixed phase and ice crystal conditions. In 2016 the wind tunnel has been further upgraded to introduce also small liquid droplets,
relevant e.g. for research on wind turbine icing and Appendix C (MVD <50 µm, FAA) inflight icing conditions. During the extensive calibration of the new spray system different measurement techniques were integrated into the wind tunnel to measure the particle size distribution (PSD) and the LWC. Measurements of the PSD of liquid particle ensembles with droplet sizes < 150 µm were performed with the PDI, the FCDP and shadowgraphy and results are compared within the instrumental measurement ranges. In addition, the LWC detected with the PDI, the FCDP, and the RCT are compared and related to LWC
calculations, based on injected water mass flow and wind tunnel flow velocity. Thereby the laboratory environment of the wind tunnel provides a homogenous ensemble of water droplets at a given constant target speed.

Similar to the experiments conducted here, Ide (1999) compared different LWC measurement techniques in the NASA Glenn Research Center Icing Tunnel with spray MVD in the range 10 to 270 µm and velocities 22 to 112 m s$^{-1}$. The instruments tested in 1999 were the icing blade, a single rotating cylinder, the Johnson-Williams and CSIRO-King hot-wire probes, the
Nevzorov LWC/TWC (Total Water Content) probe and the LWC calculated from the combined droplet distributions of two droplet sizing probes – the FSSP (range 2-47 µm) and the Optical Array Probe (range 15-450 µm) OAP Particle Measuring Systems, Inc. of Boulder, Colorado. The LWC calculated from the droplet distributions measured with OAP and FSSP was found to be overestimated. Cober et al. (2012) published a comparison of different LWC measurement techniques for large supercooled droplets in flight tests. They evaluated a Rosemount icing detector from Goodrich Corporation, which can measure

the LWC when environmental conditions lead to a temperature below the Ludlam limit. Furthermore, two FSSP and three different 2D-imaging systems FSSP (3-45 µm and 5-95 µm), 2D-C (25-800 µm), 2D-G (25-1600 µm), 2D-P (20-6400 µm), were installed during their cloud research flights. Later on, the results of these publications are used for comparison purposes.

This paper describes the experimental setup of the BIWT, the new designed spray system and its performance. After the description of the individual measurement techniques, results for MVD and LWC are discussed in sight of the different measurement methods. The outlook presents a short summary, future research topics and plans for a second update of the spray system to generate bimodal PSDs.

## 2 Experimental Setup

The following chapter contains some basic information regarding the wind tunnel setup, experimental boundary conditions and statistical estimators to evaluate our results. Furthermore, the repeatability of the wind tunnel for the aerothermal behavior as well as the droplet cloud will be presented with example test cases. The design of the test matrix will be discussed at the end of the chapter.

### 2.1 Wind Tunnel Description

The BIWT is a state-of-the-art academic research facility that complies with the SAE ARP5905 requirements and has been actively engaged in several international projects in collaboration with multiple aerospace agencies and industries. A detailed overview of the BIWT is given by Bansmer et al. (2018). The basic design is a closed-loop wind tunnel with a 0.5 m x 0.5 m cross-sectional area at the test section with adjustable velocities between 10 and 40 m s$^{-1}$. The static air temperature can be controlled between -25 °C and +30 °C. In addition to the injection of water droplets through a spray system, it is also possible to introduce a cloud of ice particles to simulate different conditions of atmospheric icing in the test section. The BIWT is not pressurized and yields Reynolds numbers up to $2 \cdot 10^6$ at its full speed. To further extend the operational envelope of the tunnel, numerous scaling methods based on similitude of geometry, droplet trajectories and the impingement heat transfer are available. A comprehensive description of the scaling methods can be found in Anderson (2004). In the present study, we do not apply any scaling to the results in order to avoid introducing additional sources of uncertainty to our results.

The spray system of the tunnel consists of 30 pulsed air-assist atomizers (see Fig. 1) from Spraying Systems Co (PulsaJet AB10000JJAU) with fluid cap PFJ-08-50 (diameter of the final liquid discharge orifice 0.2 mm) and air cap PAJ-73-1-60 (diameter of the final orifice outlet of 1.5 mm). The general random nature of the atomization process results in sprays with a wide spectrum of droplet sizes with the mean value depending on the supply pressure (Lefebvre and McDonell, 2017). The electrically-actuated atomizers are controlled by the AutoJet Spray Controller. The pulse width modulated (PWM) flow control enables an independent change of liquid mass flow at constant supply pressure (and therefore a relatively constant droplet size). The atomizers are switched on and off up to 10000 times a minute, making the spray appear constant for the purpose of

icing research. Furthermore, the electrically-actuated spray nozzles are closed if not in use, even if the system is already pressurized. This leads to a smaller delay from starting the spray to steady-state condition of the fully developed droplet size distribution. Demineralized water is used for droplet generation with a very low level of contamination to avoid clogging of the spray nozzles and freezing out of the droplets in the cold airflow. All components of the supply structure outside the tunnel were chosen with regard to small pressure losses and compatibility of materials for the demineralized water. Separate valves for every spray bar enable a selective usage of only a specific part of the atomizers. A separate management system to control every atomizer individually has been implemented to turn off specific atomizers (e.g., in the case of low flow velocities and high probability of icing of the wind tunnel walls, the spray atomizers near the wind tunnel walls can be stopped). Thermal volume flowmeters measure the averaged water flow rate for each spray bar, thus providing a hint when nozzles clog or freeze over. The actuation of the electrically controlled pressure regulators for the water and the compressed air, all valves, and the control unit of the PWM-flow system are integrated into the wind tunnel software, providing the user with remote control of the whole spray system. All aerothermal characteristics, like airflow uniformity, turbulence intensity, and total temperature of the wind tunnel flow, comply with SAE ARP5905 specifications (Bansmer et al., 2018).

The droplet measurements were conducted along the centerline in the wind tunnel test section 4 m downstream from the spray system. The bluff body shape of the spray bars (see Fig. 1) promotes a homogenous spatial dispersion of droplets in the airflow. It has been shown numerically that droplets up to a diameter of 100 µm have almost no slip to the wind tunnel speed (Bansmer et al., 2018), leading to the assumption that the droplet velocity in the test section agrees well with the adjusted air speed. The hypothesis is supported by an example PDI dataset shown in Fig. 2. 88% of the measured droplets have a velocity equal to the mean cloud velocity with 1% deviation. The high number of droplets is indicated by the darker shade and the density of the dots in Fig. 2. The plotted distribution shows a slight asymmetry towards larger droplets having smaller velocities. Especially droplets with diameters larger than 100 µm show a negative slip. This is due to their higher resistance. The typical PSDs taken into account in this study have no large droplets, so this effect can be neglected in the interpretation of the results.

## 2.2 Parameters and Statistical Quantities for Comparison

Procedures for determining appropriate sample size, size class widths, and characteristic droplet sizes for the characterization of sprays were applied according to ASTM E799-03 (Practice for Determining Data Criteria and Processing for Liquid Drop Size Analysis). In icing research, the histogram of the number of droplets with diameters between $D \pm \Delta D/2$ is used most frequently, together with cumulative curves of the liquid cloud volume. In this study, the characteristic diameters of the cumulative volume curve like the MVD (or DV0.5) and the 10 and 90 percentiles (DV0.1 and DV0.9) are used to describe the droplet distributions as shown in Fig. 3 for a test delivering a cloud ensemble with an MVD of 11.8 µm. Droplet distributions in the atmosphere typically follow a log-normal behaviour (Langmuir and Blodgett, 1961). The atomization of fluids in laboratory setups may lead to different PSD such as normal, Nukiyama–Tanasawa, Rosin–Rammler, modified Rosin-Rammler, and upper-limit distributions (Lefebvre and McDonell, 2017).

Another important variable for icing research is the already mentioned LWC, which represents the mixing of the available mass of water ($m_{Water}$) within a defined air volume ($V_{air}$):

$$LWC = \frac{m_{Water}}{V_{air}} \tag{1}$$

The repeatability of measurements is characterized based on the coefficient of variation ($\sigma$) (i.e., the standard deviations over several repeated measurements (n) normalized by the mean values) as in the following equation, exemplary for the MVD:

$$\sigma = \frac{1}{\overline{MVD}} \sqrt{\frac{\sum(MVD - \overline{MVD})^2}{n}} \tag{2}$$

To describe the consistency of the results from different measurement techniques, we use the mean absolute value of the relative error $|E_{compare-reference}|$. Therefore, the sum over all differences in the value of interest between the considered and the reference technique, normalized by the reference value, is divided by the number of comparable measurements. The absolute value of the difference avoids a cancellation of positive and negative errors. For the MVD the mean absolute value of the relative error is calculated with the PDI as reference value according to Eq. (3). For the evaluation of the LWC we use the value based on measured water flow rate (WFR) and circulating air flow as reference and calculate the mean absolute value of the relative error according to Eq. (4).

$$\left|E_{compare-PDI}\right| = \frac{1}{n}\sum \frac{|MVD_{compare} - MVD_{PDI}|}{MVD_{PDI}} \tag{3}$$

$$\left|E_{compare-WFR}\right| = \frac{1}{n}\sum \frac{|LWC_{compare} - LWC_{WFR}|}{LWC_{WFR}} \tag{4}$$

### 2.3 Wind Tunnel Repeatability and Uncertainty Estimations

For the analysis of wind tunnel repeatability and an uncertainty estimation, the pure aerodynamic performance of the tunnel and the stability of the liquid atomizers that produce the droplet cloud need to be considered.

Regarding the aerodynamic repeatability, temperature and airspeed of the flow can be regulated with the required accuracy according to SAE ARP5905, however, the humidity and static pressure cannot. The static pressure in the test section is hence dependent on the ambient pressure, whereas the humidity in the test section is governed by the duration of water injection. The relative humidity of the two-phase flow quickly increased after the first few tests at the beginning of a measurement day to $> 90\%$.

For a better description of the temporal stability of the wind tunnel and spray system test conditions, Fig. 4 shows a representative 15-minute test record. The upper three diagrams describe the quality of the wind tunnel flow and the lower three diagrams the spray system. All measured flow quality parameters meet the requirements of the SAE ARP5905 for the temporal stability along the tunnel centerline and thereby the measurement positions of the intercomparison tests: The flow velocity fluctuates by a maximum of 1.6%. The tunnel temperature varies by less than 0.5 °C and the relative humidity, as the only non-adjustable variable, varies by max. 1.5% over a time period of 15 minutes. The SAE ARP5905 allowable deviation criteria for the velocity and temperature being ±2% and ±0.5°C respectively measured with instruments having an uncertainty range

of ±1% and ±0.5°C. The precision limits of velocity and temperature are computed for a sample run as defined in Coleman and Steele (1995) and AGARD-AR-304. Their values lower than 0.01 indicate negligible temporal fluctuations in the aerodynamic performance of the tunnel.

Next, the stability of the liquid atomization is considered. The constant supply system parameters are a prerequisite for a temporally constant atomization process and thus a temporally constant droplet cloud in the test section. By monitoring the water mass flow, it can be determined that, despite previous pressurization of the pipes, approx. 15 seconds are required until the volume flow stabilizes. Thereafter, air and water pressures fluctuate in the supply system of the spray atomizers on average by 0.04 bar (3%) and 0.03 bar (1%), respectively. To estimate the influence of these fluctuations on the water flow rate, a similar spraying system with a different air cap is used that incorporates a high accuracy Coriolis flow meter (manufacturer specified accuracy 0.2%). A parametric model for LWC is developed using the pressure fluctuations as input and the Coriolis flow meter data as output. Performing an uncertainty propagation analysis and assuming a 95% confidence interval, the root of the sum of the squares (RSS) uncertainty bounds of the LWC can be conservatively considered to be within the 10% limit. In the BIWT setup the water volume flow is measured with one thermal volumetric flow meter per row of six atomizers. Due to the very low total water volume flow through every thermal volumetric flow meter (down to less than 10 ml min$^{-1}$ per row) and the pulsation of the nozzles, the uncertainty of the volume flow measurement is approximately 20%.

For the evaluation of the deviations between the different measuring techniques, an investigation of the repeatability of the droplet cloud in the wind tunnel is needed. Due to the afore mentioned small pressure fluctuations in the supply system and slight fluctuations in the wind tunnel velocity, minor variations in the droplet size distribution and the LWC may occur even with the same settings for all wind tunnel parameters. In addition, there is the non-deterministic atomization process at the pneumatic atomizers themselves (Liu et al., 2005) leading to small temporal variations in the droplet cloud. To determine the size of these variations for the MVD in the BIWT reference measurements have been performed with the PDI and selected measurement points have been repeated with exactly the same experimental setup. The results of some of these tests are shown in Fig. 5. The repeatability of MVD shows a coefficient of variation, according to Eq. (2) of $\sigma = \pm 3\%$, including uncertainties in the wind tunnel and the measurement setup.

To again underline the good temporal stability of the spray system, the PSD and corresponding cumulative mass fractions for the first 1000, middle 1000 and last 1000 droplets of a 15 minutes long single PDI measurement are plotted in Fig. 6 (right). The average acquisition rate of the measurement was 338 droplets per second. The PSDs agree very well with each other, except those of the first 1000 droplets. This is due to the transient behaviour in the first seconds of the spray ramp-up, where the atomization has not stabilized yet, what can be also seen in the water mass flow in Fig. 4.

The inherent complex interactions in the spray process makes it challenging to obtain the actual value of the distribution, therefore the uncertainty bounds of the spray were not ascertained in the present study. The high $R^2$ values in Fig. 5 indicate a promising repeatability of the spray system, facilitating a reliable relative comparison of PDI, FCDP and shadowgraphy. For the test points shown in Fig. 5, the LWC based on the water mass flow was also investigated. This resulted in a coefficient of variation of the LWC in repeated measurements of $\sigma = \pm 7\%$, indicating altogether a good repeatability of the wind tunnel

conditions with respect to particle size and LWC. The afore mentioned value of the RSS uncertainty bounds is slightly higher than the LWC repeatability characteristics and can be explained by the unsteadiness of the atomization process.

## 2.4 Test Matrix

The test matrix for the measurements was designed to test each independent variable separately. To this end, the droplet diameters were first varied using different combinations of air and water pressure. During these tests the duty cycle of the nozzles and the velocity and temperature of the tunnel were not changed. Then, the duty cycle was varied for selected pressure

combinations in order to classify its influence on the MVD and LWC. Furthermore, the flow velocity was changed from 10 up to 40 m s$^{-1}$ with exactly the same spray system settings, which should lead only to changes in LWC. Finally, the temperature was varied, which theoretically should neither have a noticeable influence on the droplet size nor the LWC. In addition, some parameters of every measurement technique were varied depending on the individual system. These tests were not further discussed here, since the investigations of the techniques themselves have been widely done in literature (see Section 3).

Overall, the comparison is made for sprays in the MVD range of 8 to 56 µm (PDI size being the reference) and corresponding LWC from 0.07 – 2.5 g m$^{-3}$ (reference LWC from water flow rate). The here tested measurement ranges as well as some comparable key parameters of the different measurement techniques are summarized in Table 4. The detailed test conditions for the comparison of the MVD and LWC can be found in the supplementary material.

Shadowgraphy, PDI, and FCDP measurements of droplet PSDs and LWC were performed in test campaigns in 2017 and some

210 PDI measurements were repeated in 2019. The RCT measurements were performed in summer 2018. The static pressure in the test section varied during these measurements between 990 hPa and 1007 hPa. Most of the shadowgraphy, PDI and FCDP experiments were conducted at -5°C to avoid fogging of the wind tunnel windows and instrument optics.

According to the assumption that all droplets are accelerated with the airflow in the long wind tunnel nozzle, the downstream position of the measurement volume in the test section should neither significantly affect the droplet diameter nor the droplet

velocity or LWC. Depending on the mechanically required window configuration of the test section for every measurement setup, the downstream coordinate of the probe volume differed slightly. The measurement position of the PDI (or the RCT) and the shadowgraphy (or the FCDP) varied by a maximum of 220 mm (see Fig. 7) in downstream position.

To determine the desired number of droplets for a test point, one exemplary test point was measured over 15 minutes with the PDI system at constant test conditions. In a typical droplet size distribution in a spray in the wind tunnel, large droplets occur

by orders of magnitude less frequent than small droplets (Rudoff et al., 1993; McDonell and Samuelsen, 1996). Therefore, the choice of the number of droplets per test point is essential for a representative and comparable determination of the MVD. Fig. 6 (left) shows the dependence of the MVD on the number of droplets taken into account. Since the MVD is sensitive to large droplets, its stability is a good hint for a representative measurement point. Taking into account more than 10000 droplets for the test point results in less than 5% deviation from the mean value over 280000 droplets. This minimum number of droplets

was set as a target value for all experiments.

# 3 Measurement Techniques to determine PSD and LWC

Fig. 7 shows an overview of the different measurement setups in the BIWT. The following sections present the measurement instruments, their parametrization, as well as their inherent advantages and shortcomings. At the end of each section an estimation of the overall combined repeatability of the wind tunnel conditions and the precision of the treated measurement setup, based on repeated measurements is presented. The mean coefficients of variations (σ), according to Eq. (2), obtained from several repetition tests are summarized in Table 4. When using optical methods, particular attention must be paid to the correct description and interpretation of the sample area, the cross-sectional area perpendicular to the flow velocity where droplets are detected. The sample area is defined by the optical and electronic configuration of the instrument (Widmann et al., 2001).

## 3.1 Phase Doppler Interferometry

The Phase Doppler Interferometry (PDI) is a single-particle counter, single point, real-time, and non-intrusive measurement technique and an extension of the Laser Doppler Anemometry (LDA), initially described in 1972 by Farmer (Farmer, 1972). Since the early 1980s, i.a. Bachalo has further advanced the principle into the PDI (Bachalo and Houser, 1984). The basic principle of LDA and PDI is based on the detection of the characteristic refraction signal of a spherical particle passing through an interference fringe pattern created by two coherent intersecting laser beams. The velocity of the particle (LDA) can be determined via the Doppler difference frequency of the scattered light signal. The spatial phase shift between the different detectors contains the size information of the particle (Durst and Zaré, 1975; Bachalo and Houser, 1984; Cossali and Hardalupas, 1992). The droplet diameter is estimated from the linear relationship of the phase difference with the diameter, a remarkable advantage over other optical probes that are based on intensity and diameter relationship which is sensitive to light attenuation and contaminated optics. In the PDI system, the receiver lens is additionally spatially partitioned into several segments. The PDI theoretically only needs an initial factory calibration because the parameters responsible for the measurement results, like the laser wavelength, beam intersection angle, transmitter and receiver focal lengths, do not change within the lifetime of the system. Thus, PDI evolved as a common well characterized technique to measure spherical droplets in technical sprays, see Kapulla et al. (2007) and Jackson and Samuelsen (1987).

The PDI system used in this investigation is the 2D modular PDI from Artium Technologies Inc. It consists of an optical transmitter (diode-pumped solid-state laser), an optical receiver, Fourier-transform-based advanced signal analyzer (signal processors), a data management computer, and the AIMS system software. The PDI Transmitter has been used within two different setups: with a transmitter focal length of 350 mm in 2017 and 500 mm in 2019. The details of the used PDI system are summarized in Table 1.

Several early investigations of the PDI system have shown the effect of the photomultiplier tube (PMT) gain on the measurements (Bachalo et al., 1988; McDonell and Samuelsen, 1996). Thus, the PMT voltages were chosen carefully with regard to the expected diameter distribution and volume flux in the range of 300 to 390 Volts. The investigation of the

sensitivity of the PDI setup to user-controlled settings of McDonell and Samuelsen (1990) showed variations in the MVD of 5%. The signal processor was therefore operated with the settings chosen by the manufacturer's automatic setup for the tests discussed here, to not add an additional source of variation in the results.

In the evaluation of the PDI results for size distribution and LWC, the probe volume correction (PVC) described inter alia in Zhu et al. (1993) was considered for all measurements. This correction is based on the assumption that smaller particles passing a Gaussian-shaped probe volume have only a smaller area where they can be detected because of their lower scattering intensity (scattering light can be taken as being proportional to the square of the droplet diameter (McDonell and Samuelsen, 1996)). Small particles need to pass the maximum intensity in the center of the probe volume to produce scattering signals high enough to be detectable. Larger droplets can still be detected when they pass at the edge of the Gaussian-shaped probe volume. Using the transit time method (Zhu, 1993), the real probe volume for every size class is measured independently and used for correction of the size distribution afterwards.

The calculation of the LWC from the PDI measurements is based on the corrected volume mean diameter ($D_{30}^{cor}$) and the corrected droplet number concentration ($N_d^{cor}$), with the following formula (Widmann et al., 2001):

$$LWC = \frac{\pi}{6}\rho(D_{30}^{cor})^3 N_d^{cor} \tag{5}$$

The corrected volume mean diameter of the size distribution ($D_{30}^{cor}$) in Eq. (5) is calculated with the probe volume corrected counts ($c_i^{cor}$) per size bin i.

$$D_{30}^{cor} = \sqrt[3]{\frac{\sum_{i=1}^n c_i^{cor} d_i^3}{\sum_{i=1}^n c_i^{cor}}} \tag{6}$$

$$c_i^{cor} = c_i \left(\frac{PV_{max}}{PV_i}\right)\left(\frac{D(d_i)_{max}}{D(d_{max})_{max}}\right) \tag{7}$$

Where $d_i$ is the diameter of the $i^{th}$ droplet size class and $D(d_{max})_{max}$ is the effective diameter where the light intensity is sufficient for the largest droplet to be detected. $c_i$ is the uncorrected count in size class $i$. The probe volume corrected counts ($c_i^{cor}$) are related to the effective probe volume per size class ($PV_i$), determined by the afore mentioned transit time method $PV_{max}$ is the effective probe volume of the largest size class. For the calculation of the corrected droplet number concentration ($N_d^{cor}$), the ratio of the total particle transit time ($t_{tran(i,j)}$) and the total sample time ($t_{Tot}$) is divided by the probe volume ($PV_i$) for each particle size class. The index $i$ corresponds to size class and the index $j$ corresponds to the droplet occurrence-

$$N_d^{cor} = \frac{1}{t_{Tot}}\sum_i \frac{\sum_j t_{tran(i,j)}}{PV_i} \tag{8}$$

The PVC has the greatest effect on the smallest size classes. Their influence on the LWC, on the other hand, is very small as it is dominated by the presence of large droplets. In addition to the PVC, an intensity validation scheme, described by Bachalo (2000), was used. This procedure supplements the PDI principle with a validity check, in which the agreement between signal intensity and droplet diameter calculated from the burst distance is checked. Overall, the approach to determine the LWC from the droplet size distribution increases the measurement uncertainties compared to direct LWC measurement methods (Lance et al., 2010), which has been shown e.g. by McDonell et al. (1994) and Widmann et al. (2001). McDonell et al. (1994) find

variations in droplet concentration of up to 50%. Widmann et al. (2001) investigated the accuracy of LWC measurements from the PDI in an application with only low data rates and find a mean absolute value of the relative error of up to 26%. From these measurements it can be concluded that the droplet concentration is generally very sensitive to instrument operation and chosen settings. Because of the high number of influencing parameters, it is not surprising to see large variations in the results of re-runs (McDonell et al., 1994; Tropea, 2011). According to Bachalo et al. (1988) and Zhu et al. (1993), the calculation of the correct probe area is the primary source of error in the calculation of the volume flux.

The overall combined repeatability of the wind tunnel conditions and the precision of the PDI setup resulted over all tests with varied instrument settings and identical spray parameters in a mean coefficient of variation of the MVD of $\sigma = 5\%$. DV0.1 and DV0.9 behave in a similar way, with an average coefficient of variation of $\sigma = 7\%$. In the context of the measurements found in the literature mentioned above, the coefficient of variation determined here indicates an adequate design of the PDI system and the correct choice of system parameters. However, the average of the coefficient of variation over all repeatability measurements in DV0.99 is slightly greater ($\sigma = 14\%$). The larger variation in DV0.99 is quite plausible since the very small proportion of large droplets can be detected statistically less frequently (see Fig. 8) but has a large impact on DV0.99 (McDonell et al., 1994). For this reason, their detection is affected by larger fluctuations even in measurements with a high number of total measured droplets. The LWC results show considerably more variability. The measured average coefficient of variation of about $\sigma = 20\%$ is four times greater than the variations in the MVD measurements but rather small if compared to McDonell et al. (1994), Widmann et al. (2001) and Tropea (2011). According to Eq. 5 the LWC calculation of the PDI is proportional to the droplet number concentration ($N_d$) and to the third power of the corrected volume mean diameter ($D_{30}$). The present coefficients of variation of the representative droplet diameters thus can lead directly to 15-21% variation in the LWC. Adding the uncertainty of the droplet number concentration the average coefficient of variation of $\sigma = 20\%$ is coherent and comparatively small.

## 3.2 Fast Cloud Droplet Probe

The Fast Cloud Droplet Probe (FCDP) manufactured by SPEC Inc. is a single particle counter, which quantifies intensities of forward scattered light by particles passing through a laser beam to derive the particle's size and collate an overall number concentration. Forward scattering probes are generally used to detect microphysical properties of liquid clouds from research aircraft (Lawson et al., 2017; Woods et al., 2018, McFarquhar, et al. 2017).

The particle size is determined via the correlation between the scattering cross-section under the assumption of Mie-theory and the signal voltage measured at the signal detector. A qualifying detector confines a focal area along the laser beam. This Sampling Area (SA) together with the incident true airspeed in transit time direction yields the sample volume (SV). A calibration of the SA size was performed by means of a beam mapping using a droplet generator according to Lance et al. (2010) or Faber et al. (2018). Detected particles are resolved into 21 size bins ranging from 1.5 µm up to 50 µm including one

over-size bin, which was removed from further analyses. Bin widths range from 1.5 µm, in the two lowest bins, up to 4 µm for larger bin sizes. Some of the spray properties that can be derived from the measurements are the droplet number concentration ($N_d$), MVD and LWC. The operation principle of the instrument, as well as general sources of uncertainties for this class of instruments are described in detail by Lance et al. (2010), Baumgardner et al. (2017), Lawson et al. (2017), Woods et al. (2018) and Faber et al. (2018).

Lance et al. (2010) report a particle sizing accuracy of a recalibrated and modified CDP of at least 10% (mainly due to the coarse size resolution of the size bins), which is also found in Faber et al. (2018). Although the referenced probes both lack the FCDP's novel optics and electronics, sizing accuracies might be of the same order of magnitude. However, Baumgardner et al. (2017) also report a propagated sizing uncertainty for single-particle scattering probes in general of 10% to 50%, where the advanced correction methods of the FCDP as a probe of the latest generation allocate this instrument at the lower side. The FCDP used in this study has novel fast electronics, which partially minimizes coincidence effects by calculating coincidence correction functions based on transit time information and other data stored with each individual particle. Further reductions in propagated uncertainty in droplet number concentration can thus be achieved under application of filtering techniques, such as transit time and inter particle arrival time filter methods of each individual droplet during post processing. Baumgardner et al. (2017) present a propagated droplet number concentration uncertainty between 10%-30% for the entire ensemble of forward scattering probes, where the FCDP again might be classified among the lower end.

Unlike the PDI system used here, the FCDP was installed inside the test section of the wind tunnel with the sample volume placed in the undisturbed particle-laden flow at the center of the test section. Table 2 gives an overview of the main characteristics of the probe. For processing of $N_d$, the true airspeed (TAS) of the wind tunnel was also assumed as droplet velocity. The realization of high droplet number concentrations during our wind tunnel study and hence the increased probability of coincidence errors urges the use of a high DoF criterion, which is the ratio of qualifier to signal voltage of a detected droplet, in order to constrict the effective sample area and to limit coincidence effects. The SPEC manual recommends high DoF ratios for accurate particle sizing (FCDP SN6, SPEC 4/28/2017).

The calibration report also specifies a DoF ratio of 0.9 as the peak value for this specific probe. Initial variations of the DoF criterion support this recommendation.

An additional filtering method to further reduce coincident particles is provided by SPEC within a Matlab software module with which the theoretical full peak transit time of a droplet (TT) through a gaussian beam profile, depending on the droplet size and TAS, can be fitted to the measured TT versus size distribution using two fit parameter C1 and C3. Qualified scatter events outside the acceptance range of 25% beyond this theoretical TT to size curve are regarded as coincident and are such discarded (SPECinc. C1C3_V4 manual, SPEC inc. Data Processing Manual 2012).

High particle number concentrations as in some conditions produced by the wind tunnel facility can be encountered in the atmosphere in polluted low clouds (Flammant et al., 2018; Taylor et al., 2019), polluted convection (e.g. Braga et al., 2017b; Ceccini et al., 2017) or in young contrails (e.g. Voigt et al., 2011; Kaufmann et al., 2014; Kleine et al., 2018).

In total, more than 80 different spray conditions have been measured each for about 120 s.

The repeatability of the wind tunnel conditions together with the precision of the measurement setup has been investigated for the FCDP, similarly to the PDI-setup. On average, a coefficient of variation of $\sigma = 7\%$ in MVD was found for all repetition measurements. Similar values were found for DV0.1 and DV0.9. Due to the large width of the size intervals (bins) of the FCDP for large particles, the determination of DV0.99 on the basis of the FCDP data was not further evaluated. Taking into account the accuracies of the FCDP for monodispersed single droplets, as mentioned above, the here found coefficients of variation of the representative droplet diameters indicate a good repeatability of the new spray system of the BIWT. Like the PDI results, the LWC calculations from the FCDP also show a significantly higher coefficient of variation ($\sigma = 17\%$), inherent in the method of deriving the LWC from measurements of the particle's size, see Baumgardner (1983) and Tropea (2011).

### 3.3 Direct Imaging: Shadowgraphy

The idea of the shadowgraphy technique is to capture a high-resolution shadow image of a particle. In our study, a Litron Nano PIV–T double-pulsed laser is used as a light source. Its coherent light of 532 nm wavelength is diffused through a fluorescent plate, which illuminates the particles passing the system between the light source and camera. The spherical droplets are deflecting the incoming light wave, resulting in a particle shadow that is perceived from the observing camera. To obtain a high resolution for the droplet shadow images, 180 mm Tamron objective and magnification lenses (tele convertor 1,4X) were mounted in front of a PCO.4000 camera. The double-pulsed laser and the double-frame capability of the camera allow for the recording of short-time-separated pictures. This enables the droplet velocity computation by tracking the displacement of particles between two frames. The laser and camera are synchronized with an external programmable timing unit. Since the image acquisition rate of the camera is limited to approximately 2 frames per second, a long measurement time is required for a statistically robust result. For the correct interpretation of the measurement images, a prior calibration is necessary. The magnification factor of the optical array is determined by placing a transparent plate with a patterned array of dots (diameters from 10 µm to 200 µm) at the focal plane. Furthermore, a depth-of-field calibration is performed using the methodology of Kim and Kim (1994). The range in which droplets can be detected and correctly sized is limited by the image area of the camera chip, the depth-of-field of the optical system and the available light intensity. Finally, the shadow pictures are post-processed with an image analysis software (DaVis from LaVision), which determines the diameters of the shadow images in the field of view.

The image processing is performed on an inverted intensity image i.e. on the resultant of the shadow image subtracted from the background reference image (without particles). The subsequent particle detection is made relative to the difference between maximum and minimum of the inverted image, the noise can be eliminated with a careful selection of minimum area and maximum area, eccentricity and other thresholds. The detailed post-treatment of shadow images has been described by Kapulla et al. (2007) and Kapulla et al. (2006).

The evaluation of the shadowgraphy pictures is rather focused on the size distribution and not on the LWC because of high uncertainties in the probe volume and consequently the droplet number concentration. The hardware settings used in the

experiment conducted here are listed in Table 3. Because of the long measurement time for every test point (10 - 20 min), only 35 measurements in total were conducted. Among these 35 measuring points, there are many 2 to 3 times repeated measurements and measuring points with varied tunnel velocity but the same spray settings, leading to almost identical droplet size distributions in the evaluation (see test matrix in the supplementary material).

The combined influence of the precision of the shadowgraphy setup and the wind tunnel repeatability leads to an average variation of $\sigma = 8\%$ for the MVD, which is within the same order of magnitude compared to the afore-mentioned methods. According to Lefebvre and McDonell (2017), the imaging system developed by the Parker-Hannifin Corporation has a repeatability of 6% in the Sauter mean diameter range from 80 µm to 200 µm. Considering the significantly smaller droplets sizes here, the slightly higher coefficient of variation is plausible, as small droplets represent the more challenging task for direct imaging systems. Thus, the here measured variations indicate a well-chosen measurement setup and data post processing for the shadowgraphy technique.

### 3.4 Rotating Cylinder Technique

According to the SAE International Standard ARP5905 (Calibration and Acceptance of Icing Wind Tunnels), a rotating cylinder based on Stallabrass (1978) was designed and constructed for the BIWT. The rotation of the cylinder ensures a uniform ice build-up around the circular cross-section that provides aerodynamic consistency while accreting ice. If the speed of droplets, cylinder geometry, ice density, and collection efficiency (known droplet diameter) are known, the LWC can be calculated by the following formula:

$$LWC = \frac{\pi \cdot \rho_e}{\alpha_1 \cdot u_\infty \cdot t} \cdot \left[ \left( \frac{m_e}{\pi \cdot \rho_e \cdot l_c} + r_c^2 \right)^{0,5} - r_c \right], \tag{9}$$

where $\rho_e$ (assumed to be 880 kg m$^{-3}$) stands for the ice density, $m_e$ for the final accreted ice mass, $t$ for the icing time (selected with regard to the maximum allowed ice accumulation), $\alpha_1$ for the collection efficiency, and $l_c$ and $r_c$ for the length and the radius of the original cylinder, respectively. The calculation of the collection efficiency is based on the assumption of a monodisperse droplet distribution with the MVD as the diameter for all droplets.

In this measurement method, several assumptions that lead to uncertainties in the LWC results are made. These are based on the SAE ARP5905 uncertainties in bulk density of ice, the simplification of the droplet size distribution to one representative diameter (MVD) and the assumption of a fixed cylinder diameter in the calculation of collection efficiency.

With the density of accreted ice depending on several parameters (temperature, droplet velocity, etc. (Macklin, 1962; Jones, 1990)), a 12.5% error in the assumed bulk density of ice leads to 3% error in LWC, according to Stallabrass (1978). King (1985) reiterates the accuracy of the rotating cylinder measurements under 10%. The simplification to regard the entire droplet cloud as a monodisperse spray with only droplets of the diameter of the MVD enters the calculation of the collection efficiency. Early investigations have shown that, for example, the assumption of a monodisperse droplet size distribution instead of a Langmuir D distribution of the droplet size leads to an overestimation of the collection efficiency of 3.5% at 25 m s$^{-1}$ and

MVD = 20 µm (Langmuir and Blodgett, 1946). According to SAE ARP5905, the average diameter between non-iced and maximum iced cylinders for the calculation of the collection efficiency leads to an error of 1-2% in collection efficiency.

Two rotating cylinders were used for this testing with 2.5 mm (according to ARP5950) and 5 mm (for comparison) in diameter. The cylinders were rotated at 60 rpm. At the beginning of every run, the cylinder was shielded until the conditions had stabilized (approximately 15 s). All tests were performed at temperatures of - 18 °C or below to create rime ice, which is an essential requirement for this method (Ludlam, 1951). Differing from the previously mentioned systems, the RCT is an integrating and intrusive system. In this application, the MVD was taken from the PDI measurements and the LWC was measured by the RCT. In total, nearly 100 test points were done with 38 different spray settings.

The performed repeatability tests with the RCT lead to a coefficient of variation of $\sigma < 10\%$. Overall, SAE ARP5905 indicates because of the mentioned sources of errors a method accuracy of $> 90\%$, which can be verified by the repetition measurements carried out in this study.

### 3.5 LWC based on Water Flow Rate and Wind Tunnel Speed

The LWC in the icing wind tunnel can be determined from the total injected water mass flow and the circulating air volume flow (Biter, 1987). There are two prerequisites for the application of this procedure:

1) there is no recirculating water;

2) there is a known moist air volume flow (depending on flow velocity and droplet size).

The first assumption is true for an air temperature below 0 °C. In these conditions, the droplets will supercool and freeze out by hitting a surface of the wind tunnel, e.g. turning vanes of the first or second corner, collecting grid, fan or heat exchanger. To determine the moistened air volume flow in the wind tunnel, several icing tests on a grid were performed. The area over which the droplets spread depends on the wind tunnel speed and the droplet size or the air pressure used at the spray nozzles for droplet generation. Several tests were performed to measure the 2D-iced area in the test section and to estimate the LWC in the borders close to the wind tunnel walls. On the basis of these assumptions, the LWC can be calculated with the following formula:

$$LWC = \frac{\dot{m}_{Water}}{\dot{V}_{Air}} = \frac{\dot{m}_{Water}}{\alpha \cdot A_{testsection} \cdot u_\infty}, \tag{9}$$

where $\dot{m}_{Water}$ is the injected water mass flow, $\alpha$ is the percentage of the moistened cross-sectional area, $A_{testsection}$ the cross-sectional area, and $u_\infty$ the tunnel velocity. These numbers are available for all measurements and take also into account the clogging of nozzles during the experiment. Thereby, this method offers a good reference for LWC comparison.

The overall accuracy of the mass-flow-based-calculation of the LWC is primarily limited by the accuracy in the measurement of the water mass flow and the uncertainty in the determination of the moistened cross-sectional area. The water volume flow is measured with one thermal volumetric flow meter per row of six atomizers. Due to the very low total water volume flow

through every thermal volumetric flow meter (down to less than 10 ml min$^{-1}$ per row) and the pulsation of the nozzles, the uncertainty of the volume flow measurement is approximately 20%. The mean coefficient of variation of repeated test cases for the calculated LWC over the water mass flow and the moistened air volume is $\sigma = 7\%$.

## 4 Experimental Results and Discussion

The results of the intercomparison of the different measurement techniques are presented in this section. Tropea (2011) identifies three main sources of error in the measurement of size distributions with optical techniques, liquid fluxes, and droplet number concentration, which are inherent in all optical measurements conducted here: errors in droplet sizing, errors in counting (missed particles, coincidence) and errors in the sampling area (or volume) estimation.

The measurement uncertainties in droplet number concentration and sizing result in greater uncertainties for higher-order products such as LWC calculated from the observed cloud droplet size distribution (Lance et al. 2010).

### 4.1 Comparison of PSD and MVD measurements from the different instruments

The PSD obtained from the different measurement techniques is studied exemplary for two spray settings, resulting in a MVD of 14.5 µm and a larger MVD of 33.8 µm, see Fig. 9. PDI data evaluation was considered under application of two specific size ranges to enable adequate comparisons. Once for the comparison with the FCDP with a maximum considered droplet size of 50µm (PDI$_{FCDP}$) and once for the comparison with the shadowgraphy without a consideration of the smallest droplets (PDI$_{Shadowgraphy}$). Both reductions were made in a post processing step. The plot shows almost a similar trend for all the measurements despite their different acquisition rates, suggesting the acquisition time is sufficiently large for each of the methods. For the MVD 14.5 µm series, all measurement techniques show a mutual agreement in the distribution of normalized droplet counts. FCDP observations show slightly higher relative counts between 7 and 9 µm. Mode maximum of the PDI$_{FCDP}$ normalized droplet curve is found to be around 5µm and shifted towards smaller sizes, compared to the other techniques, but catches up with the curves for FCDP and shadowgraphy beyond 11 µm and 15 µm respectively. Sizing of smaller droplets with the FCDP can be subject to errors due to Mie ambiguity (Baumgardner et al. 2017). The droplet sizing from PDI is obtained by using linear relations between the phase shift and size derived for a predominant reflection or refraction mode and applying principles of geometrical optics (Ofner 2001). Below 5 µm, the validity of the geometric optics tends to cease and the diffraction becomes significant leading to erroneous measurements (Chuang et al., 2008). Bachalo and Sankar (1996) reported the uncertainties resulting from these oscillations to be under ±0.5 µm. Due to the resolution limit and the depth-of-field problem of the shadowgraphy technique, its PSD is shifted towards higher droplet sizes, ultimately distorting its cumulative liquid water content plot for larger droplet diameters. For the MVD 33.8 µm series, similar observations can be made. Noteworthily, the FCDP with its sizing limit of 50 µm does not allow to evaluate the upper end of the PSD, where shadowgraphy still enables optical accessibility. The representation of larger droplets is the lowest of the presented measurement techniques, in terms of droplet counts, whereas an abundance of observed droplets is visible between 7 and 9 µm.

A further analysis of the measurement techniques is based on the MVD as a scalar representation of the PSD, using again the PDI as a reference instrument, see Fig. 10. To compare FCDP and PDI results, the range of the PDI data evaluated for the intercomparison was limited to a maximum droplet diameter of 50 µm in a post-processing step to match the upper particle size limit of the FCDP.

The linear best fit ($MVD_{FCDP} = 0.91 \cdot MVD_{PDI}$) through the data points has a coefficient of determination of $R^2 = 0.9853$. The mean absolute value of the relative difference between the FCDP and the PDI measurements, according to Eq. (3), is $|E_{FCDP - PDI}| = 7.7\% \pm 3.9\%$. When comparing the relative deviations of the two instruments, for PSDs with MVDs < 20 µm the agreement between PDI and FCDP is nearly 100% and for distributions with MVDs > 20 µm the FCDP measures on average 9% lower MVDs compared to the PDI (see Fig. 11 left). Declining counts towards larger particle sizes (> 30 µm) may cause or contribute to the measured deviation of the FCDP with respect to the PDI for large droplets.

Confining the SA by application of a strict DoF criterion as a countermeasure in order to constrain coincidence in these high droplet number conditions might reduce the sample statistics for larger droplets and thus leading to an under representation of larger droplets contributing to the MVD in respective test points.

Measured droplet number concentrations up to 2000 cm$^{-3}$ from FCDP compared to PDI follow a linear distribution with a coefficient of determination of $R^2 = 0.9299$ and a tendency of observed higher $N_d$. The mean absolute value of the relative difference is 34% ±29% (Fig. 12). Data points beyond 2000 cm$^{-3}$ are scarce and deviate clearly from the afore mentioned trend for smaller $N_d$. This saturation effect, visible in the FCDP droplet number concentrations, might indicate the onset region of remaining coincident effects on particle counts to be located between 1500 cm$^{-3}$ and 2000 cm$^{-3}$ under consideration of the applied settings.

An additional source of error might be introduced via the external geometry of the probe and modified droplet trajectories which might alter the measured cloud PSD (Weigel et al. 2016); although this is accounted for to a certain extent by the aerodynamic shape of the FCDP, which differs from those in previous studies analyzed blunt geometry of classical PMS probes. Uncertainties due to aerodynamic effects still have to be considered while comparing measurements from the FCDP with non- intrusive techniques in the comparatively small test section of BIWT.

An evaluation of the implemented shattering filter in the post processing routines provided by SPEC, based on particle inter arrival time attributes shattering a negligible role. This may be due to the absence of very large droplets and ice particles, as well as the use of anti-shattering tips (Korolev et al., 2013, McFarquhar et al., 2007).

Ice accretion on the non-heated parts of the probe might additionally alter the local two-phase flow in the upstream direction (see Fig. 7).

The generally good agreement in MVD between FCDP and PDI in the size range of 8 to 35 µm with up to 14% deviation is well within the range of other instruments intercomparisons (Faber et al., 2018; Braga et al., 2017).

To ensure mutual size ranges between the PDI and the shadowgraphy system the minimum diameter of the PDI results was corrected to 10 µm in post-processing. The results of the shadowgraphy measurements are depicted in Fig. 10 as circles. Larger variations were detected by shadowgraphy for particle sizes larger than 35 µm.

The linear best fit ($MVD_{Shadowgraphy} = 0.97 \cdot MVD_{PDI}$) through the data points with a MVD < 35 µm has a coefficient of determination of $R^2 = 0.7985$ and is therefore smaller than the one from the FCDP data. The mean absolute value of the relative difference between the shadowgraphy and the PDI measurements, according to Eq. (3), is $|E_{Shadowgraphy - PDI}| = 9.9\% \pm 6.3\%$. The eight outlier with higher $MVD_{PDI}$ have not been taken into account for the best fit curve. An explanation for these measuring points with significantly smaller MVD again can be found in the typical drop size distribution: sample statistics suffer from a declining proportion of large droplets. (see Fig. 8). Rudoff et al. (1993) showed also in NASA's Glenn Research Center's Icing Research Tunnel (IRT) that the droplet distribution can have a long tail towards large droplets, which can only be detected reliably with exceptionally long measurement durations. With the shadowgraphy setup used here, only very low data rates could be measured. As a result, often only 3000-6000 droplets per spray condition were measured despite long test times (> 15 min) for one condition only. The droplet size distribution, however, can only be slightly corrected for large droplets, if at all, by application of a border correction. With the DoF and border correction, this leads to an average of more than 20000 droplets per distribution. The overall agreement between shadowgraphy and PDI results matches previous measurements, e.g. Kapulla et al. (2007) and Rydblom et al. (2019).

## 4.2 Comparison of LWC measurements from the different instruments

Fig. 13 shows the measurement results of the PDI, the FCDP, and the RCT compared to the LWC calculated from the injected water mass flow. Generally bulk phase instruments such as the rotating cylinder or a hotwire are used for the determination of the LWC. As expected, the comparison shows a significantly greater degree of variation compared to the droplet size results, which is discussed in more detail in the following.

The mean absolute value of the relative difference between the PDI results and the LWC calculation based on the water flow rate is $|E_{PDI - WFR}| = 24\% \pm 28\%$, according to Eq. (4). Despite the large absolute value of relative difference, the mean best fit line ($LWC_{PDI} = 0.98 \cdot LWC_{WFR}$ with coefficient of determination of $R^2 = 0.8503$) fits well to the results of the water mass flow method. From over 70 data points with $LWC_{WFR} < 0.5$ g m⁻³, 84% from the PDI results fall within a range of ± 0.1 g m⁻³ around the $LWC_{WFR}$ (71% of $LWC_{WFR} < 0.3$ g m⁻³ in the range ± 0.05 g m⁻³). Chuang et al. (2008) performed an intercomparison of the airborne PDI to a Gerber Scientific Inc. PVM-100A (a probe based on forward light scattering (Gerber et al., 1994)) and obtained a good consistency for LWC of up to 0.3 g m⁻³ with an accuracy of ± 0.05 g m⁻³ containing 85% of data points, which, despite the different velocities, is in good agreement with the results obtained here. Of the more than 100 remaining measurement results of the PDI with an LWC > 0.5 g m⁻³, only 57% lie within a range of ± 20% around the LWC calculated from the water mass flow. Cober et al. (2012) compare the integrated LWC from in situ measurements in supercooled large droplet conditions from FSSP and 2D-C and 2D-P to the results of the Nevzorov probe. A slightly higher LWC result from the integrating systems was found compared to the Nevzorov probe. From the measurements of Cober et al. (2012) with

LWC > 0.1 g m⁻³, 85% of the measurement points agree within ± 43% with that of the Nevzorov results. In our experiments,
90% of all PDI results with an LWC > 0.1 g m⁻³ agree within less than ± 43% deviation to the water flow rate. Therefore, our
results, which partly also contain droplets > 100 µm, are comparable to the results of the flight tests of Cober et al. (2012).

For a detailed analysis of the LWC results of the PDI, Fig. 14 shows the ratio of $LWC_{PDI}$ to $LWC_{WFR}$ over the MVD measured
by the PDI and over the air velocity in the wind tunnel. Plotting the LWC over the MVD indicates no clear tendency. Plotting
the LWC ratio against the wind tunnel velocity shows that at low velocities the PDI results are above the LWC calculated over
the mass flow and with increasing velocities the $LWC_{PDI}$ tends to become lower than the reference values. A similar result was
obtained by Rudoff et al. (1993) for the IRT. To see whether this dependency can be attributed more to the wind tunnel and
the water mass flow methodology or the PDI, the other measurement techniques are first examined in detail.

If also assuming an uncertainty of ± 20% for the PDI results, more than 85% of the LWC measurement data overlap between
the PDI and water mass flow. The comparison of the measurement results supports the already mentioned greater uncertainty
in the LWC measurements.

The results of the FCDP show a larger variation with respect to the reference. The linear best fit $LWC_{FCDP} = 1.12·LWC_{WFR}$
has a coefficient of determination of $R^2 = 0.3276$. Overall, there is a systematic high bias of the LWC derived from the FCDP
compared to the PDI, despite eventually smaller particle sizes detected by the FCDP. This can only be explained by higher
particle number concentrations measured by the FCDP compared to the PDI, as can be seen in Fig. 12. An overestimation of
the LWC by the use of scattering spectrometers has been found previously in comparative experiments (Rydblom et al., 2019;
Faber et al., 2018; Ide, 1999). For the FSSP forward scattering probe, Baumgardner (1983) found 20-200% higher LWC values
than measured by hot-wire probes. In the measurements by Ide (1999), the LWC calculated from the droplet diameter
distributions overestimated the LWC for MVDs up to 50 µm by 50% and even up to 100% and 150% for higher MVDs. Faber
et al. (2018) have suggested the velocity difference between his laboratory measurements and aircraft measurements, for which
the CDP is originally designed, as a possible reason for the large overestimation of LWC. This could also be a possible
explanation for the results obtained here. To examine the results in detail, Fig. 15 shows the ratio of $LWC_{FCDP}$ to $LWC_{WFR}$
versus $MVD_{FCDP}$ and versus $N_{d\ FCDP}$.

Unlike the PDI, a correlation between the FCDP data for LWC and droplet size seems to be obvious. Measurements with an
MVD > 27 µm are the only ones leading to an underestimated LWC. These measurement points also correspond to the data
points with low data rates and low particle concentrations (see the right section of Fig. 15). Due to the limited size range of
the FCDP and the broad width of the size bins, the underestimation of the LWC can be caused by some of the droplets present
in the flow but not visible for the FCDP. At high $N_d$ and small droplet diameters, the FCDP significantly overestimates the
LWC. The dependence of LWC on droplet concentration was also observed by Lance et al. (2010) in observations during the
ARCPAC campaign. Also a larger contribution of small droplets to an LWC overestimation bias is confirmed from simulations
(Lance et al. (2010)). Higher droplet number concentration exhibits higher coincidence effects and lead to overestimated
particle sizes. However, Fig. 10 clearly shows an agreement of 7% within the probes and eventually a low bias of the MVD
detected with the FCDP.

The results of the RCT are illustrated by blue circles in Fig. 13. The mean absolute value of the relative difference between the rotating cylinder and the LWC calculation based on the water flow rate is $|E_{rotCyl - WFR}| = 22.9\% \pm 21.3\%$ and is therefore the smallest among the presented LWC measurements. The linear best fit ($LWC_{rotCyl} = 0.98 \cdot LWC_{WFR}$) over the data points has a coefficient of determination of $R^2 = 0.9066$. Taking into account an uncertainty of $\pm 10\%$ in the measurement results of the rotating cylinder, 78% of the measurement points are within the range of the expected value regarding to water mass flow. Cober et al. (2001) compared the integrated LWC of the droplet sizing probes to the measurement results from the Rosemount icing (ice-accretion-based) detector, where 90% of the data fall within the 1:1 correlation $\pm 64\%$. The large scatter of these data is similar to the measurements described here, although the comparison technique is different.

Fig. 14 shows that both with increasing MVD and velocity the LWC tends to be slightly overestimated by the RCT. Ide (1999) found in his measurements a good agreement between the icing blade, the RCT, and two hot-wire-probes for small droplets (MVD < 40 µm). This outcome can be supported by the results presented in this study. When compared to the PDI, the RCT behaves in the exact opposite way: the LWC from the PDI measurements tend to decrease with increasing velocity, whereas the LWC from the RCT is increasing with velocity. This contrary behaviour of the two different measurement techniques calls rather not for a cause in the methodology of the water mass flow but causes in the individual measurement techniques.

## 5 Summary and Outlook

The BIWT has been further developed to produce liquid droplets in the size range of 1 to 150 µm at LWC ranges of 0.1 to 2.5 g m$^{-3}$. The droplets were accelerated to velocities between 10 and 40 m s$^{-1}$ and supercooled to temperatures between 0 and -20 °C. Measurements with the PDI show that the icing wind tunnel–exhibits a good repeatability of the MVD with a stability better than 3% and the LWC to be better than 7%, as derived by standard variation. These test conditions permit very high reliability and stability appropriate to intercompare various droplet measuring techniques.

A probe intercomparison study of droplet size (PDI, FCDP, and shadowgraphy) and LWC (PDI, FCDP, and RCT) measurement systems was performed. Generally, the MVD measured with the FCDP agreed within 15% to measurements with the PDI, which is in the range or better than previous tests in wind tunnels. The MVD of the shadowgraphy agreed up to 35 µm well to the PDI, beyond MVD 35 µm a higher discrepancy was observed. By comparing the droplet size measurement techniques, it was possible to identify some measurement system-dependent sources of uncertainties. For the FCDP, the high sensitivity of the transit time filter to velocity differences of the droplets or a respective low sensitivity to larger particle sizes (>35 µm) was hypothesized. Our results with the shadowgraphy setup also show the importance of the upper part of the droplet size distribution, where the occurrence of larger droplets declines. The fraction of large droplets has a huge impact on characteristic quantities such as the MVD and therefore requires a high number of sampled droplets per measurement point.

In addition, LWC measurements were compared to the LWC calculated from wind tunnel input parameters and the flow rate. Here, besides the rotating cylinder bulk phase instrument, the LWC was also derived from the PSD measured with the single particle probes, albeit with larger uncertainty. 57% (59%) of the LWC results measured with the RCT (PDI) agreed within

615 20% with the LWC determined based on the water mass flow. This is also a good overall agreement compared to existing tests. Several technology-dependent differences and error sources were identified for the LWC measurements. The PDI results showed a slight overestimation of the LWC with decreasing flow velocity. The RCT results showed very good agreement to the LWC results based on water mass flow, especially for small droplet sizes, concurring well with literature studies. The FCDP results differ significantly (factor of 0.5 to 3) from the water mass flow results.

Based on these new results on the performance of the BIWT for unimodal droplet distributions and related strength and shortcomings of instruments and measurement systems detecting PSDs and LWCs, future plans are to further enhance the capacity of the BIWT's spray system to generate bimodal droplet size distributions according to EASA CS 25 Appendix O. These distributions incorporate one collective of small droplets (around 11-14 µm) and a second collective of very large droplets (around 160-200 µm), while requiring a low LWC between 0.1 and 0.45 g m⁻³. The reliable acquisition of both modes

with the associated low number density of the large droplets > 100 µm poses new challenges for droplet measurement techniques. The detection range has to be extended and the trajectory of large droplets and their sedimentation velocity has to be considered in the wind tunnel design and probe layout in order to accurately provide and measure a large particle spectrum. Existing knowledge in ice crystal icing experiments (e.g. Bansmer et al., 2018) can support these developments.

**Data availability**

The test matrix as supplement material related to this article is available online (doi: ). The detailed data sets used in this study are available from the corresponding author upon request (i.knop@tu-braunschweig.de).

**Author contribution**

Inken Knop designed and carried out most of the described experiments. The FCDP measurements were designed and carried out by Valerian Hahn, supported by Inken Knop. Stephan Bansmer and Christiane Voigt supported the post processing and evaluation of the experiments, mainly conducted by Inken Knop (and Valerian Hahn for the FCDP). Inken Knop prepared the manuscript with the contributions from all co-authors. The authors would like to thank Venkatesh Bora in supporting the review process of the paper.

**Competing interests**

The authors declare that they have no conflict of interest.

*Acknowledgements*. The presented work was conducted within the framework of the project Drífa -FKZ0325842A funded by the German Federal Ministry of Economic Affairs and Energy. Christiane Voigt and Valerian Hahn were funded by the European Union H2020 programme ICE GENESIS under contract number 824310 and by the DFG within SPP2115 PROM under contract number VO1504/5-1. The PDI measurements were strongly supported by Dr. Thomas Brämer and David Apel

from LaVision GmbH, Göttingen, Germany. The authors further express their thanks to Biagio Esposito (CIRA) and Will Bachalo (Artium) for their help in interpreting the PDI results and Stephan Sattler (TU Braunschweig) and Oliver Esselmann (former TU Braunschweig) in conducting the experiments. Finally, the authors acknowledge the support of the Open Access Publication Funds of the Technische Universität Braunschweig.

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

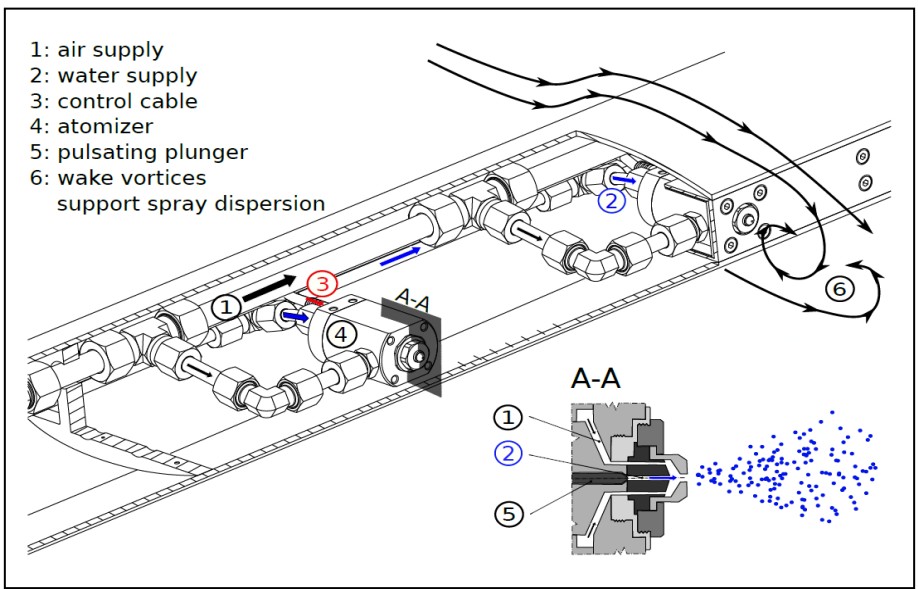

Figure 1: Spray system in the Braunschweig Icing Wind Tunnel (Bansmer et al. 2018).

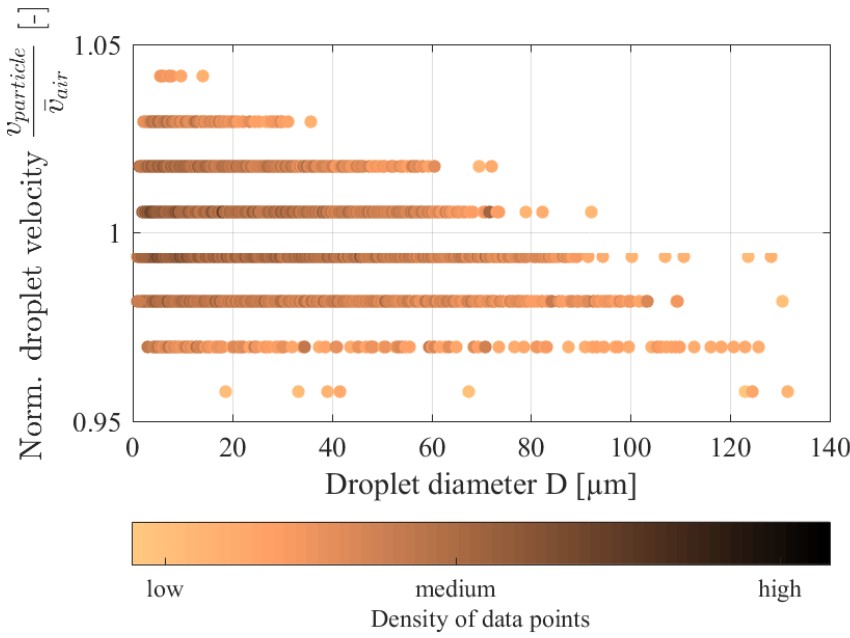

Figure 2: Droplet velocity over diameter (PDI Run 08/08/17 16:25).

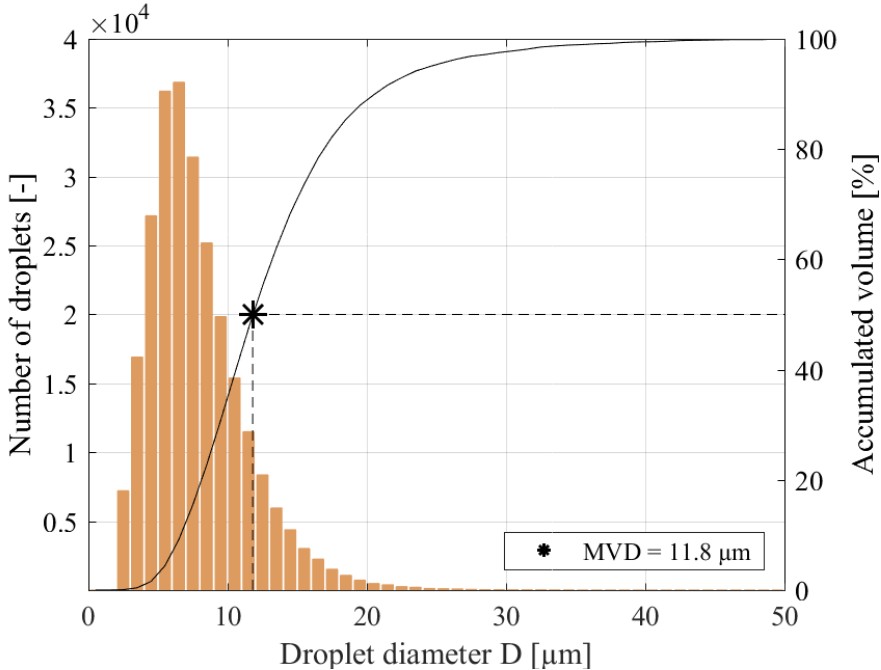

865 Figure 3: Droplet diameter histogram and cumulative volume curve at 20 m s[-1] (PDI Run 18/04/19 17:19).

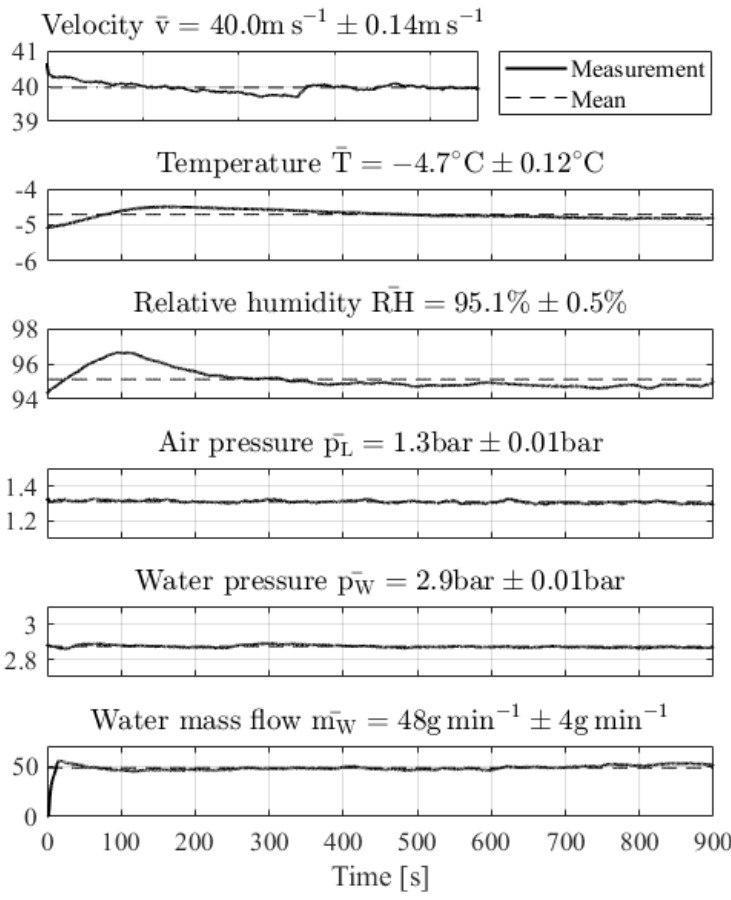

Figure 4: Exemplary stability of the wind tunnel conditions over 15 minutes test duration.


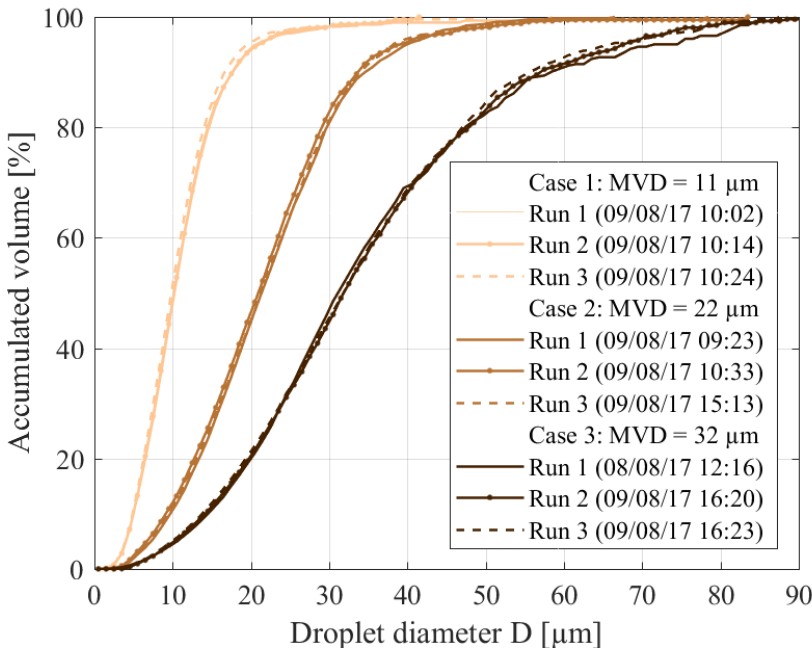

Figure 5: Wind tunnel repeatability shown with PDI measurements at 40 m s$^{-1}$, correlation coefficients R²: Case 1: Run 1-2: R²=1; Run 1-3: R²=0,999; Run 2-3: R²=0,999; Case 2: Run 1-2: R²=0,926; Run 1-3: R²=0,927; Run 2-3: R²=1; Case 3: Run 1-2: R²=0,999; Run 1-3: R²=1; Run 2-3: R²=1.

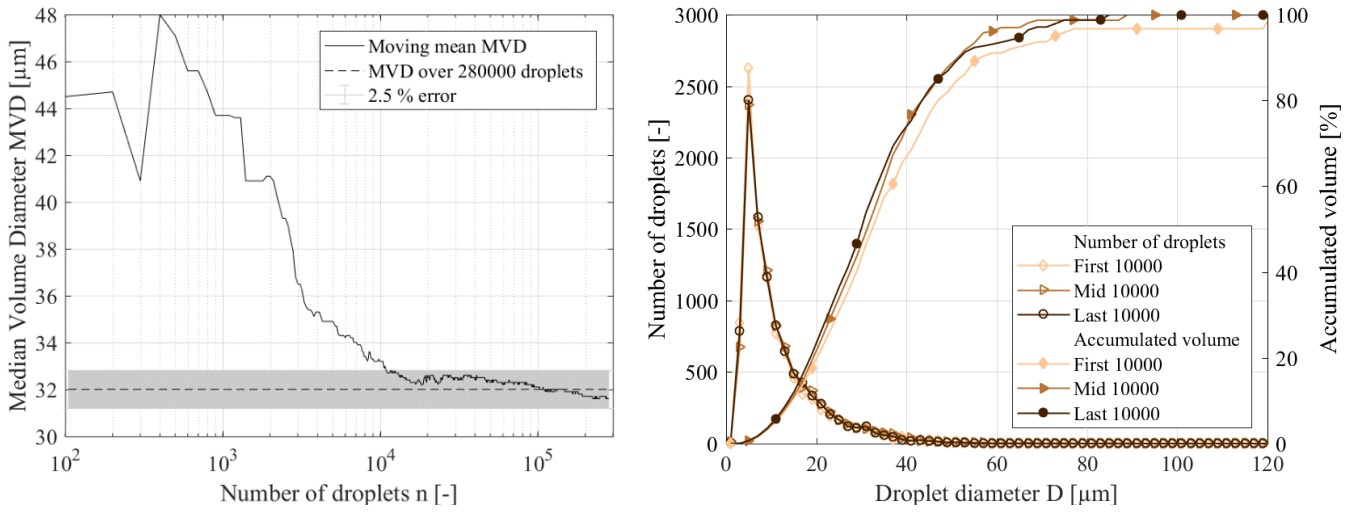

Figure 6: Convergence of MVD over number of droplets at 40 m s$^{-1}$ and Temporal Stability (PDI Run 09/08/17 17:59).

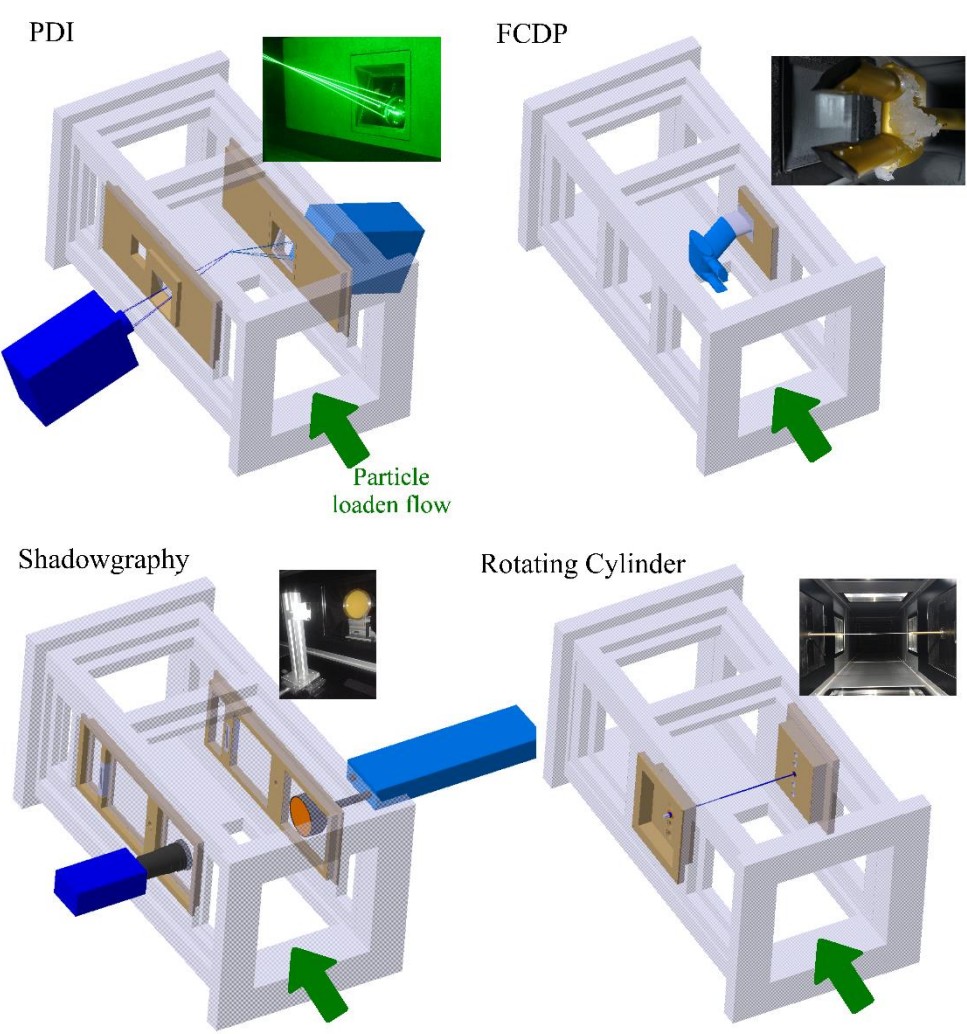

Figure 7: Measurement setups in the Braunschweig Icing Wind Tunnel.


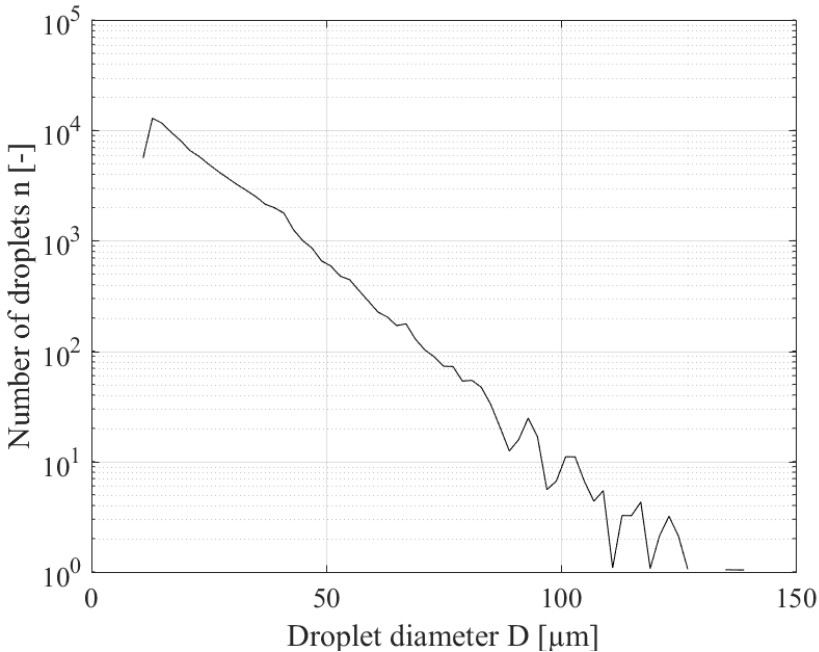

Figure 8: Number of droplets over diameter (PDI Run 18/04/19 18:25), total number of counts >95000.

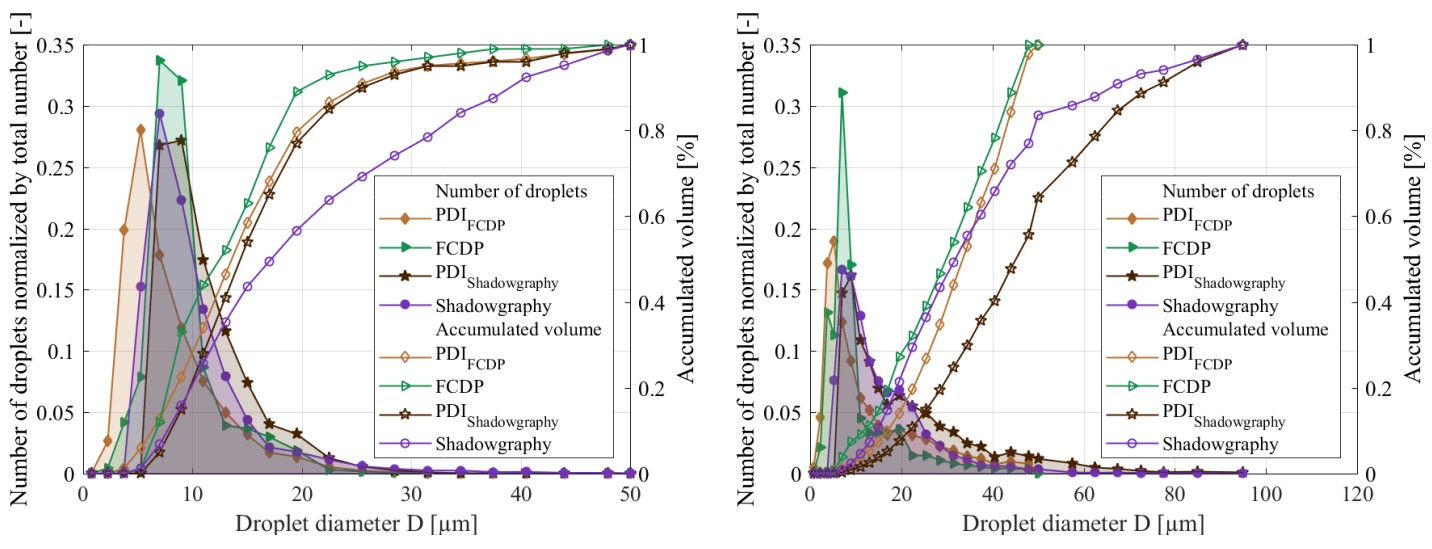


Figure 9: Droplet size distribution of different methods (left MVD 14.5 µm, right MVD 33.8 µm)

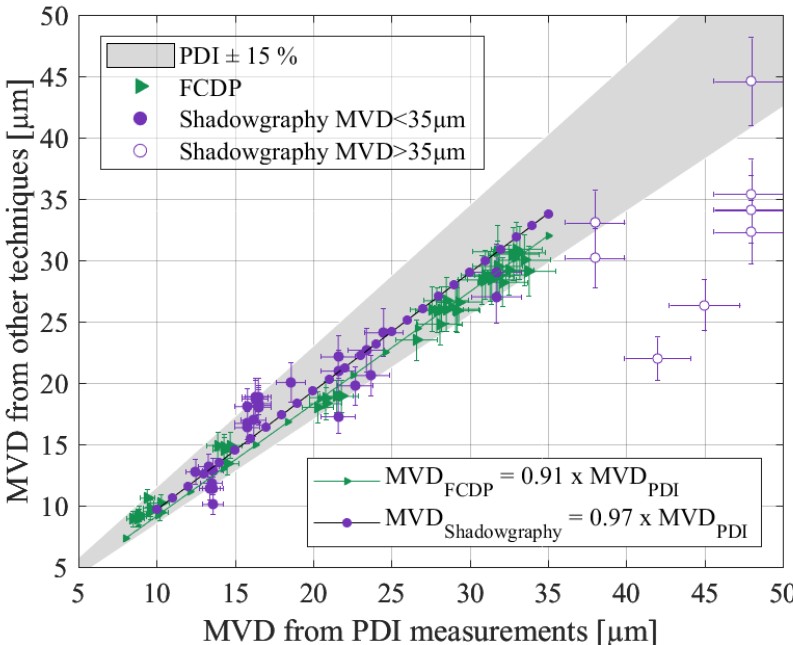

Figure 10: Intercomparison of MVD measured with the PDI, the FCDP and the shadowgraphy.


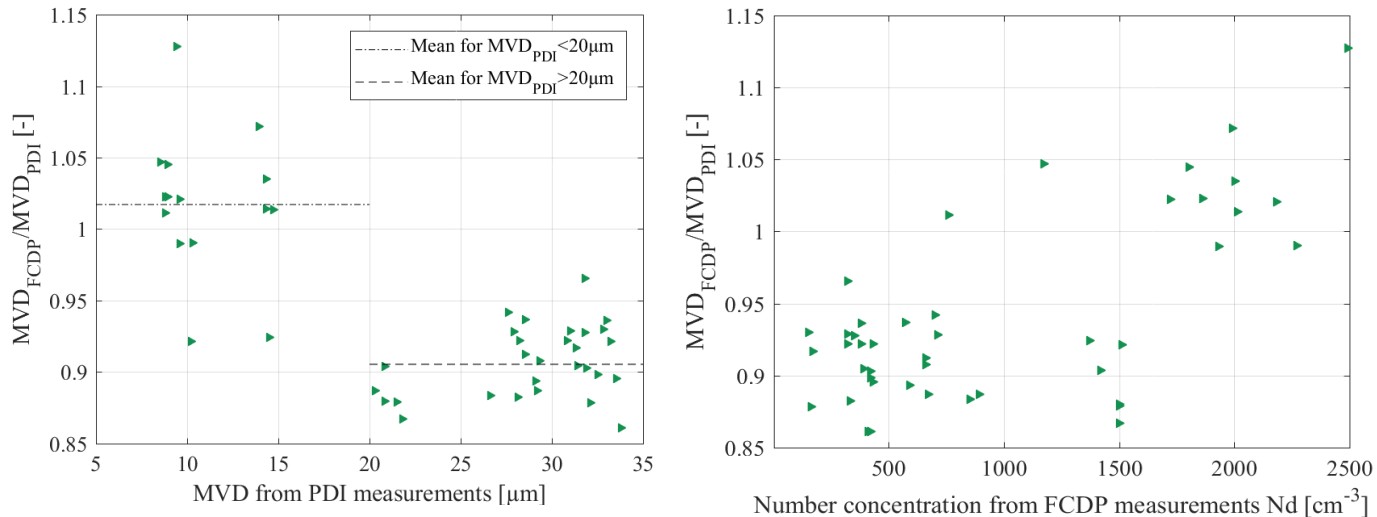

Figure 11: Effect of MVD on droplet size measurements from the FCDP (left) and effect of number concentration (right).

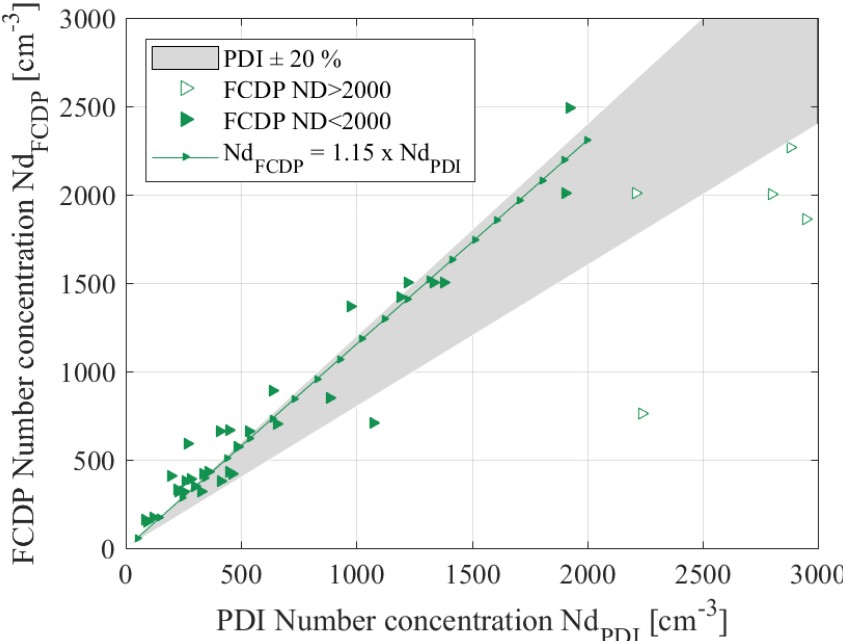

Figure 12: Comparison of Number Concentrations of PDI and FCDP.

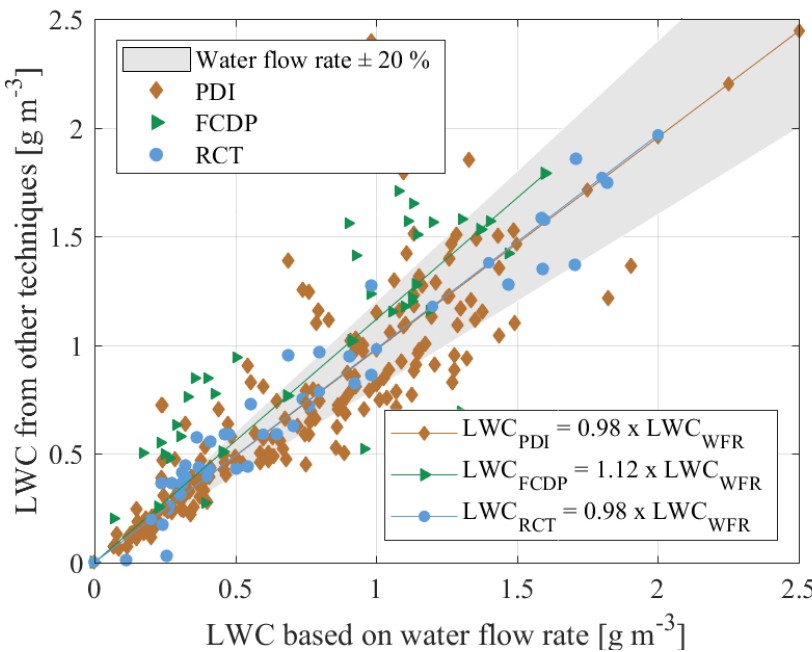

Figure 13: Intercomparison of LWC based on the water flow rate and measured with the PDI, the FCDP and the RCT.

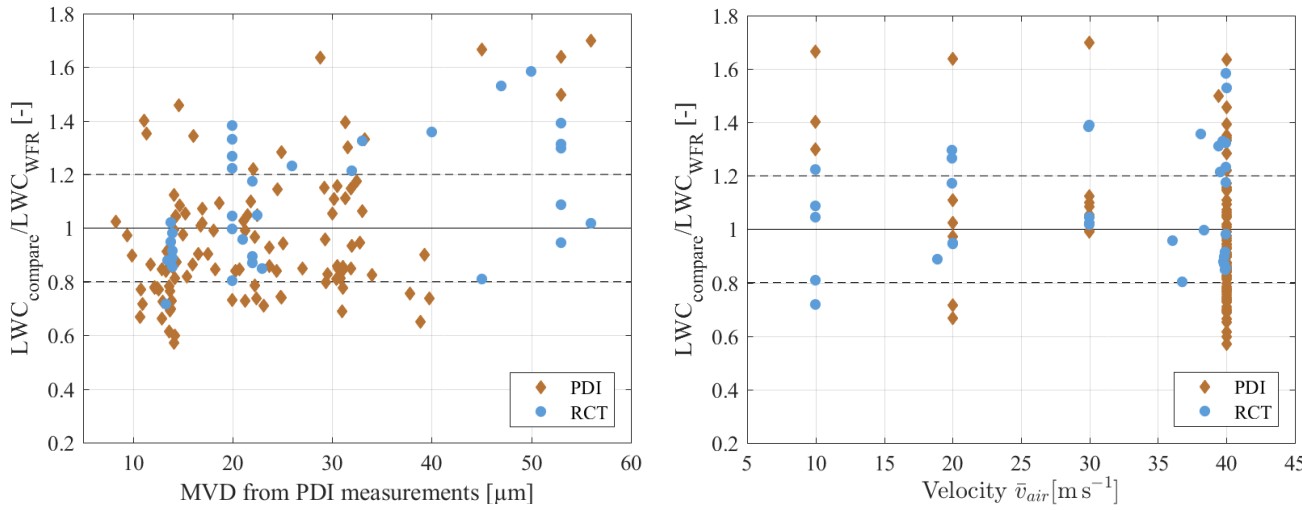


Figure 14: Effect of MVD (left) and effects of air velocity (right) on LWC measurements from the PDI and the RCT.

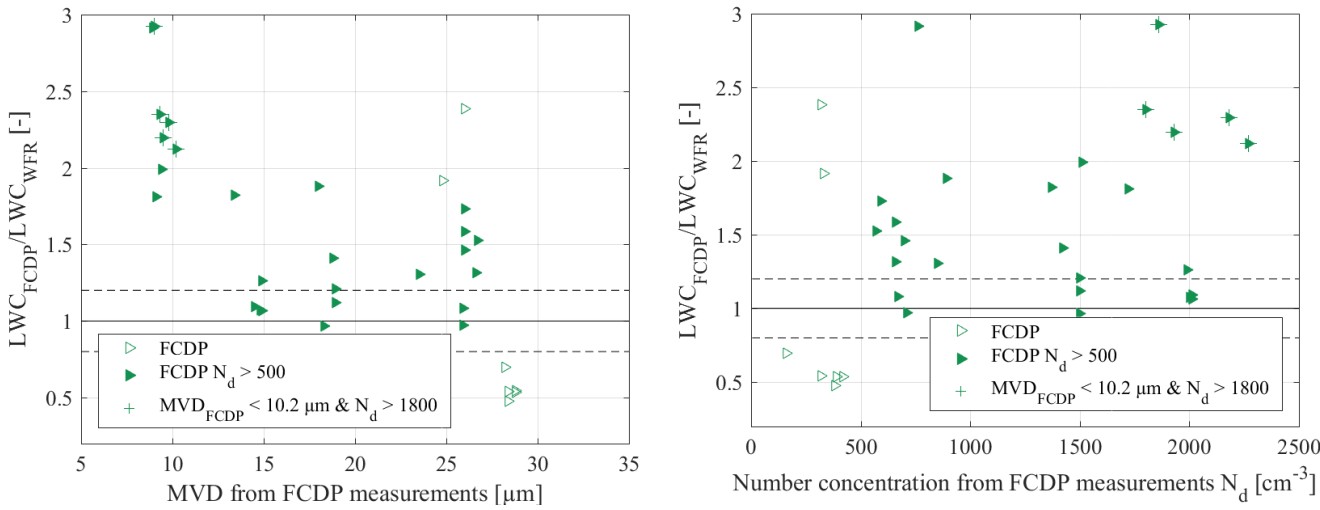

Figure15: Effect of MVD on LWC measurements from the FCDP (left) and effect of number concentration (right).


Table 1: Characteristic numbers of PDI setup Artium PDI-x00MD

| Transmitter | | Receiver | |
|---|---|---|---|
| Wave-length | 532 nm | Focal length | 500mm |
| Focal length | 350 mm/ 500 mm | Collection Angle | 40 ° ± 1° |
| Beam Separation | 59.4 mm | Slit Aperture | 100 µm |
| Beam diameter | 2.33 mm | PMT Gain | 300-500 V |
| Expander Factor | 1 | Domination scattering order | refraction |
| Frequency Shift | 40 MHz | | |
| Fringe Spacing | 3.1 µm/ 4.5 µm | Static Range | 0.9 - 134.4 µm 1.3 - 191.7 µm (2.6 -571.2 µm) |
| Beam Waist at probe volume | 101.7 µm/ 145.4 µm | | |

Table 2: Characteristic numbers of FCDP (Serial No. 6) setup, SA and size calibration as of 4/28/2017

| Wavelength | 785 nm | DoF crit. | 0.9 |
|---|---|---|---|
| Domination scattering order | Forward Scattering | Bin number | 21 |
| Collection Angle | 4-12 ° | Bin widths | 1.5-4 µm |
| Beam width diameter | 0.08 cm | Size Range | 1.5-50 µm |
| Qualifier Slit width | 0.009 cm | Beam Waist | 80 µm |
| DoF Rejection Crit. | 0.9 | Sample Area | 0.09 mm² |

Table 3: Characteristic numbers of shadowgraphy setup

| Laser | Pulsed Nd-YAG laser | Camera | PCO Sensicam 12bit |
|---|---|---|---|
| Energy | 1200 mJ | Resolution | 1376 x 1070 px |
| Pulse duration | 4 ns | Scale | 1.9 x 1.9 µm ≙ 1 Pixel |
| Objective focus | 180 mm, 1:1 macro | Tele convertor lens | 1.4X |
| Aperture | 3.5-32 | | |


Table 4: Summarized instrument and test conditions

| | | | | PDI | FCDP | shadow-graphy | RCT |
|---|---|---|---|---|---|---|---|
| **instrument measurement range** | **velocity** | m s$^{-1}$ | min | -130 | 10 | x | 1 |
| | | | max | 500 | 200 | x | >175 |
| | **droplet diameter** | µm | min | 1 | 2 | 10 | x |
| | | | max | 134 | 50 | 200 | x |
| | **data rate / image rate** | Hz | min | 0 | x | 1 | x |
| | | | max | >100000 | x | 2 | x |
| | **LWC** | g cm$^{-3}$ | min | x | 0 | x | 0 |
| | | | max | x | *2 | x | 1,9*3 |
| **tested range** | **velocity** | m s$^{-1}$ | min | 10 | 30 | 10 | 10 |
| | | | max | 40 | 40 | 40 | 40 |
| | **MVD** | µm | min | 8,3 | 8,9 | 10,1 | x |
| | | | max | 56 | 30,9 | 44,6 | x |
| | **Coefficient of variation MVD*1** | % | | 5 | 7 | 8 | x |
| | **LWC** | g cm$^{-3}$ | min | 0,062 | 0,204 | x | 0,013 |
| | | | max | 2,434 | 1,707 | x | 1,858 |
| | **Coefficient of variation LWC*1** | % | | 20 | 16 | x | 8 |
| | **number density** | cm$^{-3}$ | min | 82 | 150 | x | x |
| | | | max | 3008 | 2270 | x | x |

*1 determined precision of measurement setup in BIWT, Eq. (2)
*2 subject to coincidence
*3 Ludlam-limit at 40 m s$^{-1}$