# Peer review of "Comparison of different droplet measurement techniques in the Braunschweig Icing Wind Tunnel"

_Atmospheric Measurement Techniques, 2019_

## Referee Comment (RC1) · Anonymous Referee #1 · 10 Apr 2020

**General Comments**:

This manuscript describes an icing tunnel test facility that may be of interest to the atmospheric science community. The fundamental operation of this system is similar to existing icing tunnels, however, there is no comparison to icing tunnels that have been in existence for nearly 50 years. Of particular interest is the icing tunnel in Ottawa, Canada that is operated by the National Research Council. A description of the NRC icing tunnel can be found at https://nrc.canada.ca/en/research-development/nrc-facilities/altitude-icing-wind-tunnel-research-facility, and early papers describing experiments in the tunnel are by Strapp and Schemenauer (1982) and King et al. (1985). The NRC icing tunnel has unique advantages over the Braunschweig tunnel, specifically the NRC tunnel is capable of particle speeds up to 100 m s$^{-1}$ and altitudes to 40,000 ft (12 km). The Braunschweig maximum particle speed of the Braunschweig tunnel is 40 m s$^{-1}$ and it has not capacity to simulate altitude. This is unfortunate since research aircraft fly at various altitudes and all large aircraft fly at speeds that are at least twice the maximum speed that the Braunschweig tunnel can produce. The manuscript needs to discuss how the limitations of the Braunschweig tunnel influence their results.

Understanding discrepancies between drop concentrations and drop size distributions (DSDs) measured by various probes is of critical importance for cloud physics, albeit icing studies rely more on bulk quantities such as MVD and LWC. MVD and LWC are presented and discussed in great detail, but there is almost no quantitative discussion of drop concentrations and DSDs from the PDI, FCDP and Shadowgraphy instrumentation. The manuscript needs to include additional figures that show correlations between reported drop concentrations and DSDs measured by PDI, FCDP and Shadowgraphy.

The PDI measures LWC using eqn. 4, which proportional to the product of total drop concentration and corrected volume mean diameter. The manuscript should also show how LWC compares using this technique with LWC computed by integrating the complete DSD. The comparison should be shown as a function of MVD and drop concentration.

The manuscript compares LWC measurements from the FCDP, PDI and RCT using the WFR as a standard. It is implied that LWC using WFR as a standard is very repeatable, on the order of 7%. In Section 4 a statement refers to Section 3 as justification for this, but far as I can tell in Section 3, this repeatability comes from the literature, not from actual tunnel tests. Yet, I assume there were LWC repeatability tests, similar to the MVD tests shown in Fig. 5, so please point out where I missed the LWC repeatability tests or include figures showing results from them. Now, that said, based on the large amount of scatter shown in Figs. 12 – 14, either the tunnel flow characteristics or the measuring techniques, or both, appear to be contributing much more variability to LWC than the 20% figure quoted in the text. It would be useful to show a complete uncertainty analysis for the WFR and test instruments measurements, but this is likely to be outside the scope of this paper. However, the manuscript should address tunnel and instrument LWC uncertainties in a more rigorous manner, not just quote the literature.

The sample conditions of these tests (droplet concentrations sometimes exceeding 2000 cm$^{-3}$) are typically only found in polluted environments. This, plus the slow droplet speed, limit the usefulness of the results of these experiments. These limitations need to be discussed in detail in the manuscript.

The poor sampling statistics for drop diameters > 30 microns (Figs 4 and 7) definitely introduce uncertainties in the LWC results that need to be addressed more rigorously.

Finally, it should be pointed out that without a rigorous uncertainty analysis of the absolute accuracy of the tunnel, all of the quoted accuracies are not absolute, but instead relative. That is, in addition to the random error associated with tunnel properties, there is some degree of undetermined bias error that is not considered. This needs to be emphasized in the manuscript, albeit, hints of this are included in some of the references cited.

**Specific Comments**:

1. Introduction

Page 2: When mentioning the cloud probes used by Ide (1999) and Cober et al. (2001), the manuscript should describe the resolution and size range of these probes so results can be compared with tunnel results.

2. Experimental Setup

Add a table (perhaps as a supplement) indicating the mean operating conditions for all of the data sets presented in this manuscript (Velocity, Temperature, RH, Air Pressure, Water Pressure, Water mass flow). This would be helpful to understand the scope of conditions for each type of drop measurement system. Also, list the number of data sets collected for each drop measurement technique (e.g., FCDP: 100 samples at 20 m s$^{-1}$, 200 samples at 30 m s$^{-1}$, 300 samples at 40 m s$^{-1}$). Based on the data plotted in Fig. 11, it appears that the PDI datasets greatly exceed those of any other probe. This could be due to the high particle rejection rate and very low sample volume of the PDI. Please explain.

3.1. PDI

In this section it is noted that "The PVC has the greatest effect on the smallest size classes. Their influence on the LWC, on the other hand, is very small." We have performed an independent analysis of the FCDP data set from this icing tunnel and determined that the peak of the particle mass size distribution is between 8 um and 10 um. Therefore in this study the vast majority of drops are very small (see PDI PSD in Figure 4 for confirmation where the mode is ~6 um). Given these concerns, the manuscript should include more details of the PDI small drop corrections and better quantify the errors. From Chuang et al 2008: "At very small droplet sizes, diffraction can become significant relative to refraction, and lead to oscillations in the φ versus d relationship at the smallest drop sizes, primarily in the size range below 4 µm, but with some effects up to ~8 µm."

3.2. FCDP

The description of the FCDP sample volume is fairly convoluted. It is sufficiently described as SV=SA*TAS, where the SA is defined by calibration for a fixed qualification criteria. The SA is defined by laboratory calibration like that described in Faber et al. (2018). See the Table's section of the review for more details.

In paragraph 3 it should be noted that the CDP and FCDP have similar operating principals, but the improved optics and electronics in FCDP allow for accurate sampling in higher particle concentrations (see comments on Table 2 below). The FCDP also differs from the older FSSP-100 probe in that the qualifier detector uses a slit aperture (200 µm x 800 µm), which was first introduced on the FSSP-300 probe with data described by Brenguier et al. (1998).

Lance et al. (2010) note that accurate sizing of the CDP instrument to ~200 cm$^{-3}$ before being influenced by coincidence. However, improvements in the CDP (new limiting apertures) increased accuracy such that only 27% undercounting is estimated at concentrations of 500 cm$^{-3}$ (Lance 2012). This level of uncertainty is still problematic. The FCDP was designed to incorporate the improvements of the CDP as well as reduce particle coincidence (the dominant source of error) by reducing the laser beam waist from 200 µm to 80 µm). Flight tests indicate reasonable agreement for LWC between the FCDP and hotwire probes for small droplet concentrations as high as ~1000 cm$^{-3}$. The conditions in this study exceed these typical atmospheric conditions, so the FCDP uncertainty for these high drop concentrations range is not well described.

In paragraph 3, the authors cite up to a 50% uncertainty from Baumgardner 2017, but it should be noted that this quoted uncertainty (10 to 50% for light scattering probes), includes "Mie ambiguity, collection angles, coincidence, nonsphericity and shattering." In this study all droplets are assumed to be spherical, and shattering is minimal for the FCDP, so the 50% uncertainty does not apply here.

**3.3. Shadowgraphy**

Overall the Shadowgraphy technique is poorly described. More details of the instrument and post-processing are required such that the test could be replicated and verified by another group. In the last sentence from this section the data inter comparison is considered" almost identical." This statement requires quantification.

**3.4. Rotating Cylinder Technique**

Stallabrass (1987) should be Stallabrass (1978)

**4.1. Repeatability**
See General Comments.

Paragraph 2: "see section 4.1" within section 4.1, should be "see section 3.1".

Paragraph 4: "see section 4.2" within section 4.1, should be "see section 3.2".

**4.2.  Comparison of MVD measurements**

Paragraph 3: "A low sensitivity of the FCDP to larger particle sizes (> 30 μm) may cause or contribute to the measured deviation of the FCDP with respect to the PDI for large droplets."

Is there evidence that the FCDP has a low sensitivity to larger drops? If so, provide a reference. The number of sampled drops is relatively low, due to the small sample volume and low concentration of larger drops, but this is true for all single-particle devices, including the PVI. Also, with long runs in an icing tunnel this should not be an issue, assuming the tunnel properties are repeatable, as claimed in Section 4.1.

Paragraph 3: "The transit time filter applied to the FCDP data during post-processing to reduce coincidence causes a rejection of droplets that have a too long transit time compared to the mean reverence [sic] velocity and thus reduces the droplet size spectrum evaluated as valid by large droplets."

Is there any evidence to support this assertion? If so, please provide the evidence, a reference, or sound physical explanation.

Paragraph 5: "According to Lance et al. (2010), an additional source of error of the CDP might be the external geometry of the probe, which can alter the measured cloud particle size distribution."

This statement does not apply to the FCDP because it has "anti-shattering tips" that minimize droplet splashing, whereas the CDP Lance used did not (at that time) have anti-shattering tips. The CDP can be equipped with anti-shattering tips now.

**4.3.  Comparison of LWC measurements**

See General Comments, also:

Paragraph 6: "This can only be explained by higher particle number concentrations measured by the FCDP compared to the PDI."

Please provide particle concentrations and size distributions for the FCDP and PDI for the relevant wind tunnel datasets. Following the example in Figure 7, the authors should indicate when the PDI and FCDP sampling statistics are poor (<100 particles per bin) and possibly remove these data from consideration.

5.0. Summary

Page 17 Paragraph 2: "The characterization of cloud droplet distributions with particle sizes > 100 μm poses new challenges for droplet measurement techniques." Some Optical Array Probes are well suited to particle measurement in this range, specifically the 2D-S, which is commonly utilized for icing tunnel measurements of larger drops.

References

Biter, C. J., et al., 1987: The drop-size response of the CSIRO liquid water probe. *J. Atmos. Oceanic Technol.* **4**, 359-367.

Brenguier, Jean-Louis, et al., 1998: Improvements of droplet size distribution measurements with the Fast-FSSP (Forward Scattering Spectrometer Probe). *J. Atmos. Oceanic Technol.,* **15**, 1077-1090.

King, W. D., et al., 1985: Icing wind tunnel tests on the CSIRO liquid water probe. *J. Atmos. Oceanic Technol.,* **2**, 340-352.

Korolev, A., et al., 2013: Modification and tests of particle probe tips to mitigate effects of ice shattering. *J. Atmos. Oceanic Technol.,* **30**, 690-708.

Lance, S., 2012: Coincidence errors in a cloud droplet probe (CDP) and a cloud and aerosol spectrometer (CAS), and the improved performance of a modified CDP. *J. Atmos. Oceanic Technol.,* **29**, 1532-1541.

Strapp, J. W. and R.S. Schemenauer, 1982: Calibrations of Johnson-Williams Liquid Water Content Meters in a High-Speed Icing Tunnel. *J. Appl. Meteor.,* **21**, 98–108, https://doi.org/10.1175/1520-0450(1982)021<0098:COJWLW>2.0.CO;2

**Figures**

Figure 2: Add mean and variance values to each plot. It would also be helpful to add time-series data for the PDI, FCDP and Shadowgraphy (counts/sec or conc/sec). Perhaps it would be better to add the probes' time-series as a new figure.

Figure 4: Add additional accumulated size distributions for the FCDP and Shadowgraphy. If possible, overlay distributions from all three methods on the same figure indicating Concentration, LWC and MVD for each.

Figure 5: Add $R^2$ values.

Figure 6: Based on the results in Figure 2, the conditions (Temp, RH) are not necessarily stable for the initial portion of the sample run. As such, it is hard to separate the MVD discrepancy as a function of true fluctuations vs counting statistics. Show size distributions for the initial $10^3$ droplets and the final $10^3$ droplets.

Figure 7: This plot is very useful, but as with Figure 4 it should be amended to include lines for FCDP and Shadowgraphy. It may also be helpful to interpolate the higher resolution PDI data into the FCDP size bins for

a more accurate comparison.  Note that the FCDP size bins are chosen to smooth out Mie bumps and to improve sampling statistics for the larges drops.  Add a dashed line to indicate the threshold for rejecting data due to inadequate data points in a bin (e.g., 100 counts bin$^{-1}$).

Figure 9:  These plots are useful, but additional plots should be added to show correlation with LWC and Concentration.

Figure 11:  It hard to visually separate the WFR grey region from the PDI data points.  Considering switching this plot to color or making individual scatter plots for PDI, FCDP and RCT.

Figure 14: Combine with Figure 12, and include sample plots for FCDP.

**Tables**

Table 1:  Include the model number of the PDI in the caption.

Table 2: Amend the table to include these values:

FCDP Beam Waist = 80µm
FCDP DOF Rejection Criteria = 0.9
FCDP Sample Area = 0.09mm^2
FCDP Size Range = 2-50µm
FCDP Serial number = 6
FCDP Calibration Date = 4/28/2017 (see sizing calibration curve below from manufacturer)

[Figure]

Table 3: Include details on the Shadowgraphy optical system similar to Table 1 for the PDI (wavelength, magnification, focal length, working distance, collection angle, etc.).

Table 4:  Define the table column variables in the caption.

---

## Referee Comment (RC2) · Anonymous Referee #2 · 29 Apr 2020

**General comments:**

This technical paper reports on a wind tunnel experiment designed to compare several droplet measurement techniques. The experiment is conducted in the Braunschweig Icing Wind Tunnel where populations of supercooled droplets with size ranging from 1 to 150 μm are generated. The analysis focuses on Median Volumetric Diameter and Liquid Water Content, two key microphysical properties in the characterization of icing conditions.

The droplet measurement techniques involved, namely Phase Doppler Interferometry, shadowgraphy and FCDP, a commercial single particle counter, are commonly used by cloud physics and icing research groups. Nevertheless, there are still gaps in the understanding on their respective performances which is detrimental to the comparison of data produced by various research groups using different instruments. According to the authors, wind tunnel experiments offer a unique opportunity to test droplet measurement techniques in controlled and repeatable test conditions which in the end contributes to the definition of measurement standards. Thus, the present study may bring some valuable contributions to the field and deserve a publication in AMT journal.

However, as highlighted in Kapulla et al. 2007, a thorough interpretation of the experimental results is necessary to draw a fair comparison between techniques based on different sizing and counting principles. In my opinion, this paper needs further elaboration regarding the presentation of the test conditions and the analysis of the data:

- The article does not contain a test matrix summarizing the experiment and providing the following information: wind tunnel settings (temperature, LWC, airspeed …), number of runs for each test conditions, duration and number of points collected by each probe in each run. The information scattered in the article indicates that several directions have been investigated (e.g. measurements in various wind tunnel conditions or influence of some probe settings) and that the analysis is based on a substantial number of data points, but it is hard to identify clearly the scope of this experiment and the statistical soundness of its results.

- MVD and LWC are inferred from particle size distributions (PSD) in all but two cases (rotating cylinder and tunnel air and water flow supply system settings). Given the importance of the measured PSD, the analysis given in section 4 should include a discussion on the PSD measured by the aforementioned techniques in the same test conditions. This would provide a solid basis for the subsequent interpretation of MVD and LWC results.

I strongly encourage the authors to deepen the analysis in order to strengthen their conclusions.

You will find below a detailed list of my observations and questions.

**Specific comments:**

**Abstract:**

(general comment): could you state the range of conditions in which the presented results apply (at least a range of LWC and MVD and the type of shape/model characterizing the droplet size distributions generated at BIWT)

l17-18: about the agreement of 15 % in MVD: the validity range of this results should be indicated. For instance, regarding shadowgraphy, your experiment shows that the indicated 15% are only valid for MVD < 35μm, see discussion in section 4.

l21-22: (question) is it an agreement between the two techniques or an agreement of each of these techniques with the reference values calculated from the mass flow rate ? In the first case, the result should be discussed in the paper in order to be included in the abstract. In the second case, the conclusion need to be rephrased, because it seems to contradict the results presented in fig. 11, on which a significant number of the PDI values fall outside the ±20% cone. From discussion in section 4.3, LWC from the PDI may only be within 20% of the reference values in 65% of the cases (97 out of 280 test points, as estimated from the data provided in section 4.3) or fall into 1:1 correlation with ±43% (whatever this means) in 91% of the cases.

**Section 2:**

(General comment): Add in this section a comprehensive description of your experiment. It could be test matrices summarizing the test points in terms of W/T settings and environmental conditions, targeted MVD/LWC values, number of runs, duration and number of measurement points for each instrument.

(Suggestion): To facilitate its readability, the section could be subdivided into three paragraphs: 2.1 Description of the experimental setup (already existing), 2.2 Presentation of the test conditions (new, test matrix) and 2.3 Assessment of repeatability (group together all the already existing pieces of information mentioned throughout the paper)

l109: (suggestion) provide the fluctuation level (0.1 bar) in relative units.

l127-128: "Here, we indicate the distributions and their fits in the respective experiment": really good idea, but this has not been done, unfortunately.

l146: describe the test points and indicate the number of repetitions for each test point (use a test matrix for instance)

l147: regarding the repeatability: how do you calculate the "standard variation" for PSD? Is the standard variation equivalent to the coefficient of variation defined in equation 2?

l149: standard deviation (in g/m3) or coefficient of variation (in %)?
(suggestion) : as a complement to the comment l146, you could add a recap table (test matrix + table of results) containing test conditions, number of repetitions and statistical results (mean and standard deviation).

l159: include the test matrix here or in appendix. This is essential to give a comprehensive representation of the physical and statistical basis supporting this comparative study.

l165: This might be really interesting for your instrument assessment, since the measurement results might depend on particular instrument settings. State for each instrument, what parameters were varied and what are the results and conclusions?

**Section 3:**

l172: (general comment): can you indicate the general specifications of this instrument: size range (is it the static range in table 1?), velocity range, concentration range?

l186: in table 1: can you indicate the two setups (manufacturer settings, McDonell and Samuelsen 1990) in two different columns for the sake of clarity?

l191: (question) Is 5 % related to the differences obtained by repeating the tests with different user-controlled settings or is it just an indication of the repeatability of the PDI technique with McDonell and Samuelsen 1990 settings?

l192: $D_{32}$ is not defined. Is it comparable to MVD? What point do you intend to make by quoting the results of McDonell and Samuelsen 1994?

Equations (4) to (7): please make sure that each term in equations is properly defined (e.g. what is j in $t_{tran(i,j)}$ ?), so that your article is self-consistent.

l200: The reference Zhu 1993 is not in the reference list

l228: The FCDP is not used in the experiment reported by Voigt et al. 2017. Please remove this reference.

l233: Do you use data of the 21$^{st}$ bin (over-size bin) in your calculation? Do you use a binning different than the default one set by the manufacturer?

l235-236: Regarding uncertainty: when dealing with the FCDP, you assume implicitly that FCDP, CDP and even FSSP are truly equivalent, so that you can take conclusions derived from studies on FSSP/CDP as granted for FCDP. Although all these probes use the same measurement principle (forward light scattering) and may share a similar optical layout, they differ in many aspects (e.g. the "novel fast electronics" highlighted l259). Can you provide references to studies demonstrating clearly the strict equivalence between FCDP and CDP/FSSP?
If there are no such references available, please make it clear when you discuss uncertainty that you are referring to studies on CDP/FSSP probes for lack of more relevant references. Then just mention the most relevant ones.

l241: The 32-34% accuracy range reported in Baumgardner 1983 is likely not applicable to your study ("old" FSSP with limited electronics).
l243: I think the CDP tested in Lance et al. 2010 differs from the FCDP you use handle coincidence quite differently.

l247: The content of table 2 and/or the description of the measurement protocol has to be expanded (see App. C in Lawson et al. 2017) based on your own data processing settings (e.g. what is set in the setup.m file, binning options). Also, please indicate your calibration protocol.
(question): Does "DOF_crit = 0.9" mean that particles with Qual/Sig < 0.9 are discarded? How was the value (0.9) determined and did you assess the impact of this setting on MVD for instance?

l250-254: Could you be more precise in the description of the correction algorithms applied in post-processing? For instance, the 125% threshold in beam transit time is not directly mentioned in any of the three papers you quote.

(question) How do you estimate the transit time vs drop size relationship? Do you comply with the "Half peak transit times versus size" procedure proposed in the FCDP post-processing manual?

l260 (suggestion): this assertion needs to be quantified. It would make more sense to move it into section 4.

l277: can you indicate the "data rate" in table 3.

(general comment): Some of the "characteristic numbers" given in tables 1 - 3 are interesting, but it is hard to get a clear picture of the capacity of each setup due to the lack of common parameters. A recap table with comparable specifications such as size ranges, size resolution, sampled volume, concentration range, uncertainties, main characteristics and limitations … would be useful!

l310-324: (suggestion) to be move to section 2 to establish the repeatability of the test conditions.

l324: Can you provide quantitative estimates of the uncertainty in LWC derived from the wind tunnel settings (see also comment @l380)?

**section 4:**

l330: to support this assertion, you can either show it analytically or quote Lance et al. 2010 (best).

l332: (suggestion) I would remove this general statement drawn from Tropea 2011: it results from a broad overview of optical techniques and does not serve your work.

l334: about the title and the content of this section
(general comment) If we follow the logical construction of the paper, the repeatability of the test conditions is not a result, but a prerequisite for the comparison of measurement techniques. Since PDI is the reference method for assessing the repeatability of the test conditions, the discussion shall be moved in section 2.
(general comment): How do you define accuracy? In this study, "precision" sounds more appropriate than "accuracy".

l340: table 4: the test conditions and number of points underlying this table are not clearly stated. For instance: how is calculated the 5% value given in the cell (2,2) ? I assume this is the mean value of an unknown series of coefficients of variation, each obtained from several repetitions made at the same test points, but it needs to be clarified (test matrix).

l341: it is a good idea to assess the impact of the instrumental settings on the measured quantity. Please provide a detailed description of the setting being tested (test matrix…) and their impact on PSD or MVD/LWC.

l342-343: Unless the change in parameter settings is insignificant, it will make more sense to discuss separately the impact of different instrumental settings and the repeatability of the measurement techniques configured with "optimal" setting.

l357: "precision" rather than "accuracy"…
(question) For FCDP: did you investigate the impact of post-processing settings (inter arrival algorithm for instance) to retrieved PSD, as you did for the PDI?

l364-365: (suggestion) Are these two references useful here ? 1) The argument is already given line 330 (Lance et al. 2010) and 2) neither Baumgardner 1983 nor Tropea 2011 are actually dealing with the FCDP.

l376-381: this should be in section 2, in which the repeatability of the test conditions is discussed. The calculation of LWC from the wind tunnel settings and its associated uncertainty shall be discussed all in the section (experimental setup).

l383-384: this assertion should be moved to section 3.1, in which the PDI measurement techniques are introduced.
(Suggestion): is the reference to Basu et al. 2018 really relevant to this discussion? You've already provided enough convincing references related to the PDI measurement technique, while this one redirects the reader to a book dedicated in the first place to the physics of sprays for combustion and propulsion.

l391: (question) is 14 % the largest relative difference found between FCDP and PDI MVD (marked measurement in fig 9 left) over the entire dataset (43 data points as estimated from fig 9)?

l392 : (question) why is 5 µm the lower limit for comparing FCDP and PDI spectra? From tables 1 and 2 both PDI and FCDP seems to measure below 5µm.

l396-397: "A low sensitivity of the FCDP to larger particle sizes (> 30 µm) ….the PDI for large droplets" : what makes you think that FCDP has a low sensitivity to particles larger than 30 µm? Is it a well-known behavior of the FCDP probe? If yes, could you provide references supporting this assertion? Secondly, the argued velocity deficit for large droplets is hardly convincing: on fig 10 the density looks equally spread around unity for droplets below 50 µm (as far as I can see on my grey-printed scale picture). Finally, it would be really helpful for the reader to see how the PSD measured by PDI and FCDP differ, because at this point, one could argue that a lower MVD could either be caused by an overestimation of the number of particles in the small size bins (e.g. due to shattering), or more likely an underestimation at large end of the spectra due to poor statistics in the large size bins, as it is argued l430 during the PDI-shadowgraphy results discussion. Are the MVD values calculated from PSD integrated over the 120 sec duration reported l258 ?

l400: Have you conducted a sensitivity study, where the transit time filter is changed, in order to reach this conclusion?

l403: The references "Lance et al. 2012" and "Lance et al. 2017" are missing in the reference list.

l411: When you write "this effect can have a minor…": have you actually assessed the effect of shattering, if any? A possibility would be to count the number of particles removed by the arrival time algorithm (provided you enable it during post-processing). The Spec software package v14 (old) contains a Quality Check program allowing to plot particle counts after the noise, shattering, DOF and TT qualification filters are applied. Such an analysis would be more convincing than the cited literature.

l411: The reference Weigel et al. 2017 is not in the reference list

l413: The ice accretion shown on figure 3 is quite impressive. Is it just an extreme case shown for illustration purposes? How much time does it take for this ice accretion to build up and how close is it to the sampling volume? Could you please comment on FCDP operation in such off-design conditions: do you see variations in the measured size distributions as the ice shape grows? Do you discard data after some changes are noticed?

l416: the references to Faber et al. 2018 and Braga et al. 2017 may be misleading because neither PDI nor FCDP is included in these intercomparisons.

l422-423: Shadowgraphy instead of here? Could you double check the data

l422-423: 8 measurement points (>35 µm, 20% of your 40-point dataset according to l278) have been excluded for being consistently different (systematic underestimation) from the expected values. Discussing the discrepancy in the PDI and shadowgraphy results found for MVD above 35µm, you suggest that a technical limitation of the shadowgraphy technique makes it unable to measure PSD correctly (insufficient statistical sampling) but your main conclusions (l18 and l510) assert that MVD measured by shadowgraphy and PDI lies within 15 %. Judging from the stated $R^2$ coefficient, I presume that the 15% value is only applicable if the 8 data points are discarded from the analysis. Therefore a caveat should clearly state that this is only true for MVD < 35 µm.

(suggestion): your experiment reveals a practical limitation of the shadowgraphy technique (at least when configured as in your experiment):  this can be a valuable information for other W/T operators using this technique. Could you comment on whether or not this limitation is only applicable to your set up (low data

rates, small field of view) or whether it is general to shadowgraphy (field of view against resolution dilemma, laser flashing rate limits) and what kind of modifications could be made to improve sizing and counting of log-normally distributed droplets from 1 to 150μm (e.g.: how to improve the data rates)?

l428: Could you quantify "very low"? Data rate should be mentioned in table 3

l436-439: These general comments do not bring useful information at this point in the discussion. Suggestion to move these two sentences in section 3.

l468-469: "This can only be explained by higher particle number concentrations measured by the FCDP ": possibly yes, given that MVD from PDI and FCDP are very similar below 20μm. Please show the measured PSD for these test conditions.

l466: what is the mean absolute value of the relative error between $LWC_{FCDP}$ and $LWC_{WFR}$?

l466: (general comment) It is surprising that an instrument which only detects particles over the first third of the total size distribution overestimates the LWC! According to fig 13 right, largest overestimations (of factor of two) are registered for small MVD values, in which case FCDP measurement should be in principle most accurate (particles within its measurement range). The quoted references report overestimations ranging from 20% (Faber et al. 2018) up to a factor of 4 (Rydblom et al. 2018). Your study could potentially bring new insights and precisions on this matter, provided that the analysis is deepened. The fact that the conclusions in Lance et al. 2010 are opposite to yours (l484) raises once again the question: how far should CDP and FCDP probes be considered equivalent? If probes are truly comparable, why do your study reaches the opposite conclusions?

**section 5:**

l506: (suggestion) "test" instead of "boundary" conditions?

l509-510: The statement that the shadowgraphy values fall with 15% needs to be rephrased (range of validity, caveat about the low sampling rates, resolution vs sampling volume).
l512: "For the FCDP, the high sensitivity ... (>35 μm) was **hypothesized**", rather than determined, since the discussion in its current state is hardly conclusive.

l515: (suggestion) this is an important conclusion but could you rephrase this, so that the limitation of your shadowgraphy setup appears clearly (low sampling rate more likely) and if possible, provide some piece of advice to others on how to improve the performance of the shadowgraphy technique in such test conditions.

l521: quantify "significantly".

**page 29**

Fig 10: Can you quantitatively comment the hypothesis made l95 about the drop velocity with respect to the air speed based on the PDI data collected in various test conditions?

**Technical corrections** (compact listing of purely technical corrections, typing errors)

l52: Similarly, to the experiments conducted here, Ide (1999) compared: first comma to be deleted

l345: is the reference to section 4.1 correct or should it be section 3.1?

l401: reference instead of reverence

l476: $LWC_{WFR}$ rather than $LWC_{PDI}$. This typo error prompt me to ask whether or not "PDI" was meant l468 since the comparison is made with WFR in the first place?

l498: **t**he instead of The

l505/506: a good repeatability **of/?** the MVD…  (word missing)

---

## Author Comment (AC1) · 2 Oct 2020

**Response to the reviewer 2 of the manuscript: "Comparison of different droplet measurement techniques in the Braunschweig Icing Wind Tunnel" by Inken Knop et al.**

We thank the reviewer for the insightful constructive comments that have definitely improved the effectiveness and quality of the manuscript. Please find the brief summary of responses below. The relevant improvements are colored in red in the manuscripts.

**General comments:**
This technical paper reports on a wind tunnel experiment designed to compare several droplet measurement techniques. The experiment is conducted in the Braunschweig Icing Wind Tunnel where populations of supercooled droplets with size ranging from 1 to 150 μm are generated. The analysis focuses on Median Volumetric Diameter and Liquid Water Content, two key microphysical properties in the characterization of icing conditions.

The droplet measurement techniques involved, namely Phase Doppler Interferometry, shadowgraphy and FCDP, a commercial single particle counter, are commonly used by cloud physics and icing research groups. Nevertheless, there are still gaps in the understanding on their respective performances which is detrimental to the comparison of data produced by various research groups using different instruments. According to the authors, wind tunnel experiments offer a unique opportunity to test droplet measurement techniques in controlled and repeatable test conditions which in the end contributes to the
definition of measurement standards. Thus, the present study may bring some valuable contributions to the field and deserve a publication in AMT journal.
However, as highlighted in Kapulla et al. 2007, a thorough interpretation of the experimental results is necessary to draw a fair comparison between techniques based on different sizing and counting principles. In my opinion, this paper needs further elaboration regarding the presentation of the test conditions and the analysis of the data:
 - The article does not contain a test matrix summarizing the experiment and providing the following information: wind tunnel settings (temperature, LWC, airspeed …), number of runs for each test conditions, duration and number of points collected by each probe in each run. The information scattered in the article indicates that several directions have been investigated (e.g. measurements in various wind tunnel conditions or influence of some probe settings) and that the analysis is based on a substantial number of data points, but it is hard to identify clearly the scope of this experiment and the statistical soundness of its results.
Response: A summary table of minimum and maximum test conditions for all the measurements is added in the manuscript and the total test matrix is added as supplemental material.

 - MVD and LWC are inferred from particle size distributions (PSD) in all but two cases (rotating cylinder and tunnel air and water flow supply system settings). Given the importance of the measured PSD, the analysis given in section 4 should include a discussion on the PSD measured by the aforementioned techniques in the same test conditions. This would provide a solid basis for the subsequent interpretation of MVD and LWC results.
Response: The response of these techniques for a finer spray (MVD 14.5 μm) and a coarser (MVD 33.8 μm) is understood from the bin-wise droplet counts and the corresponding cumulative mass fractions in the additional plot in Figure 9. The trend of the PSD for all the three methods is almost similar up to 50 μm, the FCDP measured count is in general almost an order higher than the PDI. Although only a few droplets above 30 μm are found with shadowgraphy, their weight is enough to deviate the cumulative mass curve from the others.
In case of the FCDP microphysical properties of a spray such as MVD and LWC are higher order products derived from a sample statistics of droplet number and size. Uncertainties in the underlying measured parameter propagate, in the case of the LWC, with the order of three.
By constraining the FCDP's considered SA as a measure to constrain the measured droplet number and such the tendency for coincidence, decreases especially the counting statistics for larger droplets.

The droplet concentrations for FDCP and PDI are plotted in Figure 12. For the shadowgraphy technique, the droplet density is not obtained because of the difficulty in defining the probe volume. It can be seen that the FCDP and the PDI measurements give in most of the cases nearly the same number densities. The new plots are discussed in the manuscript.

I strongly encourage the authors to deepen the analysis in order to strengthen their conclusions.
Response: We strived to deepen the analysis and some of the new items in the revision are below.
- The droplet spectrum, count and the droplet concentrations of the measurement systems are compared on a finer and coarser spray (Section 4.1).
- The repeatability of the spray is quantified and the temporal stability over small samples is presented (Section 2.3)
- An attempt is made to quantify the LWC uncertainty by analogies from a similar system. (Section 2.3)

Many changes are made to improve the readability and make the paper self-consistent.

**Specific comments:**
**Abstract:**
(general comment): could you state the range of conditions in which the presented results apply (at least a range of LWC and MVD and the type of shape/model characterizing the droplet size distributions generated at BIWT)
Response: The general valid range of MVD and LWC for this study in now listed in the abstract. Regarding the cloud distribution there is no specific regulatory requirement for the droplet size distribution of these fine sprays. Typical droplet size distributions of the BIWT are shown in fig. 3 and fig. 5. These can best be described by a Rossin-Rammler distribution. However, the description and investigation of different distribution functions is not the goal of this study.

l17-18: about the agreement of 15 % in MVD: the validity range of this results should be indicated. For instance, regarding shadowgraphy, your experiment shows that the indicated 15% are only valid for MVD< 35µm, see discussion in section 4.
Response: The stated agreement between these three measurements is for the range MVD=8-35µm. This is now included in the abstract. Due to the maximum detectable diameter of the FCDP of 50µm no measurements are discussed beyond this. The reasons for deviation of the shadowgraphy are discussed.

l21-22: (question) is it an agreement between the two techniques or an agreement of each of these techniques with the reference values calculated from the mass flow rate? In the first case, the result should be discussed in the paper in order to be included in the abstract. In the second case, the conclusion need to be rephrased, because it seems to contradict the results presented in fig. 11, on which a significant number of the PDI values fall outside the ±20% cone. From discussion in section 4.3, LWC from the PDI may only be within 20% of the reference values in 65% of the cases (97 out of 280 test points, as estimated from the data provided in section 4.3) or fall into 1:1 correlation with ±43% (whatever this means) in 91% of the cases.
Response: This is an agreement of each of these techniques with the reference values calculated from the mass flow rate. Accordingly, the quantities $|E_{PDI-WFR}|$ and $|E_{rotCyl-WFR}|$ are presented in the manuscript. The deviation is found to be higher at MVD of 35µm. The sentences in the abstract have been changed to correctly summarize the results of the LWC measurements.

**Section 2:**
(General comment): Add in this section a comprehensive description of your experiment. It could be test matrices summarizing the test points in terms of W/T settings and environmental conditions, targeted MVD/LWC values, number of runs, duration and number of measurement points for each instrument.
Response: A summary table of tests maximum and minimum test conditions for the measurements is added in the manuscript and the total test matrix is added as supplemental material.

(Suggestion): To facilitate its readability, the section could be subdivided into three paragraphs: 2.1 Description of the experimental setup (already existing), 2.2 Presentation of the test conditions (new, test matrix) and 2.3 Assessment of repeatability (group together all the already existing pieces of information mentioned throughout the paper)
Response: Certainly, this improves the readability. The section is subdivided as follows:
2.1 Wind tunnel description
2.2 Parameters and Statistical Quantities for Comparison
2.3 Wind tunnel repeatability and uncertainty estimations
2.4 Test matrix

l109: (suggestion) provide the fluctuation level (0.1 bar) in relative units.
Response: The typical range of operating pressures is 2 to 5 bars, the 0,1bar fluctuation level is provided in %.

l127-128: "Here, we indicate the distributions and their fits in the respective experiment": really good idea, but this has not been done, unfortunately.
Response: The description of the distribution and their possible fits would open a new chapter that is out of the scope of the actual paper. The misleading sentence was deleted.

l146: describe the test points and indicate the number of repetitions for each test point (use a test matrix for instance)
Response: There are several repetitions of the tests, 3 example cases with each of them with 3 repetitions are presented in figure 5. The test conditions of all compared tests can be found in the test matrix that will be uploaded as supplement material.

l147: regarding the repeatability: how do you calculate the "standard variation" for PSD? Is the standard variation equivalent to the coefficient of variation defined in equation 2?
l149: standard deviation (in g/m3) or coefficient of variation (in %)?
Response: The sentences have been corrected. It is the coefficient of variation of the MVD. We have additionally added $R^2$ values for the cases shown in figure 5.

(suggestion): as a complement to the comment l146, you could add a recap table (test matrix + table of results) containing test conditions, number of repetitions and statistical results (mean and standard deviation).
l159: include the test matrix here or in appendix. This is essential to give a comprehensive representation of the physical and statistical basis supporting this comparative study.
Response: We have added a summary table of the conducted experiments in the paper and will upload the full test matrix as a supplement.

l165: This might be really interesting for your instrument assessment, since the measurement results might depend on particular instrument settings. State for each instrument, what parameters were varied and what are the results and conclusions?

Response: the discussion of the different parameter settings of each technique are not part of the paper. Our focus is on the comparison of the measurement techniques not on the investigation of every single system. We forward the reader to a lot of literature where these investigations can be found.

**Section 3:**

l172: (general comment): can you indicate the general specifications of this instrument: size range (is it the static range in table 1?), velocity range, concentration range?

Response: We added some additional essential specifications of the instruments in the tables 1-3 and the comparable specifications in table 4.

l186: in table 1: can you indicate the two setups (manufacturer settings, McDonell and Samuelsen 1990) in two different columns for the sake of clarity?

Response: the two setups used, differ in the focal length of the transmitter and the dependent variables fringe spacing and beam waist at probe volume, that are mentioned in table 1 with a backslash.

l191: (question) Is 5 % related to the differences obtained by repeating the tests with different user controlled settings or is it just an indication of the repeatability of the PDI technique with McDonell and Samuelsen 1990 settings?

Response: the 5% value is the one obtained by McDonell and Samuelsen by their tests of the PDI sensitivity to user-controlled settings.

l192: D32 is not defined. Is it comparable to MVD? What point do you intend to make by quoting the results of McDonell and Samuelsen 1994? Equations (4) to (7): please make sure that each term in equations is properly defined (e.g. what is j in ttran(i,j)?), so that your article is self-consistent.

Response D32 is the Sauter mean Diameter: a representative number for the ratio of the volume to the surface area, often used in industrial spray applications. We decided to delete the sentence with the results of McDonell and Samuelsen 1994 since it is treating the D32 that we have not used in our study. We further added the definition of all terms used in our equations.

l200: The reference Zhu 1993 is not in the reference list

Response: There was a mistake in the year, we have corrected it

l228: The FCDP is not used in the experiment reported by Voigt et al. 2017. Please remove this reference.

Response: The reference has been removed.

l233: Do you use data of the 21st bin (over-size bin) in your calculation? Do you use a binning different than the default one set by the manufacturer?

Response: Here we quote the overall measuring capabilities specified by the manufacturer. For a data evaluation and further analysis we excluded the over-size bin. Information from this bin has been qualitatively recognized as a hint for the amount of droplets sensed beyond the actual size range. A droplet size calibration has been performed for the FCDP using borosilicate and soda lime glass beads. It was decided to stick with the manufacturer's bin setting from the probe checkout protocol, in order to allow for an impartialed bin assignment.

l235-236: Regarding uncertainty: when dealing with the FCDP, you assume implicitly that FCDP, CDP and even FSSP are truly equivalent, so that you can take conclusions derived from studies on FSSP/CDP as granted for FCDP. Although all these probes use the same measurement principle (forward light scattering) and may share a similar optical layout, they differ in many aspects (e.g. the "novel fast electronics" highlighted l259). Can you provide references to studies demonstrating clearly the strict equivalence between FCDP and CDP/FSSP?
If there are no such references available, please make it clear when you discuss uncertainty that you are referring to studies on CDP/FSSP probes for lack of more relevant references. Then just mention the most relevant ones.
Response: We have to admit, that a more obvious distinction between the mentioned probes is favourable. To reference the FCDP to an FSSP and CDP might be misleading, without clearly stating its advantages. Due to a lack of a distinct study which directly compares CDP and FCDP, these probes specifics from FSSP and CDP have been employed.
The revised manuscript will point this out more clearly.

l241: The 32-34% accuracy range reported in Baumgardner 1983 is likely not applicable to your study ("old"FSSP with limited electronics).
Response: There is hardly any publication out there which explicitly gives an accuracy for forward scatter probes. The reviewer is totally right with her/his hint, that the quoted accuracy applies for early generation forward scatter probes.

l243: I think the CDP tested in Lance et al. 2010 differs from the FCDP you use handle coincidence quite differently.
Response: again it has to be brought to attention, that the quoted values only hold for a CDP, with an older optics and electronics.

l247: The content of table 2 and/or the description of the measurement protocol has to be expanded (see App. C in Lawson et al. 2017) based on your own data processing settings (e.g. what is set in the setup. m file, binning options). Also, please indicate your calibration protocol. (question): Does "DOF_crit = 0.9" mean that particles with Qual/Sig < 0.9 are discarded? How was the value (0.9) determined and did you assess the impact of this setting on MVD for instance?
Response:

| FCDP SN | SN06 |
| --- | --- |
| Calibration | as of 4/28/2017 |
| DoF criteria: | Qual/Sig Ratio>= 0.9 |
| SA | 0.09mm² |
| Transit Time method | SPEC integrated Gaussian technique |
| Shattered particle filter | Arrival time algorithm |

Operators manual are available on www.specinc.com/downloads. Matlab Software package FCDP_SP3C_V40 has been used for processing of raw files.

By selecting a Depth of Field criterion of Qual/Sig Ratio >=0.9 all droplet scatter events which do not meet this criterion are discarded. The size of the SA where droplets fulfil this respective criterion of >=0.9 has been determined within the scope of the probe calibration via a sensitivity area map using a droplet generator (Lance et al. 2010, Faber et al.,2018). A spatial resolution of this precision mapping has been 0.25mm along the laser beam direction and 0.03mm across the laser beam. Recorded particle by particle files that come with the newer electronics implemented in the FCDP, in contrast to the CDP, allows for a subsequent assignment of SA and DoFcrit pairs during post processing.
The realization of high droplet number concentrations and the increased possibility of coincidence urges the use of a high DoF ratio in order to target coincidence. The calibration specifies a DoF ratio of 0.9 as the peak value for this FCDP. SPEC recommends high DoF ratios also for accurate particle sizing.

l250-254: Could you be more precise in the description of the correction algorithms applied in post processing. For instance, the 125% threshold in beam transit time is not directly mentioned in any of the three papers you quote.

(question) How do you estimate the transit time vs drop size relationship? Do you comply with the "Half peak transit times versus size" procedure proposed in the FCDP post-processing manual?

Response: The initial step in order to reduce coincidence in high droplet number concentrations is to sharpen the DoF criterion. An additional filtering method to further reduces the influence of coincidence. SPEC provides a software module in Matlab (Vers10) with which the theoretical full peak transit time (TT) through a gaussian beam profile, depending on the droplet size and TAS can be fitted to the observed TT to size distribution from the measurement using the two fit parameter C1 and C3. Qualified scatter events that are outside the acceptance range, which is a deviation of more than 25% from this TT to size curve are regarded as coincident and are such discarded (SPEC inc. C1C3_V4 manual).

$$TT = \frac{2}{TAS}\sqrt{C1 * \log(D^2) + C3}$$

l260 (suggestion): this assertion needs to be quantified. It would make more sense to move it into section 4.

Response: Will be moved and discussed in section 4 as suggested

l277: can you indicate the "data rate" in table 3.

Response: The speed is essentially limited by the response of the camera in a single frame operation mode. The acquisition rate was 2.33 images per second. This is now mentioned in Shadowgraphy description in Chapter 3.3.

(general comment): Some of the "characteristic numbers" given in tables 1 - 3 are interesting, but it is hard to get a clear picture of the capacity of each setup due to the lack of common parameters. A recap table with comparable specifications such as size ranges, size resolution, sampled volume, concentration range, uncertainties, main characteristics and limitations … would be useful!

Response: We have added a table with the mean characteristics, together with the summary table of the conducted experiments.

l310-324: (suggestion) to be move to section 2 to establish the repeatability of the test conditions.

Response: We believe as WFR measurements itself is another measurement moving it will disturb the cohesion

l324: Can you provide quantitative estimates of the uncertainty in LWC derived from the wind tunnel settings (see also comment @l380)?

Response: The atomization physics of internal mixing nozzles are highly dependent on the supply pressures of water and air and the operating duty cycle. The fluctuations of these critical parameters lead to a higher uncertainty in the droplet sizes and the LWC, to some extent complemented by the uncertainties introduced by the wind tunnel performance.

The primary objective of this exercise is the probe inter-comparison for which the prerequisite is the repeatability or the reproducibility and the temporal stability. Accordingly, an emphasis is made on the repeatability of the wind tunnel and spray and an attempt is made to determine the uncertainty in LWC.

Firstly, the repeatability of the wind tunnel and nozzle input conditions are studied and plotted in figure 4. The precision limits for these variables for a sample run are also reported in section 2.3. The aero-thermal characteristics of the tunnel have already been calibrated according to the guidelines of SAE

ARP 5905 with the recommended instruments and uncertainties which is now included in the manuscript. Thus the temporal stability of the tunnel is guaranteed.

Secondly, the repeatability of the spray can be appreciated from the plots in Figure 5.

To better estimate the uncertainty of the spray, additional data from another new spray system (not part of this manuscript) shall be mentioned here. The new system is equipped with a high accuracy Coriolis flow meter (accuracy 0.2%), the data was used to formulate an empirical form for the LWC (the variable being the input conditions to the nozzles), the 95% confidence interval of the model with the measurements is considered as the systematic bias of the model, the highest fluctuations of the pressure are considered as precision terms and the root-summed-squared (RSS) uncertainty computed over a wide range of operating conditions was found to be 0,045 g m$^{-3}$, yielding an total uncertainty of the spray LWC of 10%. This value is slightly higher than the repeatability characteristics, which are given by the coefficient of variation of 7% of the thermal mass flow meters used in the present study.
Given this tunnel operational constraint of creating an LWC with uncertainty of 10%, the fluctuations in Figs 14-16 beyond that value can be attributed to the uncertainties of the individual measurement techniques.
We will mention these uncertainty considerations in Section 2.3 in the revised manuscript.

**section 4:**
l330: to support this assertion, you can either show it analytically or quote Lance et al. 2010 (best).
Response: We have referred in the revised manuscript to Lance et al. 2010.

l332: (suggestion) I would remove this general statement drawn from Tropea 2011: it results from a broad overview of optical techniques and does not serve your work.
Response: We have removed the statement.

l334: about the title and the content of this section
(general comment) If we follow the logical construction of the paper, the repeatability of the test conditions is not a result, but a prerequisite for the comparison of measurement techniques. Since PDI is the reference method for assessing the repeatability of the test conditions, the discussion shall be moved in section 2.
Response: we agree with the comment and have shifted the investigations of the precision to the according sections in chapter 2.

(general comment): How do you define accuracy? In this study, "precision" sounds more appropriate than "accuracy".
Response: Being a relative comparison with unaccounted biases it is appropriate to use precision instead of accuracy, changes are made wherever necessary.

l340: table 4: the test conditions and number of points underlying this table are not clearly stated. For instance: how is calculated the 5% value given in the cell (2,2)? I assume this is the mean value of an unknown series of coefficients of variation, each obtained from several repetitions made at the same test points, but it needs to be clarified (test matrix).
Response: The values of Table 4 are the mean values of the coefficients of variations, obtained from several repetitive measurements. This explanation is included at the beginning of chapter 3. The test matrixes of the repetitive measurements are also added as supplemental material.

l341: it is a good idea to assess the impact of the instrumental settings on the measured quantity. Please provide a detailed description of the setting being tested (test matrix…) and their impact on PSD or MVD/LWC.

l342-343: Unless the change in parameter settings is insignificant, it will make more sense to discuss separately the impact of different instrumental settings and the repeatability of the measurement techniques configured with "optimal" setting.

Response: the discussion of the different parameter settings of each technique are not part of the paper. Our focus is on the comparison of the measurement techniques not on the investigation of every single system. We forward the reader to a lot of literature where these investigations can be found.

A summary test matrix is now included in the manuscript and the detailed test matrix uploaded as supplemental material.

l357: "precision" rather than "accuracy"…

Response: has been replaced into "precision"

(question) For FCDP: did you investigate the impact of post-processing settings (inter arrival algorithm for instance) to retrieved PSD, as you did for the PDI?

Response: We have post processed the data under consideration of various filter techniques available in the Matlab postprocessing routine and assessed their influence on droplet number.

The inter arrival algorithm for instance was applied within the scope of a shattering filter. This filter has been applied although droplet number before and after was insensitive towards this inter arrival filter, which supports the conclusion that (maybe also caused by the presence of the anti shattering tips and with rather small droplets and no ice crystals) that shattering had no major role.

Additionally a variation of DoF criterion has been performed with 0.7 and mostly 0.8 and eventually 0.9.

l364-365: (suggestion) Are these two references useful here? 1) The argument is already given line 330 (Lance et al. 2010) and 2) neither Baumgardner 1983 nor Tropea 2011 are actually dealing with the FCDP.

Response: We agree, that this is a repetition of hinting towards the nature an error in LWC, when deriving it from droplet number and size. The subsequent references can be omitted.

l376-381: this should be in section 2, in which the repeatability of the test conditions is discussed. The calculation of LWC from the wind tunnel settings and its associated uncertainty shall be discussed all in the section (experimental setup).

Response: we agree with the comment and have shifted the investigations of the precision to the according sections in chapter 2.

l383-384: this assertion should be moved to section 3.1, in which the PDI measurement techniques are introduced.

Response: This has been moved and the advantages that make PDI more robust are discussed in section 3.1.

(Suggestion): is the reference to Basu et al. 2018 really relevant to this discussion? You've already provided enough convincing references related to the PDI measurement technique, while this one redirects the reader to a book dedicated in the first place to the physics of sprays for combustion and propulsion.

Response: the mentioned reference has been deleted.

l391: (question) is 14 % the largest relative difference found between FCDP and PDI MVD (marked measurement in fig 9 left) over the entire dataset (43 data points as estimated from fig 9)?

Response: The maximum difference in MVD is 14 % for all the 45 data points compared. The according sentence has been adapted in the manuscript.

l392: (question) why is 5 μm the lower limit for comparing FCDP and PDI spectra? From tables 1 and 2 both PDI and FCDP seems to measure below 5μm.

Response: The droplet size in PDI is obtained from the linear relations between the phase shift and size derived for a predominant reflection or refraction mode based on geometrical optics (Ofner 2001). Below 5 μm, the validity of the geometric optics tends to cease and the diffraction becomes significant leading to erroneous measurements if the linear relationships are used as mentioned in Chuang (2008). Bachalo and Sankar (1996) reported the uncertainty resulting from these oscillations to be under ±0.5 μm.

Chuang et al. propose using a large off axis angle for high accuracy of these small droplets but at the expense of the limiting the upper size.

A discussion of the above is included in the draft.

l396-397: "A low sensitivity of the FCDP to larger particle sizes (> 30 μm) ….the PDI for large droplets" : what makes you think that FCDP has a low sensitivity to particles larger than 30 μm? Is it a well-known behavior of the FCDP probe? If yes, could you provide references supporting this assertion?

Response: There are indeed hints of a lower sensitivity of the FCDP towards larger particles throughout various measurements, when comparing FCDP to CDP data e.g. ACTIVATE (current and ongoing NASA campaign) or in further wind tunnel tests with a FCDP, 2D-S combination at RTA, (Vienna, Austria) during the ICE GENESIS campaign. Unfortunately there is now reference available yet. A hint is available is the study by Thornberry et al. (2016), where the authors only use 12 size bins (out of the 21) up to only 24 μm for data evaluation. Larger sizes are covered by a 2D-S probe with a diode array resolution of 10 μm. Sizing (and imaging) capabilities of imager probes in this size range is subject to large errors (Baumgardner et al., (2017), …). Thornberry et al. (2016) even says while comparing the size range between 24μm-36μm of FCDP and 25μm-35μm of 2D-S respectively,

*"This change* (projected area of measured particles by 2D-S and FCDP) *in the relationship between the FCDP and 2-D-S is due to a greater decrease in the particle concentration measured by the FCDP in the 24–36 μm size range than that measured by the 2-D-S in the 25–35 μm bin."* So the change in his linear fit over the median projected area σ in FCDP measurements is attributed to a lower number concentration of larger particles >24μm compared to what the 2D-S has observed in the given size range.

But on the contrary Lawson et al. (2017) find a good agreement between the overlap region between FCDP and 2D-S.

Secondly, the argued velocity deficit for large droplets is hardly convincing: on fig 10 the density looks equally spread around unity for droplets below 50 μm (as far as I can see on my grey-printed scale picture).

Response: We compare velocity measurements from the PDI, a completely non-intrusive measuring technique, with an intrusive technique. With a surface area of the test section of 50cm x 50cm and a projected surface of the FCDP of approximately 171.69 cm² almost 7% ! of the cross sectional area are occupied by the probe itself, without taking a boundary layer within the test section into account (reducing the cross section by an assumed boundary layer thickness of 1cm yields a relative FCDP cover of 8%). Without assessing the stream lines around the FCDP and droplet trajectories in detail we might have to consider this effect, especially when comparing different measurement techniques. The figure below shows the velocity field across the centre plane through the SA, around a FCDP at a given true air speed of 200 m/s. Although our measurements were conducted at a lower TAS, we want to draw

the attention towards the point that the velocity field along a potential particle trajectory ahead of the probe arms, as well as directly where the SA is located is modified by the probe itself. This fluid simulation was conducted in a free flow environment and without the constraint of a test section. Velocity measurements with the PDI on the other hand are unobstructed and undisturbed by the probe itself.

[Figure]

[Figure]

Simulated droplet speeds for this specific wind tunnel setup vary with droplet diameter. According to this simulation this effect is pronounced for larger droplets (>100µm).

Ansys simulation results for single droplets accelerated with the 3D-airflow of the wind tunnel nozzle show for droplets >150µm a velocity deficit of 10% in the test section.

| Diameter in µm | simulated velocity at test section in ms$^{-1}$ |
|---|---|
| 160 | 36.07 |
| 200 | 34.86 |
| 240 | 33.73 |
| 280 | 32.73 |
| 320 | 31.92 |
| 360 | 31.36 |

In addition to the particle velocity plot (figure 2) we show in the above image the PDI velocity measurement results of a test case with only small droplets. The maximal velocity deficit is less than 5%.

Finally, it would be really helpful for the reader to see how the PSD measured by PDI and FCDP differ, because at this point, one could argue that a lower MVD could either be caused by an overestimation of the number of particles in the small size bins (e.g. due to shattering), or more likely an underestimation at large end of the spectra due to poor statistics in the large size bins, as it is argued.

Response (equal as above) : The response of these techniques for a finer spray (MVD 14.5 µm) and a coarser (MVD 33.8 µm) is understood from the bin-wise droplet counts and the corresponding cumulative mass fractions in the additional plot in Figure 9. The trend of the PSD for all the three methods is almost similar up to 50 µm, the FCDP measured count is in general almost an order higher than the PDI. Although only a few droplets above 30 µm are found with shadowgraphy, their weight is enough to deviate the cumulative mass curve from the others.

The droplet concentrations for FDCP and PDI is plotted in Figure 12. For the shadowgraphy technique, the droplet density is not obtained because of the difficulty in defining the probe volume. It can be seen that the FCDP and the PDI measurements give in most of the cases nearly the same number densities. The new plots are discussed in the manuscript.

l400: Have you conducted a sensitivity study, where the transit time filter is changed, in order to reach this conclusion?

Response: Settings for the transit time filter have been adjusted throughout the whole data analysis process, until coming up with the current and final settings. The motivation for this proposed conclusion is the observation that the C1C3 fit routine has a good agreement along the maximum occurrence of observed transit time to droplet diameter pairs for smaller droplet sizes. Advancing to larger droplet sizes the gradient of the fitted theoretical transit time versus droplet size curve gradually deviates from observed transit times. This observation is so pronounced that larger droplets along the observed TT vs droplet diameter distribution might fall beyond the acceptance range of 125% about the fitted theoretical curve. This brought us to the proposed conclusion that particle speeds of larger droplets seem to deviate more from the theoretical TT vs diameter curve. This instance can be adjusted and was partially accounted for by manually shifting the fitted curve along the TT-axis (accepting potentially more coincident particles and allowing more larger droplets into the acceptance range). Although having observed the variation in particle speed with droplet size, this effect might not sufficiently explain the declining sensitivity with larger droplet diameter. It is more likely that the reduced sensitivity might be promoted by a statistical underrepresentation due to the strict DoF criterion and the corresponding small size of the SA. Thornberry et al.(2016) can be quoted as reference.

l403: The references "Lance et al. 2012" and "Lance et al. 2017" are missing in the reference list.

Response: Lance 2012 included, Reference "Lance et al. 2017" is a typo

l411: When you write "this effect can have a minor…": have you actually assessed the effect of shattering, if any? A possibility would be to count the number of particles removed by the arrival time algorithm (provided you enable it during post-processing). The Spec software package v14 (old) contains a Quality Check program allowing to plot particle counts after the noise, shattering, DOF and TT qualification filters are applied. Such an analysis would be more convincing than the cited literature.

Response: The inter arrival algorithm for instance was applied within the scope of a shattering filter. This filter has been applied although droplet number before and after was insensitive towards this inter arrival filter, which supports the conclusion that (maybe also caused by the presence of the anti-shattering tips and with rather small droplets and no ice crystals) that shattering had no major role.

l411: The reference Weigel et al. 2017 is not in the reference list

Response: this reference will be added

l413: The ice accretion shown on figure 3 is quite impressive. Is it just an extreme case shown for illustration purposes? How much time does it take for this ice accretion to build up and how close is it to the sampling volume? Could you please comment on FCDP operation in such off-design conditions: do you see variations in the measured size distributions as the ice shape grows? Do you discard data after some changes are noticed?

Response: Ice accretion in this extent as shown in figure 3 lead to an interruption of the current test point since its effect on the surrounding flow is not quantified. Furthermore probe icing of this extent also indicated icing on the flow guiding vanes of the recirculation wind tunnel. During these breaks ice build-ups have been mechanically removed and the current test point subsequently repeated. Ice-build ups of this extent have only been observed after several test points in a row, with a certain build-up time.

l416: the references to Faber et al. 2018 and Braga et al. 2017 may be misleading because neither PDI nor FCDP is included in these intercomparisons.

Response: We will search for another reference in order to assess the PDI measurements. Unfortunately to our knowledge there is no single reference which juxtaposes both instruments.

l422-423: Shadowgraphy instead of here? Could you double check the data

Response: we have corrected the sentences and checked the data in the revised manuscript.

l422-423: 8 measurement points (>35 μm, 20% of your 40-point dataset according to l278) have been excluded for being consistently different (systematic underestimation) from the expected values. Discussing the discrepancy in the PDI and shadowgraphy results found for MVD above 35μm, you suggest that a technical limitation of the shadowgraphy technique makes it unable to measure PSD correctly (insufficient statistical sampling) but your main conclusions (l18 and l510) assert that MVD measured by shadowgraphy and PDI lies within 15 %. Judging from the stated R² coefficient, I presume that the 15% value is only applicable if the 8 data points are discarded from the analysis. Therefore a caveat should clearly state that this is only true for MVD < 35 μm.

Response: Yes the best linear fit of $MVD_{Shadowgraphy}=0.97 \cdot MVD_{PDI}$ is obtained by excluding the data points above 35 μm. Now this caveat is made explicitly both in abstract and conclusion.

(s

uggestion): your experiment reveals a practical limitation of the shadowgraphy technique (at least when configured as in your experiment): this can be a valuable information for other W/T operators using this technique. Could you comment on whether or not this limitation is only applicable to your set up (low data rates, small field of view) or whether it is general to shadowgraphy (field of view against resolution dilemma, laser flashing rate limits) and what kind of modifications could be made to improve sizing and counting of log-normally distributed droplets from 1 to 150μm (e.g.: how to improve the data rates)?

Response: The low data rate of Shadowgraphy setup is primarily from the camera speed and the laser. A tradeoff has to be made on the size resolution of the droplet to be captured, it should be noted that the intensity of the light source reduces with the square of the magnification factor of the teleconvertors and it leads to a point where the gradients between the background and shadow become weak and the lower thresholds specified would lead to large noise picked as smaller droplet, further the resultant area reduction also reduces the probability of the finest droplets being detected thus hampering the quality of the measurement. Higher resolution cameras and high intensity light sources will improve a better. A more detailed description of the Shadowgraphy setup is now included in the manuscript.

l428: Could you quantify "very low"? Data rate should be mentioned in table 3

Response: The data rate was approximately 2 frames per second. This is added in the description of the setup in Section 3.3.

l430 during the PDI-shadowgraphy results discussion. Are the MVD values calculated from PSD integrated over the 120 sec duration reported l258?

Response: the 120s duration was used for the FCDP measurements, since no online direct output of the counts is available. The MVD and LWC calculation was done over a time slot with temporal constant spray conditions (starting point of the evaluation after the ramp-up of the spray system). The FCDP samples consist thereby of at least 35000 droplets and in average of approx. 60000 droplets.

All the PDI measurements are made with at least 10000 droplets per sample. This led to a probe volume corrected total counts of 20000 for individual cases with low data rates at very low LWC and in average approx. 60000 counts, independent of the duration of the measurement. The MVD and LWC calculation was done over the entire data set, since the data recording was started always appr. 20s after the start of the spray system, so the ramp-up of the droplet cloud is not included in the results.

The Shadowgraphy measurements were done for at least 15 minutes to capture a minimum of 3000 droplets. This leads to more than 10000 counts with the applied DOF and border-correction. The data recording was started always appr. 20s after the start of the spray system, so the ramp-up of the droplet cloud in not included in the results.

l436-439: These general comments do not bring useful information at this point in the discussion. Suggestion to move these two sentences in section 3.

Response: The Sentences have been moved.

l468-469: "This can only be explained by higher particle number concentrations measured by the FCDP ":possibly yes, given that MVD from PDI and FCDP are very similar below 20μm. Please show the measured PSD for these test conditions.

Response: We have added a new figure (figure 9) to compare the PSD from the three measurement techniques and its discussion in the manuscript and as well the comparison of the measured particle concentrations of PDI and FCDP.

l466: what is the mean absolute value of the relative error between LWCFCDP and LWCWFR?

Response: As per the definition used in equation 3 it is 68,2%

l466: (general comment) It is surprising that an instrument which only detects particles over the first third of the total size distribution overestimates the LWC! According to fig 13 right, largest overestimations (of factor of two) are registered for small MVD values, in which case FCDP measurement should be in principle most accurate (particles within its measurement range). The quoted references report overestimations ranging from 20% (Faber et al. 2018) up to a factor of 4 (Rydblom et al. 2018). Your study could potentially bring new insights and precisions on this matter, provided that the analysis is deepened. The fact that the conclusions in Lance et al. 2010 are opposite to yours (l484) raises once again the question: how far should CDP and FCDP probes be considered equivalent? If probes are truly comparable, why do your study reaches the opposite conclusions?

Response: The FCDP's overestimation in LWC is promoted by high droplet number concentrations especially measured at small droplet sizes, as can be seen in the new Figures 9. Measuring conditions in the wind tunnel lie outside the customary environment in which the FCDP normally operates. References regarding FCDP's measuring capabilities are scarce.

The reference towards an opposite conclusion by Lance et al. (2010) was revised in the new Version of the manuscript and omitted. In detail Figure 7 in Lance et al. (2010) show a positive trend in LWC bias with increasing droplet number concentrations. Figure 15 shows simulated data where larger LWC biases are promoted by a high number concentration of small droplets rather than by larger droplets. Such a behavior indicates coincidence effects.

**section 5:**

l506: (suggestion) "test" instead of "boundary" conditions?

Response: It has been replaced

l509-510: The statement that the shadowgraphy values fall with 15% needs to be rephrased (range of validity, caveat about the low sampling rates, resolution vs sampling volume).
Response: It is revised with a caveat.

l512: "For the FCDP, the high sensitivity … (>35 μm) was **hypothesized**", rather than determined, since the discussion in its current state is hardly conclusive.
Response: It has been replaced

l515: (suggestion) this is an important conclusion but could you rephrase this, so that the limitation of your shadowgraphy setup appears clearly (low sampling rate more likely) and if possible, provide some piece of advice to others on how to improve the performance of the shadowgraphy technique in such test conditions.
Response: Improvements have been made in setup description and post processing description. Also some recommendation discussed previously are presented in section 3.3

l521: quantify "significantly"
Response: From figure 13 left it can be observed that LWC from FCDP varies by a factor of 0.5 to 3 of the LWC of WFR. Modified the lines to reflect the same.

**page 29**
Fig 10: Can you quantitatively comment the hypothesis made l95 about the drop velocity with respect to the air speed based on the PDI data collected in various test conditions?
Response: The velocity at the test section is computed with a computational model. At 40m/s of air speed, a 100 μm will be decelerated by the drag to a velocity of 37,8 m/s (5% deficit). As the droplet size decreases, the inertia of the droplet and the drag are negligible and therefore will have the same velocity as the surrounding air all along its path from injection to the test section. Accordingly, with tunnel fluctuations (±1,5%) and measurement errors, the smaller particle are expected to have a large band of normalized velocities, the same is being reflected in PDI. Larger particles have higher inertia and little less sensitivity to instantaneous fluctuation in the tunnel and also experience considerable drag that causes velocity deficit (5% for 100 μm) this demonstrates the consistency and robustness of PDI for velocity measurement.

**Technical corrections** (compact listing of purely technical corrections, typing errors)
Response: All of the typing errors are fixed

l52: Similarly, to the experiments conducted here, Ide (1999) compared: first comma to be deleted
l345: is the reference to section 4.1 correct or should it be section 3.1?
l401: reference instead of reverence
l476: LWCWFR rather than LWCPDI. This typo error prompt me to ask whether or not "PDI" was meant l468 since the comparison is made with WFR in the first place?
l498: **t**he instead of The
l505/506: a good repeatability **of/?** the MVD… (word missing)

---

## Author Comment (AC2) · 2 Oct 2020

**Response to the reviewer 1 of the manuscript: "Comparison of different droplet measurement techniques in the Braunschweig Icing Wind Tunnel" by Inken Knop et al.**

We thank the reviewer for the encouraging comments. We highly appreciate the outstanding knowledge and experience of the reviewer. Our responses to the suggested revisions can be found below. Changes in the manuscript are marked in red color.

This manuscript describes an icing tunnel test facility that may be of interest to the atmospheric science community. The fundamental operation of this system is similar to existing icing tunnels, however, there is no comparison to icing tunnels that have been in existence for nearly 50 years. Of particular interest is the icing tunnel in Ottawa, Canada that is operated by the National Research Council. A description of the NRC icing tunnel can be found at https://nrc.canada.ca/en/research-development/nrc-facilities/altitude-icing-wind-tunnel- research-facility, and early papers describing experiments in the tunnel are by Strapp and Schemenauer (1982) and King et al. (1985). The NRC icing tunnel has unique advantages over the Braunschweig tunnel, specifically the NRC tunnel is capable of particle speeds up to 100 m s$^{-1}$ and altitudes to 40,000 ft (12 km). The Braunschweig maximum particle speed of the Braunschweig tunnel is 40 m s$^{-1}$ and it has not capacity to simulate altitude. This is unfortunate since research aircraft fly at various altitudes and all large aircraft fly at speeds that are at least twice the maximum speed that the Braunschweig tunnel can produce. The manuscript needs to discuss how the limitations of the Braunschweig tunnel influence their results.

Response: There are indeed many icing wind tunnels worldwide. The authors are well aware of the NRC facilities and have many excellent collaborations with the research teams in Canada. However, the goal of our paper is not to perform a tunnel-intercomparison exercise. Instead, we focus on the intercomparison of several droplet sizing techniques in one tunnel, the Braunschweig Icing Wind Tunnel.

The Braunschweig Icing Wind Tunnel has many applications, not only limited to aircraft icing. Also the presented measurement techniques are not only limited to applications in civil aviation. Therefore, the maximum speed of the tunnel and its sea level pressurization do not impede the validity of the results presented.

Nevertheless, we want to mention that numerous scaling methods based on similitude of geometry, droplet trajectories and the impingement heat transfer have been developed to scale the model and test conditions to simulate the conditions beyond the IWT abilities. A comprehensive description of the scaling methods can be found in Anderson (2004). Scaling one of the parameters requires changes in other parameters to maintain the similitude, often it is not possible to concurrently match all the similarity parameters, especially the pressure which in most of the tunnels cannot be controlled. The AIWT operated by NRC and CIRA icing wind tunnel have the ability to control the pressure to simulate the high altitude icing conditions. Anderson (2004) reports the factors that highly are highly sensitive to pressure have only a limited influence on the parameters that influence the icing the most. From the icing data available at AEDC Barlett (1988) states the influence of pressure on icing is insignificant, the joint NRC CIRA experiments to study the effect of pressure (Oleskiw et al. 1996) showed "relatively small changes in the forward-facing portion of the profiles".

We have modified the paper to include some of the above mentioned comments, and also mentioned the NRC facility in the introduction section.

Understanding discrepancies between drop concentrations and drop size distributions (PSDs) measured by various probes is of critical importance for cloud physics, albeit icing studies rely more on bulk quantities such as MVD and LWC. MVD and LWC are presented and discussed in great detail, but there is almost no quantitative discussion of drop concentrations and PSDs from the PDI, FCDP and Shadowgraphy instrumentation. The manuscript needs to include additional Figures that show

correlations between reported drop concentrations and PSDs measured by PDI, FCDP and Shadowgraphy.

Response: The response of these techniques for a finer spray (MVD 14.5 μm) and a coarser (MVD 33.8 μm) is understood from the bin-wise droplet counts and the corresponding cumulative mass fractions in the additional plot in Figure 9. The trend of the PSD for all the three methods is almost similar up to 50 μm, the FCDP measured count is in general almost an order higher than the PDI. Although only a few droplets above 30 μm are found with shadowgraphy, their weight is enough to deviate the cumulative mass curve from the others.

The droplet concentrations for FDCP and PDI are plotted in Figure 12. For the shadowgraphy technique, the droplet density is not obtained because of the difficulty in defining the probe volume. It can be seen that the FCDP gives slightly higher number concentrations The new plots are discussed in detail the manuscript.

The PDI measures LWC using eqn. 4, which proportional to the product of total drop concentration and corrected volume mean diameter. The manuscript should also show how LWC compares using this technique with LWC computed by integrating the complete PSD. The comparison should be shown as a function of MVD and drop concentration.

Response: This could be an additional interesting consideration. However, it would only broaden the knowledge about PDI. In contrast, the focus of the paper is on the comparison of measurement techniques rather than the detailed investigation of a single measurement technique. Therefore, we refrain from integrating the proposed tests for LWC calculation of the PDI into our investigation.

Furthermore, the LWC computed using equation 4 has shown a good agreement with the results of the WFR, so the manufacturer default (equation 4) seems suitable for our application.

The manuscript compares LWC measurements from the FCDP, PDI and RCT using the WFR as a standard. It is implied that LWC using WFR as a standard is very repeatable, on the order of 7%. In Section 4 a statement refers to Section 3 as justification for this, but far as I can tell in Section 3, this repeatability comes from the literature, not from actual tunnel tests. Yet, I assume there were LWC repeatability tests, similar to the MVD tests shown in Fig. 5, so please point out where I missed the LWC repeatability tests or include Figures showing results from them.

For all the experiments involving the LWC computation, the water flow rate is recorded and the corresponding LWC is computed. In total LWC from WFR is computed for more than 400 individual cases, of which several sets have been repeated to determine the variance of the water flow meters. The mean coefficient of variation of these repetitions is calculated to be 7% as reported in section 2.3.

Now, that said, based on the large amount of scatter shown in Figs. 12 – 14, either the tunnel flow characteristics or the measuring techniques, or both, appear to be contributing much more variability to LWC than the 20% Figure quoted in the text. It would be useful to show a complete uncertainty analysis for the WFR and test instruments measurements, but this is likely to be outside the scope of this paper. However, the manuscript should address tunnel and instrument LWC uncertainties in a more rigorous manner, not just quote the literature.

Indeed, a full uncertainty analysis based on the detection physics of each measurement technique is out of the scope of our manuscript. Nevertheless, we want to provide some estimates here. The uncertainties of the tunnel flow characteristics and the measurement techniques are both acting on the results presented in Figs. 13-15.

Let us therefore comment on the tunnel characteristics first. Here, the aerodynamic performance of the tunnel and the liquid atomizers that produce the droplet cloud need to be considered.

1) Aerodynamics: The repeatability of the wind tunnel and nozzle input conditions are studied and plotted in Figure 4 The precision limits for these variables for a sample run are also reported in section 2.3. The aero-thermal characteristics of the tunnel have already been calibrated as per the guidelines of SAE ARP 5905 with the recommended instruments and uncertainties which is now included in the manuscript. Thus, the temporal stability of the tunnel is guaranteed.

2) Liquid Atomizers: the repeatability of the spray can be appreciated from the plots in Figure 5. The temporal stability can be seen in Figure 6.

The atomization physics is highly dependent primarily on the supply pressures of water and air and the operating duty cycle. The fluctuations of these critical parameters lead to a higher uncertainty in the PSD and the LWC. To better estimate the uncertainty of the spray, additional data from another new spray system (not part of this manuscript) shall be mentioned here. The new system is equipped with a high accuracy Coriolis flow meter (accuracy 0.2%), the data was used to formulate an empirical form for the LWC (the variable being the input conditions to the nozzles), the 95% confidence interval of the model with the measurements is considered as the systematic bias of the model, the highest fluctuations of the pressure are considered as precision terms and the root-summed-squared (RSS) uncertainty computed over a wide range of operating conditions was found to be 0.045 g m$^{-3}$, yielding an total uncertainty of the spray LWC of 10%. This value is slightly higher than the repeatability characteristics, which are given by the coefficient of variation of 7% of the thermal mass flow meters used in the present study.

Given this tunnel operational constraint of creating an LWC with uncertainty of 10%, the fluctuations in Figs 13-15 beyond that value can be attributed to the uncertainties of the individual measurement techniques.

We mention these uncertainty considerations in Section 2.3 in the revised manuscript.

The sample conditions of these tests (droplet concentrations sometimes exceeding 2000 cm$^{-3}$) are typically only found in polluted environments. This, plus the slow droplet speed, limit the usefulness of the results of these experiments. These limitations need to be discussed in detail in the manuscript.

Response: It is true that the conditions used in this experiment only covers the lower boundary for what e.g. the FCDP is intended for, particle speed wise and at the same time uses large droplet number concentrations, which increases the likelihood of coincidence.

The wind tunnel is also for other applications than simulating flight conditions e.g. icing on wind turbine blades etc.

Nevertheless, the overall special conditions and limitations of the probes are going to be discussed in more detail in the new draft.

The poor sampling statistics for drop diameters > 30 microns (Figs 4 and 7) definitely introduce uncertainties in the LWC results that need to be addressed more rigorously.

Finally, it should be pointed out that without a rigorous uncertainty analysis of the absolute accuracy of the tunnel, all of the quoted accuracies are not absolute, but instead relative. That is, in addition to the random error associated with tunnel properties, there is some degree of undetermined bias error that is not considered. This needs to be emphasized in the manuscript, albeit, hints of this are included in some of the references cited.

Response: The higher bounds of the PSD for some conditions in BIWT can be approximated for example with a Langmuir D type distribution where $D_{max}$ is 2.2·MVD. The expected $D_{max}$ for the spray in Figure 3 (prev Fig .4) is 26.2 µm. The measurement shows droplets above 28 µm contribute less than 0.05% of the total volume agreeing with the expectated thus confirming the validty of the measurement. When the droplet count above 28 µm is doubled the change is in $D_{30}$ is neglible from 9.38 µm to 9.48 µm that also results in negligible change in LWC. For this sample, the vicinity of 5.5 µm dominates the mass curve.

However, the presence of larger droplets in a small sample (1000 droplets) has a perceivable change in the MVD thus the $D_{30}$ (Figure 6 right). As the sample size is increased (above 10000), the presence of the large droplets is reliably accounted and the uncertainity in MVD and $D_{30}$ and LWC will be reduced as shown in Figure 6 left. Accordingly, all the PDI measurements are made with at least 20000 droplets per sample (for individual cases with low data rates at very low LWC) and in average approx.. 60000 droplets. With the FCDP the samples consist of at least 35000 droplets and in average of approx. 60000 droplets.

On the contrary, if the primary mode is not adequately sampled like the shadowgraphy data in Figure 9 left, any change in the count on the tail end of the distribution would alter the D30 and thus the LWC significantly. However, in the present paper, no LWC estimates are made from shadowgraphy.

The inherent complexity makes it impossible to derive a theoretical model for the PSD and lack of any other means in the present project to determine the actual PSD makes it difficult to quantify the bias. Albeit, the repeatability is higher as shown in Figure 5. Therefore, all of the measurements have some amount of unaccounted bias, but with a high precision thus a quality inter-comparison of the methods can be reliably made. These aspects are discussed explicitly in section 2.

**Specific Comments**:

1. Introduction

Page 2: When mentioning the cloud probes used by Ide (1999) and Cober et al. (2001), the manuscript should describe the resolution and size range of these probes so results can be compared with tunnel results.

Response: Ide (1999) performed the icing experiments in NASA IWT with MVD in the range 10 to 270 µm and velocities 22 to 112 m/s. LWC calculated by integrating the PSD spectra obtained from a combination of FSSP and OAP was reported to be significantly higher (1.2 to 2.7 times). than the LWC measured with icing blade an RCT. The large deviation was attribted to spectral broadening and coincidence errors. This is discussed in the manuscript in the introduction and in section 4.2.

The measurement ranges of the probes used by Cober et al. (2012) in the flight test campaign is also included in the manuscript.

2. Experimental Setup

Add a table (perhaps as a supplement) indicating the mean operating conditions for all of the data sets presented in this manuscript (Velocity, Temperature, RH, Air Pressure, Water Pressure, Water mass flow). This would be helpful to understand the scope of conditions for each type of drop measurement system. Also, list the number of data sets collected for each drop measurement technique (e.g., FCDP: 100 samples at 20 m s$^{-1}$, 200 samples at 30 m s$^{-1}$, 300 samples at 40 m s$^{-1}$).

Response: Certainly, this table would benefit in establishing the validity regions of these measurement and the degree of statistical reliability. The test conditions of all the runs are uploaded as a supplement and a short summary table is included in the paper.

Based on the data plotted in Fig. 11, it appears that the PDI datasets greatly exceed those of any other probe. This could be due to the high particle rejection rate and very low sample volume of the PDI. Please explain.

Response: LWC from WFR is computed for all the measurements along with PDI, FDCP and RCT and has therefore the largest number of measurements.

In this project PDI is used for reference calibration of the tunnel therefore more measurements are made with PDI. The reliability over a wide range of droplet sizes and the low acqisition time needed for a reliable sample enabled more than 300 measurements including repetitions.

On the other hand, due to the limited detectable droplet size range of FCDP and a limited time availability of the probe, only 34 valid LWC measurement were made with FCDP. Also, the measurements with RCT were limited to 37 different spray conditions due to the long measurement duration and the necessity to maintain extremely cold temperature (<-18°C).
The large disparity in number of measurements is purely from the above and not related to the low sample volume.

3.1. PDI
In this section it is noted that "The PVC has the greatest effect on the smallest size classes. Their influence on the LWC, on the other hand, is very small." We have performed an independent analysis of the FCDP data set from this icing tunnel and determined that the peak of the particle mass size distribution is between 8µm and 10µm. Therefore in this study the vast majority of drops are very small (see PDI PSD in Figure 4 for confirmation where the mode is ~6 um). Given these concerns, the manuscript should include more details of the PDI small drop corrections and better quantify the errors. From Chuang et al 2008: "At very small droplet sizes, diffraction can become significant relative to refraction, and lead to oscillations in the φ versus d relationship at the smallest drop sizes, primarily in the size range below 4µm, but with some effects up to ~8µm."
Response: Thanks for invigorating our discussion on this aspect. This will be critical for extremely fine sprays where the mode is observed below 8 µm. The droplet size in PDI is obtained from the linear relations between the phase shift and size derived for a predominant reflection or refraction mode based on geometrical optics (Ofner 2001). Below 5 µm, the validity of the geometric optics tends to cease and the diffraction becomes significant leading to erroneous measurements if the linear relationships are used as mentioned in Chuang (2008). Bachalo and Sankar (1996) reported the uncertainty resulting from these oscillations to be under ±0.5 µm.
Chuang et al. propose using a large off axis angle for attaining higher accuracy of these small droplets but at the expense of the limiting the upper size. A discussion of the above is now included in the manuscript in chapter 4.1.

3.2. FCDP
The description of the FCDP sample volume is fairly convoluted. It is sufficiently described as SV=SA*TAS, where the SA is defined by calibration for a fixed qualification criteria. The SA is defined by laboratory calibration like that described in Faber et al. (2018). See the Table's section of the review for more details.
Response: We agree that the description of the sample area can be facilitated.

In paragraph 3 it should be noted that the CDP and FCDP have similar operating principals, but the improved optics and electronics in FCDP allow for accurate sampling in higher particle concentrations (see comments on Table 2 below). The FCDP also differs from the older FSSP-100 probe in that the qualifier detector uses a slit aperture (200µm x 800µm), which was first introduced on the FSSP-300 probe with data described by Brenguier et al. (1998).
Response: We see that the FCDP is equipped with state of the art electronics and an advanced optics, compared to the CDP. When we reference system and operating specifics of the CDP, we do this as to show capabilities of an example for forward scattering spectrometer probes, without the intention to attribute its specifics to the FCDP. It is rather to put this measuring technique in general in comparison to the other techniques.
Furthermore comparable specific references solely applicable to the FCDP have not been found by the authors.
In the revised manuscript the difference between both systems, CDP and FCDP will be emphasized.

Lance et al. (2010) note that accurate sizing of the CDP instrument to ~200 cm$^{-3}$ before being influenced by coincidence. However, improvements in the CDP (new limiting apertures) increased accuracy such that only 27% undercounting is estimated at concentrations of 500 cm-3 (Lance 2012). This level of uncertainty is still problematic. The FCDP was designed to incorporate the improvements of the CDP as well as reduce particle coincidence (the dominant source of error) by reducing the laser beam waist from 200µm to 80µm). Flight tests indicate reasonable agreement for LWC between the FCDP and hotwire probes for small droplet concentrations as high as ~1000 cm-3. The conditions in this study exceed these typical atmospheric conditions, so the FCDP uncertainty for these high drop concentrations range is not well described.

Response: In our citations we address mainly sources related to CDPs. So that repeated references to the (older) CDP are made throughout this chapter. A thorough comparison of both probes specifics and a comparison to other forward scatter cloud probes lies beyond the scope of this paper. But we agree that the major improvements of the FCDP versus the CDP, namely the electronics and optics should be addressed on the course of this chapter, especially when it comes to coincidence.

A usable citation, where confidence towards droplet concentration measurements ~1000cm-3 is expressed would be of help. The regime in which the FCDP is operated in this experiment will be discussed.

In paragraph 3, the authors cite up to a 50% uncertainty from Baumgardner 2017, but it should be noted that this quoted uncertainty (10 to 50% for light scattering probes), includes "Mie ambiguity, collection angles, coincidence, nonsphericity and shattering." In this study all droplets are assumed to be spherical, and shattering is minimal for the FCDP, so the 50% uncertainty does not apply here.

Response: It would be better to emphasize that the quoted citation from Baumgardner, 2017 is a maximum value for generic particle forward scattering probes, including all limitations for this measuring principle, including internal and external factors, which contribute to the cited up to 50% uncertainty. It is a good hint from the reviewer to elaborate the composition of the overall uncertainty and to point out what really applies for this probe and this experimental setup.

3.3. Shadowgraphy

Overall the Shadowgraphy technique is poorly described. More details of the instrument and post-processing are required such that the test could be replicated and verified by another group.

Response: The description is improved in chapter 3.3 with the details of the optics and light source. A description of the calibration is made. The DoF and border correction terms are discussed. Some recommendations are made from the experience. A description of the post processing is also made in the manuscript. The equipment specifications are also appended in the corresponding table 3.

In the last sentence from this section the data inter comparison is considered" almost identical." This statement requires quantification.

Response: In total 35 measurements are made with shadowgraphy, only 20 have individual conditions. The remaining15 are either repetitions or measurements with change in velocity for the same spray settings. Sixteen samples have $MVD_{PDI}$ below 35 µm, which show the correlation of $MVD_{Shadow}=0.96·MVD_{PDI}$. The details on the measurement points are included in chapter 3.3 and the test matrix is added as a supplement.

3.4. Rotating Cylinder Technique

Stallabrass (1987) should be Stallabrass (1978)

Corrected

4.1. Repeatability

See General Comments.

The discussions of the wind tunnel temporal stability and its repeatability can be found in the new chapter 2.3. The analyses of the overall combined wind tunnel and measurement setup precision is now added at the end of each section in chapter 3.

Paragraph 2: "see section 4.1" within section 4.1, should be "see section 3.1".
Corrected
Paragraph 4: "see section 4.2" within section 4.1, should be "see section 3.2".
Corrected

4.2. Comparison
Paragraph 3: "A low sensitivity of the FCDP to larger particle sizes (> 30µm) may cause or contribute to the measured deviation of the FCDP with respect to the PDI for large droplets." Is there evidence that the FCDP has a low sensitivity to larger drops? If so, provide a reference. The number of sampled drops is relatively low, due to the small sample volume and low concentration of larger drops, but this is true for all single-particle devices, including the PVI. Also, with long runs in an icing tunnel this should not be an issue, assuming the tunnel properties are repeatable, as claimed in Section 4.1.
Paragraph 3: "The transit time filter applied to the FCDP data during post-processing to reduce coincidence causes a rejection of droplets that have a too long transit time compared to the mean reverence [sic] velocity and thus reduces the droplet size spectrum evaluated as valid by large droplets." Is there any evidence to support this assertion? If so, please provide the evidence, a reference, or sound physical explanation.
Response: In this case the authors refer to an effect not to be attributed to the FCDP probe itself, but to a slip of larger droplets within the airflow, compared to smaller droplets and the subsequent application of the transit time coincidence filter. Since droplet speed of larger droplets seem to be slightly lower than the airstream and in addition their sample number is overall fairly low, we have observed the tendency that the current transit time filtering for coincidence, based on the given TAS assumed as the particle airspeed, can lead to discarding of genuine counts of larger droplets, which deviate more than 25% from the C1C3 distribution, based on the given TAS.

Paragraph 5: "According to Lance et al. (2010), an additional source of error of the CDP might be the external geometry of the probe, which can alter the measured cloud particle size distribution."
This statement does not apply to the FCDP because it has "anti-shattering tips" that minimize droplet splashing, whereas the CDP Lance used did not (at that time) have anti-shattering tips. The CDP can be equipped with anti-shattering tips now.
Response: We have to thank the reviewer for pointing out the fact that both CDP (in its initial design) and FCDP differ among other points in probe geometry. One major advantage of the FCDP's shape is the application of anti-shattering tips. The challenge of droplet splashing is thus reduced. Nevertheless exposing in-situ probes into a droplet laden airstream alters the flow locally. As can be seen from the CFD-Simulation below and Spanu, et al. (2020), Weigl et al. (2016). When comparing different measuring techniques, this is an important factor to be mentioned, when discussing measurements.

[Figure]

4.3. Comparison of LWC measurements

See General Comments, also:

Paragraph 6: "This can only be explained by higher particle number concentrations measured by the FCDP compared to the PDI."

Please provide particle concentrations and size distributions for the FCDP and PDI for the relevant wind tunnel datasets.

Response: The droplet number densities acquired by FCDP and PDI show a good agreement, that is shown in figure 12. In average the FCDP gives little higher concentrations than the PDI, what might be a possible explanation for the higher LWC results of the FCDP. The compared size distributions are shown in Figure 9. Both new figures are discussed in Chapter 4.1.

Following the example in Figure 7, the authors should indicate when the PDI and FCDP sampling statistics are poor (<100 particles per bin) and possibly remove these data from consideration.

Response: Our typical wind tunnel droplet size distribution has in almost all cases a long end with only very few large diameter droplets, as discussed in section 2.4. This can also be found in the literature (Rudoff et al., 1993; McDonell and Samuelsen, 1996). The number of particles per bin is as well a question of the bin size. In figure 7 we choose a bin width of 2µm. Increasing the bin width would automatically lead to higher droplet counts per bin. With a minimum droplet count of 10000 per measurement we present in the paper only measurement data with statistically secured MVD values.

5.0. Summary

Page 17 Paragraph 2: "The characterization of cloud droplet distributions with particle sizes > 100µm poses new challenges for droplet measurement techniques." Some Optical Array Probes are well suited to particle measurement in this range, specifically the 2D-S, which is commonly utilized for icing tunnel measurements of larger drops.

Response: Measurements with the 2D-S have already been carried out at TUBS IWT, in the size range from 10 to 1280µm (Bansmer et al. 2018). These comments were made in regard to the SLD icing conditions, where the LWC is around (0-1 0.4 g m⁻³) with sizes often extending over 250µm.

The SLD clouds exhibit a bi-modal nature (Cober and Isac 2012) and the calibration of such a cloud in the wind tunnel requires to effectively capture the first mode (of small droplets) at 4-8µm as well as the second mode round 80-120µm. As discussed earlier the probes FCDP and FSSP have size limitations. Although shadowgraphy has no limitation on size simultaneously measuring both extremes of the SLD is highly challenging. The low LWC and broad spectrum of PSD of SLD conditions pose challenges for the individual measurement methods. Combinations like CCP, FSSP+OAP or others are to be employed. Although there are no such restrictions on PDI theoretically, the limited dynamic range and optimal selection of PMT voltage is difficult. Furthermore, the droplets in the cloud are sparse and gaining statistical confidence is more difficult than the conditions studied here.
This is briefly discussed in the revised manuscript.

References
Biter, C. J., et al., 1987: The drop-size response of the CSIRO liquid water probe. *J. Atmos. Oceanic Technol.* **4**, 359-367.
Response: Reference is now included in the manuscript.

Brenguier, Jean-Louis, et al., 1998: Improvements of droplet size distribution measurements with the Fast-FSSP (Forward Scattering Spectrometer Probe). *J. Atmos. Oceanic Technol.,* **15**, 1077-1090.
Response: Already cited, DOI link corrected

King, W. D., et al., 1985: Icing wind tunnel tests on the CSIRO liquid water probe. *J. Atmos. Oceanic Technol.,* **2**, 340-352.
Response: Already cited, DOI link corrected

Korolev, A., et al., 2013: Modification and tests of particle probe tips to mitigate effects of ice shattering. *J. Atmos. Oceanic Technol.,* **30**, 690-708.
Response: Reference is now included in the manuscript.

Lance, S., 2012: Coincidence errors in a cloud droplet probe (CDP) and a cloud and aerosol spectrometer (CAS), and the improved performance of a modified CDP. *J. Atmos. Oceanic Technol.,* **29**, 1532-1541.
Response: Reference is now included in the manuscript.

Strapp, J. W. and R.S. Schemenauer, 1982: Calibrations of Johnson-Williams Liquid Water Content Meters in a High-Speed Icing Tunnel. *J. Appl. Meteor.,* **21**, 98–108, https://doi.org/10.1175/1520-0450(1982)021<0098:COJWLW>2.0.CO;2
Response: Cited in the introduction

**Figures**
Figure 2: Add mean and variance values to each plot. It would also be helpful to add time-series data for the PDI, FCDP and Shadowgraphy (counts/sec or conc/sec). Perhaps it would be better to add the probes' time-series as a new Figure.
Response: We have added mean and standard variation in Figure 4. The water flow rate exhibits strong initial transient but stabilizes approximately after 15 seconds, this results in high variance in the water flow rate. A high precision, endurance and stability of other parameters can be appreciated from the low variance.

[Figure]

The time evolution of the droplet acquisition of PDI is plotted above, it can be seen that the data acquisition rate is fairly linear. Further the consistent pattern suggests two minutes should be long enough for a good measurement.
This Figure is not included in the manuscript. The temporal stability of the droplet cloud measured by the PDI can be seen in Figure 6 in the manuscript.

Figure 4: Add additional accumulated size distributions for the FCDP and Shadowgraphy. If possible, overlay distributions from all three methods on the same Figure indicating Concentration, LWC and MVD for each.
Response: We added a new figure (Figure 9) where we show and compare the accumulated size distributions of FCDP, PDI and Shadograhpy for two different droplet clouds.

Figure 5: Add R2 values.
Response: We calculated the correlation coefficients $R^2$ between all runs and added them in the caption of the figure to maintain the clear structure of the figure.
The high $R^2$ values show promising repeatability of the spray.

Figure 6: Based on the results in Figure 2, the conditions (Temp, RH) are not necessarily stable for the initial portion of the sample run. As such, it is hard to separate the MVD discrepancy as a function of true fluctuations vs counting statistics. Show size distributions for the initial 1000 droplets and the final 1000 droplets.
Response: The RH is initially unstable during start of test day, but over a few tests the tunnel will be saturated and can expect a stable humidity. We plotted the data for the initial 10000 and the final 10000 droplets in Figure 6 right and no large difference is found between them (overall 280000 droplets, duration 900 sec). This shows the temporal stability of the spray and complements the data in Figure 6.

Figure 7: This plot is very useful, but as with Figure 4 it should be amended to include lines for FCDP and Shadowgraphy. It may also be helpful to interpolate the higher resolution PDI data into the FCDP size bins for a more accurate comparison. Note that the FCDP size bins are chosen to smooth out Mie bumps and to improve sampling statistics for the larges drops. Add a dashed line to indicate the threshold for rejecting data due to inadequate data points in a bin (e.g., 100 counts bin-1).

Response: we have added a comparison of the droplet counts (fig. 12) as well as the relative cumulative volume curves in figure 9 for PDI, FCDP and Shadowgraphy.

Figure 9: These plots are useful, but additional plots should be added to show correlation with LWC and Concentration.
Response: The correlation of the LWC-ratio of the FCDP to the number concentration is shown in figure 15 right. We have additionally added a figure showing the number concentration from PDI and FCDP.

Figure 11: It hard to visually separate the WFR grey region from the PDI data points. Considering switching this plot to color or making individual scatter plots for PDI, FCDP and RCT
Response: We have changed plots to color.

Figure 14: Combine with Figure 12, and include sample plots for FCDP.
Response: We have combined the two plots and changed them to color.

**Tables**
Table 1: Include the model number of the PDI in the caption.
Response: The table is appended with additional important parameters

Table 2: Amend the table to include these values:
FCDP Beam Waist = 80µm
FCDP DOF Rejection Criteria = 0.9
FCDP Sample Area = 0.09mm^2
FCDP Size Range = 2-50µm
FCDP Serial number = 6
FCDP Calibration Date = 4/28/2017 (see sizing calibration curve below from manufacturer)

[Figure]

Response: thanks for sharing the data, appended the table

Table 3: Include details on the Shadowgraphy optical system similar to Table 1 for the PDI (wavelength, magnification, focal length, working distance, collection angle, etc.).
Response: Added details of the optics

Table 4: Define the table column variables in the caption.
Response: Changed the table to give a better overview and do not further use any variable names.

---

## Referee Report (RR1)

The Manuscript is improved with the addition of figures and descriptions suggested by both reviewers. In reading the comments from Reviewer 2, I see that she/he has carefully examined the manuscript and posed critical questions that need to be addressed. Unfortunately, in their reply, the authors have pushed back on the most critical questions from both reviewers and have only provided facile explanations.

This reviewer patently disagrees with the statements in the revised manuscript:

For the FCDP, the high sensitivity of the transit time filter to velocity differences of the droplets or a respective low sensitivity to larger particle sizes (>35 µm) was hypothesized.

Even though the qualifier, "was hypothesized" is added to the statement, there is no evidence to suggest that the FCDP has a low sensitivity to drops > 35 µm. The statement that conflates two inappropriate references is particularly misleading:

A hint is available is the study by Thornberry et al. (2016), where the authors only use 12 size bins (out of the 21) up to only 24 µm for data evaluation. Larger sizes are covered by a 2D-S probe with a diode array resolution of 10 µm. Sizing (and imaging) capabilities of imager probes in this size range is subject to large errors (Baumgardner et al., (2017), …). Thornberry et al. (2016) even says while comparing the size range between 24µm-36µm of FCDP and 25µm-35µm of 2D-S respectively,

*"This change* (projected area of measured particles by 2D-S and FCDP) *in the relationship between the FCDP and 2-D-S is due to a greater decrease in the particle concentration measured by the FCDP in the 24–36 µm size range than that measured by the 2-D-S in the 25–35 µm bin."* So the change in his linear fit over the median projected area σ in FCDP measurements is attributed to a lower number concentration of larger particles >24µm compared to what the 2D-S has observed in the given size range. But on the contrary Lawson et al. (2017) find a good agreement between the overlap region between FCDP and 2D-S.

The Thornberry et al. (2016) study was conducted in the TTL in regions with very low concentrations of ice particles (median value 18 $L^{-1}$ according to Thornberry et al. 2016). In striking contrast, the studies in the BIWT are with water drops in concentrations of 1 to 2 x $10^6$ $L^{-1}$. The sample volume of the FCDP is much smaller than the 2D-S for particles > 25 µm, so it is appropriate to use the 2D-S measurements for the larger particles in the TTL study. This has nothing to do with the ability of the FCDP to count drops > 25 µm in concentrations of 1 to 2 x $10^6$ $L^{-1}$. It is only because the 2D-S has better sampling statistics at the larger (> 25 µm) particle sizes. The Thornberry et al. (2016) reference simply does not apply to data collected in the BIWT.

The statement by Baumgardner et al. (2017) is also misrepresented. Gurganus and Lawson (2018) show that the largest counting uncertainty for the 2D-S is in the 10 and 20 µm bins, and that the uncertainty is significantly reduced at the 30-µm bin and larger. Also, the Baumgardner et al. (2017) article is referring to an OAP with 25 µm pixel resolution, so the first two bins with the largest uncertainties are the 25 and 50 µm bins.

There are several examples of good overlap between the FCDP, FFSSP and 2D-S probes in water and mixed-phase clouds.  The manuscript needs to delete the hypothesis that the FCDP is less sensitive to drops > 35 µm.  What the manufacturer of the FCDP does state, however, is that the FCDP erroneously sizes some drops > ~ 25 µm that are outside the DOF, and that these signals fall into the smallest size bins.  This can account for the higher concentrations of drops in the ~ 5 to 8 µm size range shown in Fig. 9.  However, the FCDP is also felt to be more sensitive to detection of actual drops in the 5 to 8 µm size range, so this is a conundrum.  On the other hand, as pointed out previously in the Chuang paper, the PDI may be less sensitive to drops in this size range.  The manuscript should only discuss these sizing anomalies and not jump to the conclusion that the FCDP is less sensitive to drops in the larger size bins. It should  also be highlighted that the FCDP (i.e., *FAST* cloud droplet probe), is designed for applications on research aircraft that fly at airspeeds from about 100 to 200 m s$^{-1}$, not in an icing tunnel at 40 m s$^{-1}$.

My main concern, however, is with the implication throughout the manuscript that the BWIT can be used for studies of aircraft icing. The seventh and eighth words in the Introduction are "aircraft icing".  While the sentence is valid, this sets the stage for less subtle inferences that the BWIT is suitable for studies of icing of aircraft with airspeeds of 100 m s$^{-1}$ and faster, which includes all commuter and transport class aircraft. A much more egregious statement is found in the Summary where the manuscript suggests that the BIWT can be equipped for studies of SLDs in accordance with Appendix O.  This is simply not true because aircraft that operate in icing conditions typically fly at 3 to 5 times faster than the maximum tunnel velocity.

The manuscript makes the argument that similitude (Anderson 2004) can be used to configure the BIWTfor studies at velocities higher than its maximum velocity of 40 m s$^{-1.}$ This does not appear to be a valid argument. The principal dimensionless numbers used in icing studies are the Reynolds number (*Re*) and Weber number (*We*). *Re* is the ratio of inertial to viscous forces and is proportional to airspeed.  A typical chord width of an airfoil that can be tested in the BWIT appears to be about 50 mm, which is roughly the cross section of the FCDP. *Re* for an airfoil that is 50 mm in width at an airflow of 40 m s$^{-1}$ at -10 °C is 1.6 x 10$^5$, which is considerably less than 2 x 10$^6$ reported in the manuscript.  An explanation of the calculation in the manuscript should be provided. *We* is the ratio of kinetic energy to surface tension and is proportional to airspeed squared.  *Re* largely controls the flow around an airfoil and *We* influences the shapes of accreted ice (along with the Nusselt number and other less significant factors).  In the figure below, Anderson (2004) shows that to achieve similitude for a  reference velocity ($V_R$) of 200 mph (89 m s$^{-1}$), the  scale (tunnel) velocity would have to be 400 mph (179 m s$^{-1}$) for a 4:1 ratio of actual airfoil chord to an airfoil in the tunnel. That is, the smaller the airfoil used in the tunnel, the larger the tunnel velocity has to be to achieve similitude.  Or in other words, at the maximum velocity of the BIWT tunnel (40 m s$^{-1}$), the size of the airfoil would be much larger than the BIWT.  Obviously, the BIWT cannot be appropriately used to study inflight aircraft icing, but may be suitable for icing on wind turbine blades, slow flying UAVs and power lines.  This needs to be explicitly stated in the manuscript, because as it is currently written, it appears that the reference to Anderson (2004) implies that a similitude approach can be used to study icing on aircraft wings. Also, the implication that the BIWT is in the same category as the Glenn and Ottawa high-speed icing tunnels is misleading.

[Figure]

The manuscript represents a considerable amount of work, and the authors are commended for their efforts. However, my recommendation is that this paper be retained as a Discussion paper.

---

## Author Response (AR2)

We thank the Reviewers for their detailed and distinct comments towards our second revision of the submitted work. Please find our responses below.

Reviewer 1:

The Manuscript is improved with the addition of figures and descriptions suggested by both reviewers. In reading the comments from Reviewer 2, I see that she/he has carefully examined the manuscript and posed critical questions that need to be addressed. Unfortunately, in their reply, the authors have pushed back on the most critical questions from both reviewers and have only provided facile explanations.

This reviewer patently disagrees with the statements in the revised manuscript:
For the FCDP, the high sensitivity of the transit time filter to velocity differences of the droplets or a respective low sensitivity to larger particle sizes (>35 µm) was hypothesized.
Even though the qualifier, "was hypothesized" is added to the statement, there is no evidence to suggest that the FCDP has a low sensitivity to drops > 35 µm.

This statement follows two observations which have been made during data analysis:

1) The application of the TransitTime coincidence correction software provided by SPEC shows, that larger droplets along the transit time to diameter distribution are getting discarded due to a deviating fit gradient at larger droplets. Mesh points for the fit function are positioned only close to smaller diameter, where the distribution curve shows a large change in gradient. Where the so derived fit curve holds for smaller particles, a larger deviation towards increasing diameter was detected.

2) Compared to the other instruments an underestimation of the FCDP with droplet sizes larger 35µm has to be considered. Hence the phrase "was hypothesized" was used. Other recent studies (by other authors), which involve a FCDP 2D-S combination find a similar behavior in flight conditions, thus outside an artificial environment of a wind tunnel. A publication addressing this is in preparation.

The statement that conflates two inappropriate references is particularly misleading:

A hint is available is the study by Thornberry et al. (2016), where the authors only use 12 size bins (out of the 21) up to only 24 µm for data evaluation. Larger sizes are covered by a 2D-S probe with a diode array resolution of 10 µm. Sizing (and imaging) capabilities of imager probes in this size range is subject to large errors (Baumgardner et al., (2017), …). Thornberry et al. (2016) even says while comparing the size range between 24µm-36µm of FCDP and 25µm-35µm of 2D-S respectively,

*"This change* (projected area of measured particles by 2D-S and FCDP) *in the relationship between the FCDP and 2-D-S is due to a greater decrease in the particle concentration measured by the FCDP in the 24–36 µm size range than that measured by the 2-D-S in the 25–35 µm bin."* So the change in his linear fit over the median projected area σ in FCDP measurements is attributed to a lower number concentration of larger particles >24µm compared to what the 2D-S has observed in the given size range. But on the contrary Lawson et al. (2017) find a good agreement between the overlap region between FCDP and 2D-S.

The Thornberry et al. (2016) study was conducted in the TTL in regions with very low concentrations of ice particles (median value 18 L$^{-1}$ according to Thornberry et al. 2016). In striking contrast, the studies in the BIWT are with water drops in concentrations of 1 to 2 x 10$^6$ L$^{-1}$. The sample volume of the FCDP is much smaller than the 2D-S for particles > 25 µm, so it is appropriate to use the 2D-S measurements for the larger particles in the TTL study. This has nothing to do with the ability of the FCDP to count drops > 25 µm in concentrations of 1 to 2 x 10$^6$ L$^{-1}$. It is only because the 2D-S has better sampling statistics at the larger (> 25 µm) particle sizes. The Thornberry et al. (2016) reference simply does not apply to data collected in the BIWT.

The statement by Baumgardner et al. (2017) is also misrepresented. Gurganus and Lawson (2018) show that the largest counting uncertainty for the 2D-S is in the 10 and 20 µm bins, and that the uncertainty is significantly reduced at the 30-µm bin and larger. Also, the Baumgardner et al. (2017) article is referring to an OAP with 25 µm pixel resolution, so the first two bins with the largest uncertainties are the 25 and 50 µm bins.

The reviewer clearly states that the FCDP's reduced sensitivity towards larger droplets would be speculative. In order to rule out a different meaning with the word "sensitivity" we also could propose: sample area size evoked reduced statistical counting efficiency, under application of the post processing parameter (DoF$_{crit}$). The term sensitivity is not intended to address a missizing in this droplet diameter range. In our view this effect is exactly pinpointed with an instrument's sizing efficiency or its sensitivity.

The Baumgardner et al. (2017) (fig 9-4) publication gives an example of a simulated 2D-S response to a 10 µm droplet, where (over-) sizing errors, due to out of focus transits through the laser beam result in overestimation between 110% - 190% (uncorrected) and 130% (corrected, mean), with significantly larger deviations towards farther away from the center of focus.

This is just below the regarded size bin range used by Thornberry et al. (2016) and, as pointed out by the reviewer, Gurganus and Lawson (2018) yield good agreements as low as in the 30µm size range for their lab calibration. However a newly published analysis by Oshea et al.(2020) address the presence of a small ice mode in ice cloud measurements by OAPs (in this study they refer to a CIP-15 and a 2D-S probe) might be caused by an intrinsic instrument error, immanent in this type of instrument.

We agree, concerning the Thornberry et al. (2016) publication, that the approach to prioritize a 2D-S with its larger DoF over the FCDP for cirrus measurements in the TTL, is a reasonable measure. Consequently, we no longer regard this citation as a watertight reference to support the point and refrain from using it in the manuscript.

There are several examples of good overlap between the FCDP, FFSSP and 2D-S probes in water and mixed-phase clouds. The manuscript needs to delete the hypothesis that the FCDP is less sensitive to drops > 35 µm. What the manufacturer of the FCDP does state, however, is that the FCDP erroneously sizes some drops > ~ 25 µm that are outside the DOF, and that these

signals fall into the smallest size bins. This can account for the higher concentrations of drops in the ~ 5 to 8 µm size range shown in Fig. 9.

We thank the reviewer to point this out. This effect is not uncommon among forward scattering probes e.g., the Cloud and Aerosol Spectrometer (CAS) manufactured by DMT.

However, the FCDP is also felt to be more sensitive to detection of actual drops in the 5 to 8 µm size range, so this is a conundrum. On the other hand, as pointed out previously in the Chuang paper, the PDI may be less sensitive to drops in this size range. The manuscript should only discuss these sizing anomalies and not jump to the conclusion that the FCDP is less sensitive to drops in the larger size bins.

The possible error in the PDI measurements due to its smaller sensitivity to small droplets is mentioned in line 479 with the reference to the publication of Chuang.

It should also be highlighted that the FCDP (i.e., FAST cloud droplet probe), is designed for applications on research aircraft that fly at airspeeds from about 100 to 200 $m\ s^{-1}$, not in an icing tunnel at 40 $ms^{-1}$.

Although this aspect already has been addressed in the present manuscript version, this consideration will be further highlighted. Nevertheless, the manufacturer SPEC Inc. certifies the capability of particle sizing within a velocity range between 10 -200 $ms^{-1}$.

My main concern, however, is with the implication throughout the manuscript that the BWIT can be used for studies of aircraft icing. The seventh and eighth words in the Introduction are "aircraft icing". While the sentence is valid, this sets the stage for less subtle inferences that the BWIT is suitable for studies of icing of aircraft with airspeeds of 100 $m\ s^{-1}$ and faster, which includes all commuter and transport class aircraft.

We made no such statement of "icing of aircraft with airspeeds of 100 $m\ s^{-1}$ and faster" in our manuscript. For us, it is clear that the Braunschweig Icing Wind tunnel is not intended to use for aircraft certification. It is a research tunnel, with a certain size, speed and operational envelope, which is specified in the manuscript and also in the references mentioned. Indeed, the BIWT is used for many research topics, including aircraft icing. We are looking back on a track record of nearly 10 years of collaboration with airframers. For many investigations, it turned out, that it is not mandatory to go up to the flight speed with the investigations.

A much more egregious statement is found in the Summary where the manuscript suggests that the BIWT can be equipped for studies of SLDs in accordance with Appendix O. This is simply not true because aircraft that operate in icing conditions typically fly at 3 to 5 times faster than the maximum tunnel velocity.

We made the statement "future plans are to further enhance the capacity of the Braunschweig Icing Wind Tunnel's spray system to generate bimodal droplet size distributions according to EASA CS 25 Appendix O." We do not say anything about tunnel speed. Our approach of upgrading the tunnel with the mentioned capabilities is supervised by airframers their TIER-1 suppliers. We trust their experience and judgement.

The manuscript makes the argument that similitude (Anderson 2004) can be used to configure the BIWT for studies at velocities higher than its maximum velocity of 40 m s$^{-1}$. This does not appear to be a valid argument. The principal dimensionless numbers used in icing studies are the Reynolds number ($Re$) and Weber number ($We$). $Re$ is the ratio of inertial to viscous forces and is proportional to airspeed. A typical chord width of an airfoil that can be tested in the BWIT appears to be about 50 mm, which is roughly the cross section of the FCDP. $Re$ for an airfoil that is 50 mm in width at an airflow of 40 m s$^{-1}$ at -10 °C is 1.6 x 10$^5$, which is considerably less than 2 x 10$^6$ reported in the manuscript. An explanation of the calculation in the manuscript should be provided. $We$ is the ratio of kinetic energy to surface tension and is proportional to airspeed squared. $Re$ largely controls the flow around an airfoil and $We$ influences the shapes of accreted ice (along with the Nusselt number and other less significant factors). In the figure below, Anderson (2004) shows that to achieve similitude for a reference velocity ($VR$) of 200 mph (89 m s$^{-1}$), the scale (tunnel) velocity would have to be 400 mph (179 m s$^{-1}$) for a 4:1 ratio of actual airfoil chord to an airfoil in the tunnel. That is, the smaller the airfoil used in the tunnel, the larger the tunnel velocity has to be to achieve similitude. Or in other words, at the maximum velocity of the BIWT tunnel (40 m s$^{-1}$), the size of the airfoil would be much larger than the BIWT.

Our exact words are "To further exceed the operational envelope of the tunnel, numerous scaling methods based on similitude of geometry, droplet trajectories and the impingement heat transfer are available.", we then further say, "In the present study, we do not apply any scaling to the results in order to avoid introducing additional sources of uncertainty to our results.".

Since in the present manuscript, we are investigating the comparison of different droplet measurement techniques, and not aircraft icing, we should not engage into a discussion of scaling laws of aircraft icing. Nevertheless, we want to give some clarifications on the comments of the reviewer.

Typically, the chord lengths of airfoils that are investigated in icing wind tunnels are much larger than those you would apply in classical aerodynamic testing. One example can be found for instance on the homepage of the NASA Icing Research Tunnel (https://www1.grc.nasa.gov/facilities/irt/, accessed on Jan 15, 2021). The reason for that is that icing researchers want to achieve similitude at those locations, where ice accretion happens, at the leading edge.

It is misleading from the reviewer to state the Reynolds number and the Weber number are the principle dimensionless numbers of icing studies. Because the scaling of icing studies is such a difficult task, many more parameters need to be considered, e.g. the stagnation line freezing fraction and the accumulation parameter.

Obviously, the BIWT cannot be appropriately used to study inflight aircraft icing, but may be suitable for icing on wind turbine blades, slow flying UAVs and power lines. This needs to be explicitly stated in the manuscript, because as it is currently written, it appears that the reference to Anderson (2004) implies that a similitude approach can be used to study icing on aircraft wings.

The BIWT was appropriately used for several studies with commercial airframers in the past. We also refer to the above argumentation.

Also, the implication that the BIWT is in the same category as the Glenn and Ottawa high-speed icing tunnels is misleading.

We made no statement of "categories" in our manuscript. The reviewers brought up the icing wind tunnel facilities at NRC in the previous review, and therefore we mentioned them in the manuscript, highlighting the excellent collaboration with the researchers at NRC. Indeed, there is a high appreciation between the icing researchers at NRC, NASA and those in Europe, including their complementary experimental facilities.

Reviewer 2:

Most of the reviewer's comments have been taken into account and the authors did provide comprehensive answers to my questions. I think both the article and the response to reviewer include relevant information and interesting discussions for everyone in the field. Therefore, it is my belief that this article is worth being published in AMT.

Minor/technical correction:

section 2.1: the reader is referred to section 4.1 for an analysis of Fig. 2 but I couldn't find any discussion of Fig. 2 in the section 4.1.

We now included a short discussion of Fig. 2 in section 2.1 and deleted the reference to section 4.1.

section 2.2: (suggestion) depending on the final text formatting, it is sometime helpful to insert a reference to an equation in the sentence where it is defined, such as "The repeatability of measurements is characterized based on the coefficient of variation (σ) i.e., the standard deviations over several repeated measurements (n) normalized by the mean values as in eq. (2)"

We included at some positions the proposed references to equations and added also the used symbols.

section 3.2: (typo) MWD-> MVD (typo)

Corrected.

fig. 9 and discussion related to it in section 4: the "PDIFCDP" notation is unclear: is it PSD measured by the PDI during PDI vs FCDP tests ?

We now included an explanation at the beginning of section 4.